# Darwin Gödel Machine: Open-Ended Evolution of Self-Improving Agents

**Jenny Zhang**[*,1,2]   **Shengran Hu**[*,1,2,3]   **Cong Lu**[1,2,3]   **Robert Lange**[†,3]   **Jeff Clune**[†,1,2,4]
[1]University of British Columbia   [2]Vector Institute   [3]Sakana AI   [4]Canada CIFAR AI Chair
`{jennyzzt,srhu,conglu}@cs.ubc.ca, robert@sakana.ai, jeff.clune@ubc.ca`

## Abstract

Most of today's AI systems are constrained by human-designed, fixed architectures and cannot autonomously and continuously improve themselves. The scientific method, on the other hand, is a cumulative and open-ended system, where each innovation builds upon previous artifacts, enabling future discoveries. There is growing hope that the current manual process of advancing AI could itself be automated. If done safely, such automation would accelerate AI development and allow us to reap its benefits much sooner. This prospect raises the question of how AI systems can endlessly improve themselves while getting better at solving relevant problems. Meta-learning can automate the discovery of novel algorithms, but is limited by first-order improvements and the human design of a suitable search space. The Gödel machine (Schmidhuber, 2007) proposed a theoretical alternative: a self-improving AI that repeatedly modifies itself in a provably beneficial manner. Unfortunately, proving that most changes are net beneficial is impossible in practice. We introduce the Darwin Gödel Machine (DGM), a novel self-improving system that iteratively modifies its own code (thereby also improving its ability to modify its own codebase) and empirically validates each change using coding benchmarks. Inspired by Darwinian evolution and open-endedness research, the DGM grows an archive of generated coding agents. It samples agents from this archive, which self-modify to create new, interesting versions of themselves. This open-ended exploration forms a growing tree of diverse, high-quality agents and allows the parallel exploration of many different paths through the search space. Empirically, the DGM automatically improves its coding capabilities (e.g., better code editing tools, long-context window management, peer-review mechanisms), increasing performance on SWE-bench from 20.0% to 50.0%, and on Polyglot from 14.2% to 30.7%. Furthermore, the DGM significantly outperforms baselines without self-improvement or open-ended exploration. All experiments were done with safety precautions (e.g., sandboxing, human oversight). Overall, the DGM represents a significant step toward self-improving AI, capable of gathering its own stepping stones along a path that unfolds into endless innovation.

## 1 Introduction

Scientific progress is cumulative and open-ended, with each breakthrough standing on the shoulders of countless prior insights. In the same way, our most advanced AI systems are built upon a long lineage of innovations. For instance, transformers (Vaswani et al., 2017), the backbone of current large language models (LLMs) (Brown et al., 2020), did not emerge in isolation but were built upon years of past innovations, such as recurrent neural networks (Linnainmaa, 1970; Amari, 1972; Hopfield, 1982; Rumelhart et al., 1985) and attention mechanisms (Schmidhuber & Huber, 1990; Bahdanau et al., 2015; Kim et al., 2017; Parikh et al., 2016). However, most of today's AI systems remain bound by fixed, human-designed architectures that learn within predefined boundaries, without the capacity to autonomously rewrite their own source code to self-improve. As a result, each advancement in AI development still leans heavily on human interventions, tethering the pace of progress. This paper investigates the intriguing possibility of safely automating the search for ever-better AI. One can

---

[*]co-first authors    [†] co-senior authors

imagine an AI system that, like scientific discovery itself, becomes an engine of its own advancement: building upon its past, recursively improving, and propelling itself toward more advanced capabilities.

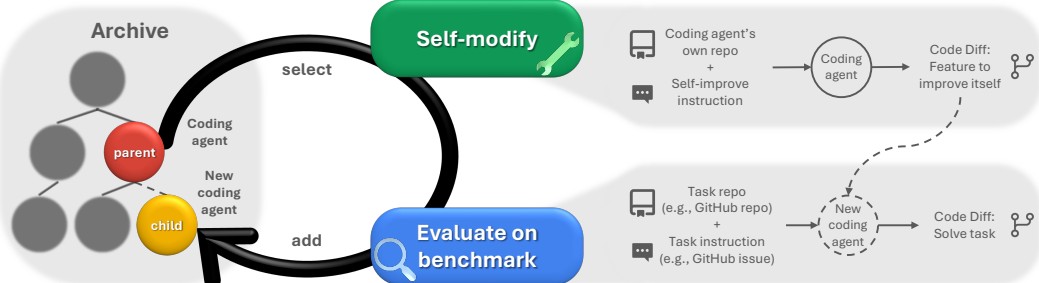

Figure 1: **Darwin Gödel Machine.** The DGM iteratively builds a growing archive of agents by interleaving self-modification with downstream task evaluation. Agents in the archive are selected for self-modification through open-ended exploration.

Schmidhuber (2007) presented a class of mathematically rigorous, self-referential, self-improving problem solvers. It relies on formal proofs to justify code rewrites, ensuring that any self-modification is provably beneficial. However, in practice and without restrictive assumptions about the system, it is impossible to formally prove whether a modification to an AI system will be beneficial. For example, while it may seem that an LLM-based coding agent would benefit from access to more tools (e.g., code search, test runners), the actual impact depends heavily on the model's training and task context (e.g., a testing tool that is optimized for one setup may confuse the agent when working with others). Instead of requiring formal proofs, we empirically validate self-modifications against a benchmark, allowing the system to improve and explore based on observed results. This approach mirrors biological evolution, where mutations and adaptations are not verified in advance but are produced, trialed, and then selected via natural selection. We also take inspiration from Darwinian evolution (Darwin, 2023) and investigate the effectiveness of maintaining a library of previously discovered agents to serve as stepping stones for future generations.

We propose the **Darwin Gödel Machine (DGM)**, a self-referential, self-improving system that writes and modifies its own code to become a better coding agent. Each self-modification requires the DGM to edit its own codebase. We use Python, which is Turing-complete, giving the DGM the potential to build any computable machine. Our framework envisions agents that can rewrite their own training scripts (including training a new foundation model (FM)). However, we do not show that in this paper, as training FMs is computationally intensive and would introduce substantial additional complexity, which we leave as future work. Instead, this paper focuses on improving the design of coding agents with frozen pretrained FMs (e.g., tool use, workflows). The DGM alternates between self-modification and evaluation phases. During the self-modification phase, selected coding agents from the archive generate modified versions of themselves. During the evaluation phase, each modified agent is tested on a coding benchmark, estimating the agent's coding capabilities, and then added to the archive. By improving its own capabilities through this loop, the DGM becomes better at both solving coding tasks and making future self-improvements. A key assumption is that an increase in performance on coding benchmarks indicates better coding capabilities, and hence better ability to self-modify and self-improve. Furthermore, the DGM maintains an archive of generated coding agents, initialized with only one agent, and continuously accumulates all generated variants over time. To support continual self-improvement, the DGM draws inspiration from open-endedness research (Wang et al., 2019; Fernando et al., 2024; Faldor et al., 2025), accumulating diverse stepping stones (i.e., interesting yet suboptimal solutions or features that may enable future breakthroughs). This open-ended exploration encourages the discovery of novel and potentially useful self-modifications beyond immediate performance gains.

We present results on two coding benchmarks: SWE-bench (Jimenez et al., 2024) and Polyglot (Paul Gauthier, 2024). The DGM automatically improves itself from 20.0% to 50.0% on SWE-bench, and from 14.2% to 30.7% on Polyglot. We show that self-improvement enables continued progress, as the DGM outperforms the baseline where the same base agent is repeatedly used to modify and generate new agents without self-improvement. We also show that open-ended exploration and

keeping an archive of all previously generated agents lead to the discovery of better coding agents. The DGM outperforms the baseline of not having open-ended exploration (i.e., a baseline without the accumulation of an archive of interestingly different stepping stones), where the coding agent always builds off the most recent version of itself. Overall, the DGM represents a step toward AI systems that can build upon their own prior innovations and improve recursively. We consider and discuss safety aspects extensively, including sandboxing and traceability of self-modifications, to ensure responsible experimentation (Section 5). By advancing the possibility of safe, self-referential, self-improving models, the DGM moves us closer to AI that not only learns but evolves in an open-ended, self-accelerating trajectory, much like science itself.

## 2 RELATED WORK

**Open-Endedness.** A grand challenge for driving unbounded innovation is designing open-ended AI systems that continuously generate novel and learnable artifacts (Stanley et al., 2017). Hughes et al. (2024) characterized open-endedness as a system's capacity to generate sequences of artifacts that are both novel and learnable from an observer's perspective. A central difficulty lies in structuring and exploring vast search spaces to consistently produce artifacts that are interesting to humans (Clune, 2019; Jiang et al., 2023). Early progress drew on quality-diversity algorithms, goal-directed exploration, intrinsic motivation, and learning-progress frameworks (Pugh et al., 2016; Ecoffet et al., 2019; Lehman & Stanley, 2011; Oudeyer et al., 2007), while recent advances leverage large-scale foundation models (FMs) as proxies for human interestingness and versatile engines for generating and evaluating novel behaviors across diverse domains (Brown et al., 2020; Hu et al., 2025; Zhang et al., 2024b). However, these approaches have yet to close the self-referential self-improvement loop, meaning improvements on downstream tasks do not translate into enhanced capabilities for self-modification or the acceleration of further innovations. We aim to mimic the acceleration of science and technology, where new tools and discoveries catalyze the creation of even more discoveries. How can we emulate nature's arc of evolution, which bends not only toward complexity but also an ever greater capacity to evolve (Dawkins, 2019; Gerhart & Kirschner, 2007; Hendrikse et al., 2007)?

**Meta-Learning FM Agents.** Many FM-based agents are handcrafted. Some building blocks include prompt engineering (Chen et al., 2023; Schulhoff et al., 2024), chain-of-thought (Wei et al., 2022; Yao et al., 2023; Hu & Clune, 2024; Guo et al., 2025; Lightman et al., 2023; Muennighoff et al., 2025; Zelikman et al., 2024a), self-reflection (Shinn et al., 2023; Yao et al., 2023; Madaan et al., 2023), multi-agent debate (Zhuge et al., 2023; Liang et al., 2023; Khan et al., 2024), memory (Liu et al., 2023; Zhong et al., 2024; Modarressi et al., 2023), temperature sampling (Zhu et al., 2024), and retrieval augmented generation (Lewis et al., 2020). The manual composition of these components limits the system's abilities to the ingenuity of its human designer. More recently, several meta-learning approaches have emerged that leverage FM to automatically optimize prompts (Fernando et al., 2024; , FAIR; Khattab et al., 2023; Cheng et al., 2024; Yuksekgonul et al., 2024; Yuan et al., 2024) and design agentic modules (Zhang et al., 2024c; Zhou et al., 2024; Yin et al., 2024; Zhuge et al., 2024; Rosser & Foerster, 2025; Zhang et al., 2025a; Ye et al., 2025; Gao et al., 2025; Nie et al., 2025; Su et al., 2025; Zhang et al., 2025b; Niu et al., 2025). The Automated Design of Agentic Systems (ADAS, Hu et al., 2025) iteratively generates downstream agents with a fixed meta-agent, evaluates them against a target benchmark, and incorporates feedback to refine subsequent generations. In contrast, the DGM is a single system that both solves downstream tasks (i.e., coding problems) and refines its own implementation (i.e., its codebase), removing the need for a fixed, handcrafted meta-agent and enabling self-referential improvements.

**Self-Improving AI.** Early on, various researchers outlined theoretical and conceptual approaches to self-improvement (Good, 1966; Schmidhuber, 1987; 2007). Some practical approaches to automated self-improvement include systems defined by neural network weight parameterizations (Schmidhuber, 1993; Hall, 2007; Hobbhahn, 2025; Kirsch & Schmidhuber, 2022; Irie et al., 2022; 2025; Lu et al., 2023; Havrilla et al., 2024b). Metz et al. (2021) developed a gradient-based optimizer that is self-referentially meta-trained using a variant of population-based training (Jaderberg et al., 2017). Lange et al. (2023) extended this approach to gradient-free learning. Silver et al. (2017) used self-play to continuously evolve agents, achieving superhuman performance in challenging domains such as chess and Go. More closely related to the DGM are recent approaches that leverage FM-based agents for self-improvement (Yin et al., 2024; Robeyns et al., 2025; Hu et al., 2024; Zelikman et al., 2024b; Huang et al., 2022; Singh et al., 2023). Zelikman et al. (2024b) use a meta-agent

to generate downstream agents, updating the meta-agent based on the meta-utility derived from the generated solutions. Yin et al. (2024) use a single system to both solve downstream tasks and recursively modify itself. However, the downstream tasks or the meta-utility do not always align with the capabilities required for self-improvement. In the DGM, improvement in downstream tasks directly reflects an increase in self-improvement ability, enabling the potential for self-accelerating progress. Most similar is concurrent work by Robeyns et al. (2025), which also has a single agent recursively solving coding problems and modifying its own codebase. The main difference from Robeyns et al. (2025) (and also Zelikman et al. (2024b); Yin et al. (2024)) is that the DGM has an open-ended exploration loop, encouraging self-modifications beyond immediate performance gains and thus avoiding stagnation in suboptimal states. Appendix B also discusses additional related work on program synthesis and Darwinian evolution.

## 3  DARWIN GÖDEL MACHINE

A Gödel Machine is a theoretical idea of an AI that searches for ways that *provably* improve itself (Schmidhuber, 2007). In this paper, we propose Darwin Gödel Machine (DGM), an attempt to realize the long-held dream of creating a Gödel Machine. The DGM relaxes the Gödel Machine's impractical requirement of theoretically *proving* that a change will improve the system, instead requiring *empirical evidence* from experiments to demonstrate that a proposed new version enhances performance. Additionally, since the DGM relies on empirical evidence of improvement, it may get stuck in a local optimum within the vast search space of possible systems (i.e., all computable algorithms). To address this, the DGM maintains an archive of discovered solutions during the search, facilitating open-ended exploration rather than relying on evolving a single solution. Since the principles echo Darwinian evolution (Darwin, 2023) (Appendix B), where new innovations emerge by selecting an entity from an archive of previously discovered solutions, modifying it, and keeping it if it is interestingly new (Zhang et al., 2024b; Faldor et al., 2025; Stanley & Lehman, 2015), we call our algorithm a Darwin Gödel Machine (Figure 1).

**Self-referential Self-improvement of Coding Agents.** The DGM is initialized with only one coding agent, and its progression is evaluated on coding benchmarks. A coding agent is defined as a single system, implemented with a code repository and powered by frozen pretrained foundation models (FMs), capable of reading, writing, and executing code. Code, when expressed in a general-purpose Turing-complete language (e.g., Python), is a powerful medium for building and improving intelligent systems because it can represent any computable process. Recent works (Hu et al., 2025; Zhang et al., 2024c) demonstrate that such agents can be improved through meta-learning of their designs (e.g., prompts, workflows, and tools), which are implemented in their code repository. Therefore, we define self-improvement as a coding task that involves modifying the design of an agent's own components (i.e., its own code, which does not include the open-ended exploration process described in the next paragraph). The key motivation is that the empirical evidence must reflect the system's ability to both self-improve and solve downstream tasks. By configuring the DGM as a coding agent and testing its coding capabilities, the observed improvements demonstrate not only enhanced performance in downstream tasks but also the capacity for further self-improvement, as self-improvement is fundamentally a coding task that modifies the coding agent's own code repository.

**Population-based Open-ended Exploration.** Starting from a single initial coding agent, the DGM builds an archive of all discovered agents. In each iteration, the DGM selects parent agents to self-modify and branch off to produce new agents. Parent selection is roughly proportional to each agent's performance score and inversely proportional to the number of its children with codebase-editing functionality (Appendix C.2). This favors high-performing agents that have been underexplored (i.e., have fewer existing children), thereby promoting both exploitation of strong performers and exploration of promising but less-sampled lineages. All agents retain a non-zero selection probability, ensuring that any path to improvement remains feasible given sufficient compute. Each selected parent analyzes its own benchmark evaluation logs, proposes the next feature to implement, and receives this proposal as a problem statement to execute (Appendix C.3). The parent then implements the suggested feature into its own codebase, generating a new coding agent. Each newly generated agent is quantitatively evaluated on a chosen coding benchmark to estimate its coding abilities. Only agents that compile successfully and retain the ability to edit a given codebase are added to the DGM archive, as only they can continue self-modification. All others are discarded. The cycle of parent selection, self-modification, and evaluation continues, progressively growing the archive of

solutions. Importantly, we note that archived solutions can serve as stepping stones that result in improvements much later than their original discovery, making our approach substantially different from hill-climbing agentic design approaches (Robeyns et al., 2025). Currently, the open-ended exploration process (i.e., archive maintenance, parent selection) is fixed and not modifiable by the DGM, which we leave as an avenue for future work. Appendix C.4 shows the pseudocode for the DGM algorithm.

## 4 EXPERIMENTS

Section 4.1 describes the experimental setup, including the initial coding agent that initializes the DGM. We conduct experiments on two coding benchmarks: SWE-bench (Jimenez et al., 2024) and Polyglot (Paul Gauthier, 2024) (Section 4.2). For each benchmark, we compare the DGM against two baselines: DGM without self-improving agents and DGM without open-ended exploration (Section 4.3). Across all experiments, we find that the DGM outperforms both baselines, showing that the self-improvement mechanism and open-ended exploration are essential for sustained performance gains (Section 4.4). Furthermore, we show that the features discovered by the DGM transfer across models, benchmarks, and tasks (Section 4.4).

### 4.1 EXPERIMENT SETUP

The DGM is initialized with a single coding agent. This base agent is built around a frozen FM and augmented with tool use capabilities (Schick et al., 2023; Anthropic, 2024b). The FM autonomously determines its action at each step, using whichever tool it deems appropriate. The base agent has access to two tools: a Bash tool for executing bash commands, and an edit tool for viewing and editing directories and entire files (Appendix C.1). The agent receives a single input prompt that specifies the location of the target repository, the problem statement, and the repository's testing framework (Appendix C.1). The base agent is intentionally lightweight to provide a simple starting point from which self-improvement can be clearly demonstrated. We run the DGM for 80 iterations (generating one new agent per iteration), with two iterations running in parallel for SWE-bench and four for Polyglot (Appendix C.2). During self-modifications, coding agents are powered by Claude 3.5 Sonnet (New) (Anthropic, 2024a) in both SWE-bench and Polyglot experiments. During benchmark evaluation, coding agents are powered by Claude 3.5 Sonnet (New) for SWE-bench and o3-mini (OpenAI, 2025) for Polyglot (Appendix D.1).

### 4.2 BENCHMARKS

We evaluate the DGM on two popular benchmarks that assess different aspects of coding tasks to validate the algorithm's effectiveness across various use cases. Both SWE-bench and Polyglot are benchmarks that evaluate how well AI agents solve coding tasks automatically. Given a code repository and a task instruction, the agent is expected to make changes to the repository in order to fulfill the task. Both SWE-bench and Polyglot are widely used benchmarks (Zhang et al., 2024d;a; Xia et al., 2024; Cao et al., 2024; Google DeepMind, 2025; Gauthier, 2024) that require the AI agent to navigate a code repository, understand the interplay between functions in different files, and spot small errors in convoluted code. SWE-bench only has Python tasks, while Polyglot has tasks in multiple programming languages. Another difference is that each SWE-bench task may require edits to multiple files, whereas each Polyglot task primarily involves implementing a solution from scratch in a single file (although the agent still needs to examine other files to understand what changes are necessary), resulting in fewer file edits overall.

**SWE-bench.** To avoid wasting compute on unsolvable tasks, we use SWE-bench Verified (OpenAI, 2024), a human-filtered subset of SWE-bench (Jimenez et al., 2024) where all tasks are solvable. Throughout this paper, the term SWE-bench refers by default to to the SWE-bench Verified subset.

**Polyglot.** Polyglot includes tasks in multiple programming languages (C++, Rust, Python, etc.) (Paul Gauthier, 2024). Compared to SWE-bench, one of the most widely used coding benchmarks and likely included in the training sets of FMs, Polyglot is more niche and less likely to be included in FMs' post-training data. Additionally, Polyglot is primarily used by its developer to evaluate Aider (Gauthier, 2024). This provides an opportunity to compare automatically designed agents with a representative coding agent in the community, which has been continuously developed and tested

against the Polyglot benchmark over a year by human developers. Finally, Polyglot enables testing of the DGM's ability to generalize across multiple programming languages beyond just Python. We follow the Polyglot leaderboard setup (Paul Gauthier, 2024), with one key difference: the leaderboard reports pass@2, where the agent can view feedback from ground-truth tests once, whereas we report pass@1, where the agent never sees ground-truth test results.

Since it is expensive to evaluate on the full benchmarks (Appendix E.1), we use a staged evaluation strategy to estimate the coding agent's performance. In general, we evaluate the coding agent on more tasks when we are more confident that it is a relatively strong performer. We first evaluate each coding agent on a set of 10 tasks to verify basic codebase-editing functionality. Only agents that retain the ability to edit code can solve downstream coding tasks or perform further self-modifications. Agents that pass this initial stage are then evaluated on an expanded set of 50 tasks. For SWE-bench, tasks are selected based on SWE-bench-verified-mini (Hobbhahn, 2025), which is designed to be a representative sample of the full benchmark (Appendix E.2). For Polyglot, tasks are chosen as a random subset of the full benchmark (Appendix E.3). Because the LLMs we use are inherently stochastic, performance can be noisy. We use smaller subset evaluations to estimate general effectiveness and identify top candidates. For SWE-bench, if a coding agent achieves over 40% success (a heuristic chosen based on the noise observed in preliminary runs) on the 60-task subset and ranks among the top two performers in the archive, it is then evaluated on 200 tasks (140 plus the previous 60) to more accurately assess its true coding capability (Appendix E.2). For Polyglot, if a coding agent achieves over 40% success on the 10-task subset, it is evaluated on an expanded 50-task subset (Appendix E.3).

## 4.3 BASELINES

To isolate and evaluate the contribution of the DGM's core components, we compare the DGM against two baselines: DGM without self-improving agents (**DGM w/o self-improve**) and DGM without open-ended exploration (**DGM w/o open-ended exploration**). DGM w/o self-improve replicates the approach of ADAS (Hu et al., 2025) in this setting, wherein the meta agent responsible for modifying the coding agents remains fixed as the base agent throughout the experiment (Algorithm 2). This baseline allows us to assess the role of iterative self-improvement in accelerating the evolution of better coding agents. DGM w/o open-ended exploration eliminates the use of an archive and always self-modifies the latest stored version of itself (Algorithm 3). If a coding agent self-modifies to the point where it loses the basic functionality required to edit a codebase, it can no longer modify itself or solve any coding task. Therefore, DGM w/o open-ended exploration retains the latest version of itself that still maintains the basic functionality for codebase editing. This baseline allows us to evaluate the impact of having an archive and the well-documented beneficial principles of open-ended exploration (Clune, 2019; Stanley & Lehman, 2015; Zhang et al., 2024b; Fernando et al., 2024; Lee et al., 2020; Samvelyan et al., 2024; Colas et al., 2022b) in guiding the agent's evolution.

In addition to the learned baselines, we compare the DGM against handcrafted, open-source solutions. For SWE-bench, we take the state-of-the-art (SoTA) open-source solution that has been checked (i.e., the SWE-bench team was able to reproduce the results) (Appendix E.4). For Polyglot, we take the representative agent (Aider) (Gauthier, 2024), which is open-sourced and designed to support multiple programming languages and large codebase editing (Appendix E.5). For a fair comparison, we measure the percentage of solved tasks on the same benchmark subsets used to evaluate the DGM (Appendix E.2, Appendix E.3). These values are shown as dotted horizontal lines in Figure 2.

## 4.4 RESULTS

After 80 iterations of the DGM, the coding agent's performance increases from 20.0% to 50.0% on SWE-bench, and from 14.0% to 38.0% on Polyglot (Figure 2). Since the DGM is evaluated on only 50 tasks in the Polyglot experiment setup (Section 4.2), we additionally evaluate both the base agent and the best DGM-discovered agent on the full Polyglot benchmark to more accurately estimate the improvement. On the full Polyglot benchmark, the DGM improves the coding agent from 14.2% to 30.7%. This shows that the DGM can automatically self-improve to create a better coding agent. Moreover, the performance of the best DGM-discovered agent is comparable to that of the checked, open-source, human-designed SoTA on SWE-bench (Figure 2). On Polyglot, although the DGM starts with a base agent whose performance is lower than that of Aider, it discovers an agent that far

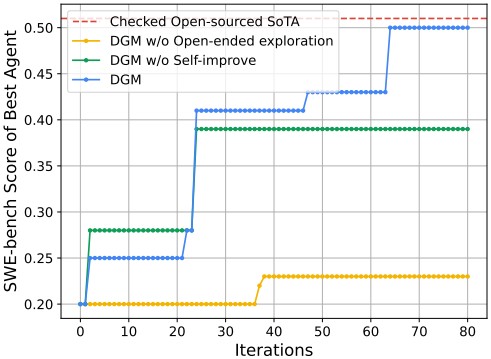 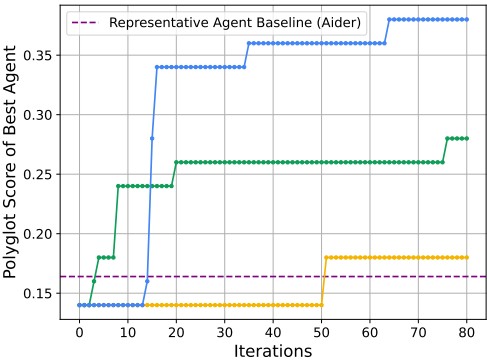

Figure 2: **Self-improvement and open-ended exploration enable the DGM to continue making progress and improve its performance.** The DGM automatically discovers increasingly better coding agents and performs better on both (Left) SWE-bench and (Right) Polyglot. It outperforms baselines that lack either self-improvement or open-ended exploration, showing that both components are essential for continual self-improvement. These scores are obtained from evaluating on the benchmark subsets detailed in Section 4.2.

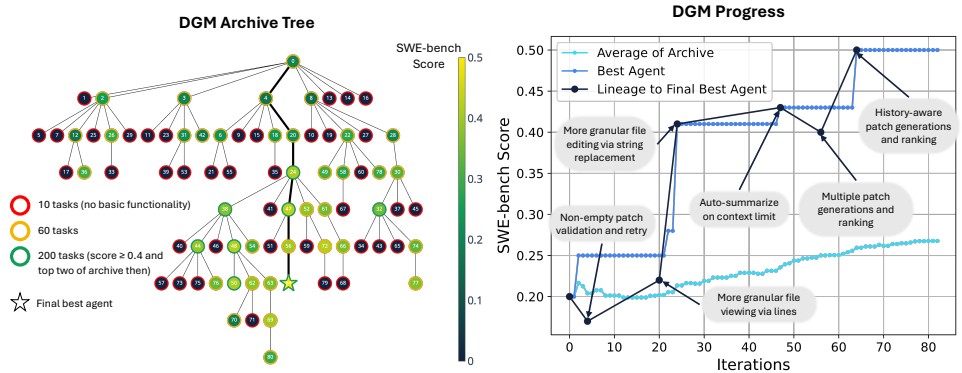

Figure 3: **The DGM automatically self-improves to become a better coding agent.** (Left) Archive of coding agents generated during the DGM run on SWE-bench. Each node represents a coding agent, with node 0 corresponding to the base agent. Node color indicates performance on SWE-bench (percentage of solved tasks), while border color reflects the number of tasks for which the agent was evaluated. Edges show which agents self-modified to produce the offsprings. Many paths to innovation traverse lower-performing nodes, and key innovations (like node 24) lead to an explosion of innovations built on top of them. Both properties underscore the benefits of open-ended search. (Right) Progress plot of the DGM on SWE-bench. The light blue line shows the average score of all agents possessing basic codebase-editing functionality. The blue line tracks the best score achieved by any agent in the archive at each iteration. The dark line shows the lineage of the final best-discovered agent and its precursor nodes, which includes two performance dips. This illustrates the benefits of open-ended search, which explores a diverse set of interesting stepping stones instead of focusing only on branching off the best solution found so far.

surpasses Aider (Figure 2). The DGM-discovered agents are comparable to or outperform handcrafted agents on both benchmarks. While the SoTA SWE-bench agent and Aider were painstakingly shaped by human efforts, the DGM hints at a future in which such ingenuity is automated, evolving through self-referential cycles of continuous self-improvements.

The DGM automatically improves both the tools and the workflow of how FMs are utilized (Figure 3). For example, the DGM enhanced the edit tool to allow more granular file viewing (by lines) and more precise file editing (by string replacement), instead of always viewing or replacing the entire

file. Workflow improvements include making multiple attempts to solve a task and using another FM to evaluate and select the best solution. Other workflow improvements include considering previous attempts when generating subsequent ones. Appendix F.1 and Appendix F.2 show all modifications leading up to the final best-discovered agents on SWE-bench and Polyglot respectively.

Because open-ended exploration allows branching from any agent in the archive with non-zero probability, the DGM can get out of deceptive dips or peaks in performance. For example, at iterations 4 and 56 of the experiment on SWE-bench, although the agent's score temporarily fell below that of its parent, the DGM was still able to explore innovations along that path and create a new agent that outperformed all of its predecessors (Figure 3). Furthermore, open-ended exploration allows different implementations of the same target functionality to be attempted. For example, while the goal is to provide finer-grained editing tools, the specific implementation of this feature can vary greatly and hence lead to very different performance (Appendix G). The DGM can explore multiple implementations to find the most suitable one and avoid getting trapped in a suboptimal one.

The DGM outperforms the baselines of DGM w/o self-improve and DGM w/o open-ended exploration on both benchmarks (Figure 2). Without updating the meta agent that modifies coding agents, DGM w/o self-improve improves the agents in early iterations, but its gains taper off quickly (Appendix A.1). In DGM w/o open-ended exploration, only the most recent agent is retained, so a poorly performing self-modification makes subsequent improvements harder to achieve (Appendix A.1).

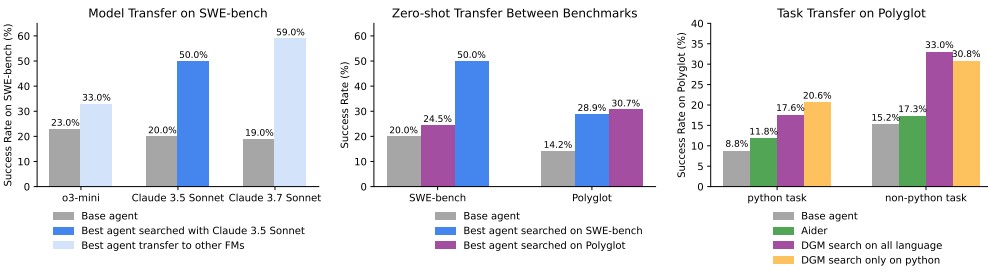

Figure 4: **Transfer between Models, Benchmarks, and Tasks.** The superior performance of DGM-discovered agents can be transferred across (Left) different models, (Middle) benchmarks, and (Right) different programming language tasks in Polyglot, such as from Python tasks to C++ tasks.

To evaluate the generality of the improvements from the DGM, we tested the base agent (Section 4.1) and the best agent discovered during the DGM run (Figure 2) with different FMs than those used during optimization. For SWE-bench, where the DGM was run using Claude 3.5 Sonnet (New), we replaced the FM with Claude 3.7 Sonnet (Anthropic, 2025) or o3-mini, and evaluated on 200 tasks (Figure 4, Left). With o3-mini, the base agent achieved 23.0% and the DGM-discovered agent 33.0%. With Claude 3.7 Sonnet, the base agent achieved 19.0% and the DGM-discovered agent 59.5%. These results suggest that the DGM yields improvements that generalize across FMs, rather than being tightly coupled to the specific FM used during its run (Figure 4). Model transfer results on Polyglot are presented in Appendix A.2.

Furthermore, we investigate the transferability of the DGM-discovered agent across different benchmarks and programming languages. First, we evaluate the best DGM-discovered agent from one benchmark (e.g., SWE-bench) on a completely held-out benchmark (e.g., Polyglot), and vice versa (Figure 4, Middle). The best agent evolved on SWE-bench achieves 28.9% on Polyglot, compared to the initial agent's baseline of 14.2%. Conversely, the best agent evolved on Polyglot achieves 24.5% on SWE-bench, outperforming the original baseline of 20.0%. Since each agent was optimized without ever accessing the alternate benchmark, these evaluations represent truly held-out tests. The consistent performance gains across benchmarks support our claim that DGM's improvements reflect general skill acquisition rather than overfitting or exploitation of benchmark-specific artifacts. Second, we experiment with a version of the DGM trained exclusively on Python tasks from Polyglot and then transfer the discovered agent to tasks in other languages. Focusing primarily on Python tasks slightly improves performance on Python tasks but reduces performance on non-Python tasks compared to the DGM trained on all languages (Figure 4, Right). However, after being transferred from Python to other unseen languages during the search, the agent still achieves performance comparable to that of the DGM trained on all languages and substantially outperforms both the base agent and Aider.

These results demonstrate the robustness of the discovered improvements, showing that they do not overfit to a specific programming language. We also present additional results in Appendix A.

## 5 SAFETY DISCUSSION

Systems capable of self-improvement, such as the DGM, represent a step toward more autonomous AI development, aligning with long-standing goals in the field of making capable AI that can benefit humanity (Schmidhuber, 1987; Clune, 2019; Markoff, 2016; Lehman, 2023). However, this capability introduces unique safety considerations stemming from the system's ability to autonomously modify its own code. Modifications optimized solely for benchmark performance might inadvertently introduce vulnerabilities or behaviors misaligned with human intentions, even if they improve the target metric (Bostrom, 2020). In particular, if evaluation benchmarks do not fully capture all desired agent properties (e.g., safety and robustness), the self-improvement loop could amplify misalignment over successive generations. Iterative self-modification could also lead to increasingly complex and uninterpretable internal logic, hindering human understanding, oversight, and control (Sheth et al., 2025; Anwar et al., 2024; Greenblatt et al., 2024; Ganguli et al., 2022).

Recognizing these challenges, the current implementation and experimental setup of the DGM incorporates several safeguards. All agent execution and self-modification processes are conducted within isolated sandboxed environments, limiting their ability to affect the host system, and thereby mitigating the risk of unintended actions. Each execution within the sandbox is subjected to a strict time limit, reducing the risk of resource exhaustion or unbounded behavior. The self-improvement process is currently confined to the well-defined domain of enhancing performance on specific coding benchmarks by modifying the agent's own Python codebase, thus limiting the scope of potential modifications. Additionally, we actively monitor agent performance and code changes, with the DGM archive providing a traceable lineage of modifications for review. At this stage, we have found no evidence of harmful or malicious behavior in the generated agents, and the self-modifications have been primarily focused on improving coding capabilities.

Conversely, a significant potential benefit of the self-improvement paradigm is that it could, in principle, be directed toward enhancing safety and interpretability themselves. We conduct a preliminary investigation into how the DGM can be deployed in AI safety settings to develop countermeasures for FM hallucination (Appendix H). Just as the DGM learns to improve its coding capabilities, it could potentially discover and integrate better internal safeguards or modify itself for greater transparency (e.g., incorporating principles akin to Constitutional AI (Bai et al., 2022)), if such properties were included in its evaluation criteria (Rosser & Foerster, 2025). This suggests a promising, albeit challenging, pathway in which self-improvement becomes a tool for building more trustworthy AI systems. Additional research could also explore weaving Constitutional AI in from the start, though the challenge would be incentivizing the system to retain these directives (an option worth exploring is to create an unmodifiable part of the system to be able to evaluate at halt the rest).

The DGM demonstrates the potential of self-improving AI while still operating within safe research boundaries due to the current limitations of frontier FMs and effective mitigations like sandboxing. Appendix I presents additional discussion on broader safety uncertainties. We include this safety discussion proactively to raise awareness about the emerging prospect of self-improving AI systems and their associated safety implications, particularly as these systems inevitably become more capable (Yudkowsky et al., 2008; Bostrom, 2002; Ecoffet et al., 2020; Bengio et al., 2024; Clune, 2019). Accordingly, we advocate for continued investigation into the safe and beneficial evolution of AI-Generating Algorithms (Clune, 2019) and self-improving systems.

## 6 CONCLUSION AND LIMITATIONS

We introduce the Darwin Gödel Machine (DGM), the first self-improving system powered by FMs with open-ended exploration, where progress on its evaluation benchmarks can directly translate into better self-improvement capabilities. We demonstrate the automatic discovery of better tools and FM systems, resulting in better performance on two benchmarks: SWE-bench and Polyglot. Through self-improvement and open-ended exploration, the DGM shows a continuous increase in performance, bringing us one step closer to self-accelerating, self-improving AI systems.

We demonstrate that the DGM can autonomously achieve performance on par with openly available solutions. However, it still falls short of closed-source SoTA SWE-bench solutions. An open question is whether running the DGM for longer would continue to yield performance gains and eventually surpass closed-source solutions. These closed-source solutions often rely on elaborately handcrafted techniques developed by teams of highly skilled experts. Since FMs have yet to match the capabilities of such experts (e.g., in reasoning), the DGM currently requires extensive compute to discover improvements. A single run of the DGM on SWE-bench, as presented in Section 4, takes about 2 weeks and incurs significant API costs (Appendix E.1). We hypothesize that further progress will require more efficient use of computational resources and the development of better reasoning skills.

Since this version of the DGM is mainly powered by FMs, it is inherently limited by the capabilities of the underlying FM. Hence, an exciting future direction is to extend self-modification beyond just prompts or FM workflows, to include more computationally intensive methods, such as rewriting its own training script to update the FM itself. While this version of the DGM focuses on coding, AI systems are increasingly applied across a wide range of domains (e.g., computer vision, creative writing). Another promising extension is to develop self-improving AI systems capable of enhancing themselves beyond just the coding domain. A key assumption in this work is that coding benchmarks are a good reflection of the agent's ability to self-improve, since the self-modification task requires the agent to modify its own codebase. However, one could envision an alternative approach that co-evolves the target task distribution (Faldor et al., 2025; Wang et al., 2023c), thereby removing the constraint of self-improvement being tied to a single objective, as in true open-ended processes. Appendix J presents additional potential directions for future work. As we continue to explore this powerful technology, we must also keep safety front and center, as discussed in Section 5.

In conclusion, the DGM represents a significant step toward the automation of AI development through self-improving systems capable of editing their own codebase. While current limitations in compute and reasoning constrain its full potential, continued advances in FMs and infrastructure may unlock more powerful and general-purpose self-improvements. Provided that the safety concerns are carefully navigated (Section 5), the future of self-improving AI systems and AI-Generating Algorithms (Clune, 2019) holds immense promise to open-endedly evolve AI, continually rewriting or retraining itself in pursuit of greater capabilities aligned with human values.

ETHICS STATEMENT

We affirm compliance with the ICLR Code of Ethics. This work studies self-improving AI systems in the limited context of code-editing agents evaluated on standard programming benchmarks. No human subjects were involved and no personally identifiable information (PII) was collected or processed; IRB approval was therefore not required.

**Safety and misuse.** Self-modifying systems can pose safety risks if allowed to act without constraints or if optimizations inadvertently introduce unsafe behaviors. To mitigate this, all agents in our experiments ran inside isolated sandboxes with strict resource and time limits; agents had limited network access and no ability to modify the host environment. The self-improvement scope was restricted to the agent's own Python codebase and evaluation harnesses. We maintained a complete, auditable lineage (archive) of code changes and evaluations, enabling rollback and post-hoc analysis. We did not deploy discovered agents in real development environments. Our release plan (code, prompts, and evaluation artifacts) will exclude any components that grant elevated system access and will include default sandboxing, guardrails, and clear documentation of intended use.

**Dual-use, downstream impact, and limitations.** Stronger autonomous coding agents could be dual-use (e.g., aiding software maintenance, but also potentially facilitating creation of harmful code if misapplied). We believe the research benefits (e.g., advancing methods for controlled, auditable self-improvement and demonstrating practical safeguards) outweigh the risks. Nevertheless, we explicitly discourage security-sensitive or unsandboxed deployment and provide concrete safety recommendations (Section 5). Our empirical focus on benchmark optimization may not capture all desirable properties (robustness, interpretability, or broader social values). We therefore treat benchmark gains as necessary but insufficient indicators of general AI development, and discuss avenues to integrate other objectives (e.g., safety, reasoning) into the optimization loop.

**Data governance, IP, and licensing.** We evaluate on SWE-bench Verified and Polyglot, which are composed of open-source repositories and tasks. We complied with dataset licenses and usage terms to the best of our knowledge. We did not introduce or distribute proprietary code. Foundation models (FMs) were accessed via provider APIs under their terms of service; we did not submit sensitive data, nor attempt to circumvent usage policies. Logs released with this work will be scrubbed of API keys and any incidental sensitive strings.

**Bias, fairness, and equity.** Although our domain is software code rather than human-centered text, FM behavior can still reflect biases (e.g., language or ecosystem preferences) and may unevenly benefit communities whose tooling is better represented in training data. We partially address this by evaluating across multiple languages (Polyglot) and reporting cross-benchmark transfer. Future work should add diagnostics for biased failure modes and include broader, community-driven task sets.

**Conflicts of interest and funding.** No author has a financial interest in products whose performance is evaluated here. Sponsors and employers did not influence experimental design, analysis, or the decision to publish, beyond providing salary or standard research support. Any external compute or API credits are acknowledged in the appendix.

REPRODUCIBILITY STATEMENT

We will open-source all code and full agent logs, including the complete archive lineage of self-modifications (diffs, prompts, and configs) as well as the evaluation harness. To support exact replication, we reference the following: algorithmic details and pseudocode (Section 3, Appendix C.4); parent selection and open-ended exploration settings (Appendix C.2); foundation model choices and hyperparameters (Appendix D.1); benchmark task subsets for SWE-bench and Polyglot (Appendix E.2, Appendix E.3); staged evaluation protocols and scripts (Section 4.2); implementations and diffs for the best discovered agents (Appendix F.1, Appendix F.2); and compute and cost estimates (Appendix E.1). The released code repository will include environment specifications and scripts to reproduce all results, figures, and tables.

ACKNOWLEDGMENTS

This research was supported by the Vector Institute, the Canada CIFAR AI Chairs program, a grant from Schmidt Futures, an NSERC Discovery Grant, and a generous donation from Rafael Cosman. Resources used in preparing this research were provided, in part, by the Province of Ontario, the Government of Canada through CIFAR, and companies sponsoring the Vector Institute (https://vectorinstitute.ai/partnerships/current-partners/). Any opinions, findings, and conclusions or recommendations expressed in this material are those of the authors and do not necessarily reflect the views of the sponsors. We also thank Aaron Dharna, Ben Norman, Cédric Colas, Sam Devlin, and Shyam Sudhakaran for insightful discussions and feedback.

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

# APPENDIX

## TABLE OF CONTENTS

# A ADDITIONAL RESULTS

## A.1 BASELINES ON SWE-BENCH

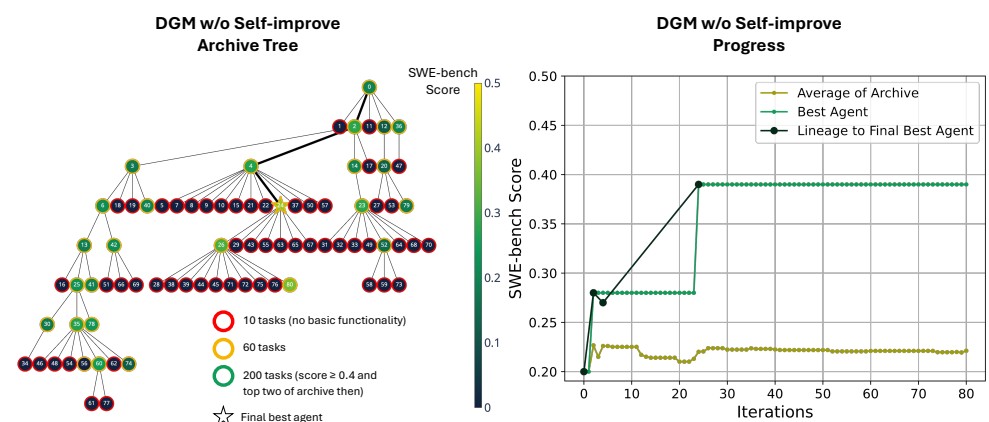

Figure 5: **DGM without self-improving agents.** Keeping the meta-agent that is modifying and producing the next coding agents the same, DGM w/o self-improve is unable to continuously improve over time. (Left) Archive of coding agents generated during the DGM w/o self-improve run on SWE-bench. Each node represents a coding agent, with node 0 corresponding to the base agent. Node color indicates performance on SWE-bench (percentage of solved tasks), while border color reflects the number of tasks for which the agent was evaluated. Edges show which agents self-modified to produce the offsprings. (Right) Progress plot of the DGM w/o self-improve on SWE-bench. The light green line shows the average score of all agents possessing basic codebase-editing functionality. The green line tracks the best score achieved by any agent in the archive at each iteration. The dark line shows the lineage of the final best-discovered agent and its precursor nodes.

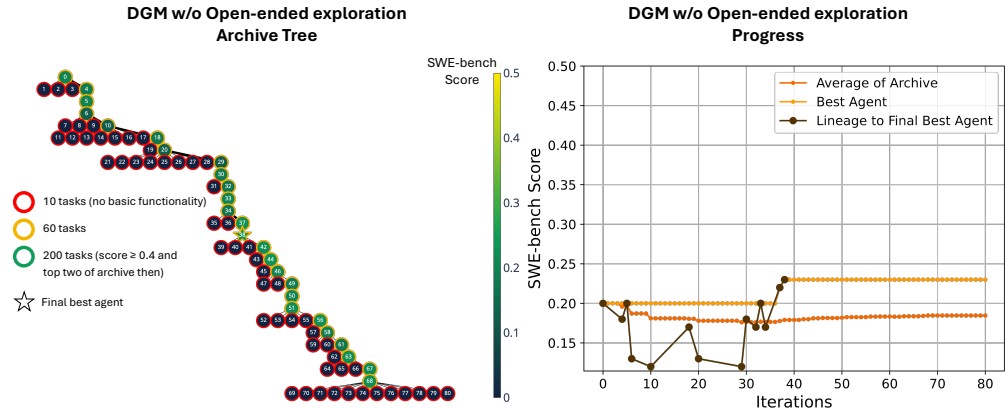

Figure 6: **DGM without open-ended exploration.** Removing the archive, DGM w/o open-ended exploration always uses the most recent agent to self-modify and makes very little progress on SWE-bench. (Left) Archive of coding agents generated during the DGM w/o open-ended exploration run on SWE-bench. Each node represents a coding agent, with node 0 corresponding to the base agent. Node color indicates performance on SWE-bench (percentage of solved tasks), while border color reflects the number of tasks for which the agent was evaluated. Edges show which agents self-modified to produce the offsprings. (Right) Progress plot of the DGM w/o open-ended on SWE-bench. The orange line shows the average score of all agents possessing basic codebase-editing functionality. The light orange line tracks the best score achieved by any agent in the archive at each iteration. The dark line shows the lineage of the final best-discovered agent and its precursor nodes.

## A.2 Generality across models on Polyglot

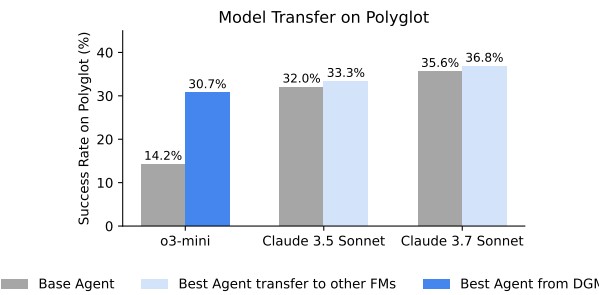

Figure 7: **Transfer between Models on Polyglot**

In addition to testing the transfer models on SWE-bench (see Section 4.4, Figure 2), we also present the transfer results on Polyglot in this section. On Polyglot (Figure 7), where the DGM was run with o3-mini, we replaced the FM with Claude 3.5 Sonnet (New) or Claude 3.7 Sonnet, and evaluated on the full benchmark (Figure 4, Middle). With Claude 3.5 Sonnet (New), the initial agent achieved 32.0% and the DGM-discovered agent 33.3%. With Claude 3.7 Sonnet, the initial agent achieved 35.6% and the DGM-discovered agent 36.8%. These results suggest that the DGM yields improvements that generalize across FMs, rather than being tightly coupled to the specific FM used during its run (Figure 4).

## A.3 Ablation of Parent Selection

To further study the impact of the parent selection mechanism in DGM, we introduce DGM Greedy. DGM Greedy always selects the best-performing node as the parent to branch from, rather than giving every node a non-zero probability of being branched off (roughly proportional to their performance score and number of children) as in this implementation of the DGM (Appendix C.2). This ablation replicates the approach of Robeyns et al. (2025) in this setting. As shown in Table 1, DGM Greedy achieves 39.7% and 30.0% on SWE-bench and Polyglot, respectively, compared to 50.0% and 38.0% by this implementation of DGM. These results demonstrate that allowing all solutions in the archive to serve as potential stepping stones can lead to greater improvements over time, underscoring the importance of open-ended exploration.

Table 1: Comparison of DGM, its ablations, and baselines on SWE-bench and Polyglot benchmarks.

| Method | SWE-bench | Polyglot |
|---|---|---|
| DGM | 50.0% | 38.0% |
| DGM w/o Open-ended exploration | 23.0% | 14.0% |
| DGM w/o Self-improve | 39.0% | 28.0% |
| DGM Greedy | 39.7% | 30.0% |

## A.4 Additional Statistics of DGM runs

**Percentage of Generated Agents with Basic Code-Editing Functionality.** To gain deeper insights into the DGM process, we analyze the percentage of generated agents that possess basic code-editing functionality on the SWE-bench benchmark. As shown in Table 2, DGM exhibits the highest percentage of producing agents with basic codebase-editing functionality. These results highlight the effectiveness of both open-ended exploration and self-improvement components in the DGM, where open-ended exploration enables the search to escape local optima, while self-improvement enhances the ability to generate better agents.

Table 2: Percentage of generated agents with basic code-editing functionality on SWE-bench.

| Method | Percentage with Basic Code-Editing Functionality |
| --- | --- |
| DGM | 51.3% |
| DGM w/o Open-ended exploration | 32.5% |
| DGM w/o Self-improve | 32.5% |

**Stability of DGM Runs.** To evaluate the stability of DGM, we run the DGM algorithm three times on the Polyglot benchmark and analyze the variance in performance. The DGM achieved a mean accuracy of 40.7% with a standard deviation of 2.3%, indicating that the DGM can achieve consistent and reproducible results across runs.

# B  ADDITIONAL RELATED WORK

**Open-Endedness (part 2).** Early approaches to open-endedness explored different mechanisms to balance learnability and interestingness. Quality-diversity algorithms sought to illuminate vast solution spaces with diverse, high-performing behaviors (Pugh et al., 2016; Chatzilygeroudis et al., 2021; Mouret & Clune, 2015; Nguyen et al., 2015). Other methods emphasized goal-directed exploration (Ecoffet et al., 2019; 2021; Schaul et al., 2015; Andrychowicz et al., 2017; Eysenbach et al., 2018), intrinsic motivation (Lehman & Stanley, 2011; Oudeyer et al., 2007; Li et al., 2014; Pathak et al., 2017), or learning progress frameworks (Kanitscheider et al., 2021; Gaven et al., 2025; Baranes & Oudeyer, 2013; Colas et al., 2019; 2022b; Jiang et al., 2021; Dennis et al., 2020; Schmidhuber, 2008; 2013; Kompella et al., 2017). More recently, large-scale foundation models (FMs) (Brown et al., 2020; Radford et al., 2019) have emerged as powerful proxies for human notions of interestingness (Zhang et al., 2024b; Faldor et al., 2025; Sancaktar et al., 2025) and effective mutation operators to propose novel solutions in code (Romera-Paredes et al., 2024; Novikov et al., 2025; Lehman et al., 2023; Faldor et al., 2025; Hu et al., 2025). FMs can guide autotelic agents (Colas et al., 2022b; 2023; 2022a), model human preferences for quality and diversity (Bradley et al., 2024; Ding et al., 2024; Wang et al., 2023b; Klissarov et al., 2023; 2024; Samvelyan et al., 2024; Lim et al., 2024; Havrilla et al., 2024a), design reward functions (Wiering & Schmidhuber, 1997; Wang et al., 2023a; Ma et al., 2023; Faldor et al., 2025), create simulated environments (Sudhakaran et al., 2023; Nasir & Togelius, 2023; Aki et al., 2024; Nasir et al., 2024; Bruce et al., 2024; Parker-Holder et al., 2024; Schmidhuber, 2013), drive ever-evolving multi-agent dynamics (Dharna et al., 2024; Zhou et al., 2025), search diverse ambulating robot morphologies (Lehman et al., 2023), and search expansive solution spaces for benchmark or objective optimization (Lange et al., 2024; Zhang et al., 2024b; Faldor et al., 2025; Hu et al., 2025; Lu et al., 2024b; Romera-Paredes et al., 2024; Fernando et al., 2024; Lu et al., 2024a; Khan et al., 2024; Lu et al., 2025; Liu et al., 2024; Novikov et al., 2025).

**Program Synthesis.** Program synthesis (Alur et al., 2018; Polozov & Gulwani, 2015; Buchi & Landweber, 1990; Gulwani, 2011; Ellis et al., 2021) seeks to generate code meeting external specifications such as input-output examples or logical formulas. Hybrid approaches combine symbolic methods with neural or FM guidance: for instance, Li et al. (2024) uses LLM suggestions to steer symbolic search in SyGuS settings, improving over pure enumeration. Barke et al. (2024) blends LLM completions with a learned surrogate model to guide synthesis in DSLs. Shi et al. (2023) uses neural policies to build higher-order and lambda abstractions during search, outperforming both LLM-only and symbolic baselines on list manipulation tasks. The DGM differs in focusing not just on producing programs for external tasks, but also on agent self-modification, rewriting its own implementation to improve its capacity for future self-improvement.

**Inspiration from Darwinian Evolution.** This work is heavily inspired by the mechanisms of Darwinian evolution (Darwin, 2023), notably variation (mutation), selection, and the preservation of lineages (stepping stones), and brings them into the realm of self-modifying coding agents. In DGM, an archive of past agent versions is maintained, from which parent agents are sampled; then mutations (i.e., code edits) generate new offspring agents, which are empirically evaluated on coding benchmarks; successful ones are added to the archive, enabling parallel exploration of multiple evolutionary trajectories (Section 3). This mirrors how biological evolution (Edwards, 2000; Wright,

1932) preserves genetic diversity (Mayr, 1982), leverages variation (Kimura, 1979), and uses natural selection to retain beneficial mutations (Dobzhansky, 1970).

# C  ALGORITHMIC DETAILS

## C.1  INITIAL CODING AGENT

In this section, we present the details of the tools available to the initial coding agent (Section 4.1) and its task prompt.

Information of the given Bash tool:

```
def tool_info():
    return {
        "name": "bash",
        "description": """Run commands in a bash shell\n
* When invoking this tool, the contents of the "command" parameter does NOT need to be
    ↪ XML-escaped.\n
* You don't have access to the internet via this tool.\n
* You do have access to a mirror of common linux and python packages via apt and pip.\n
* State is persistent across command calls and discussions with the user.\n
* To inspect a particular line range of a file, e.g. lines 10-25, try 'sed -n 10,25p
    ↪ /path/to/the/file'.\n
* Please avoid commands that may produce a very large amount of output.\n
* Please run long lived commands in the background, e.g. 'sleep 10 &' or start a server in
    ↪ the background.""",
        "input_schema": {
            "type": "object",
            "properties": {
                "command": {
                    "type": "string",
                    "description": "The bash command to run."
                }
            },
            "required": ["command"]
        }
    }
```

Information of the given Edit tool:

```
def tool_info():
    return {
        "name": "editor",
        "description": """Custom editing tool for viewing, creating, and editing files\n
* State is persistent across command calls and discussions with the user.\n
* If `path` is a file, `view` displays the entire file with line numbers. If `path` is a
    ↪ directory, `view` lists non-hidden files and directories up to 2 levels deep.\n
* The `create` command cannot be used if the specified `path` already exists as a file.\n
* If a `command` generates a long output, it will be truncated and marked with `<response
    ↪ clipped>`.\n
* The `edit` command overwrites the entire file with the provided `file_text`.\n
* No partial/line-range edits or partial viewing are supported.""",
        "input_schema": {
            "type": "object",
            "properties": {
                "command": {
                    "type": "string",
                    "enum": ["view", "create", "edit"],
                    "description": "The command to run: `view`, `create`, or `edit`."
                },
                "path": {
                    "description": "Absolute path to file or directory, e.g. `/repo/file.py` or
                        ↪ `/repo`.",
                    "type": "string"
                },
                "file_text": {
                    "description": "Required parameter of `create` or `edit` command,
                        ↪ containing the content for the entire file.",
                    "type": "string"
                }
            },
            "required": ["command", "path"]
        }
    }
```

Task prompt:

```
I have uploaded a Python code repository in the directory {self.git_tempdir}. Help solve
    ↪ the following problem.

<problem_description>
{self.problem_statement}
</problem_description>

<test_description>
{self.test_description}
</test_description>

Your task is to make changes to the files in the {self.git_tempdir} directory to address
    ↪ the <problem_description>. I have already taken care of the required dependencies.
```

## C.2 PARENT SELECTION

At each DGM iteration, we select a subset of agents from the archive as parents to self-modify and produce new child agents (Section 3). The details of the parent selection process, inspired by Ecoffet et al. (2019), are outlined below. Future work could explore alternative search and exploration methods (Coulom, 2006; Silver et al., 2016; Herr et al., 2025).

At each iteration $t$ of the DGM run, let

$$\mathcal{A}^t = \{a_1^t, a_2^t, \ldots, a_N^t\}$$

be the archive of candidate agents. We first define the eligible set

$$\mathcal{E}^t = \big\{ a_i^t \in \mathcal{A}^t : \alpha_i < 1 \big\},$$

i.e. only those agents whose performance score is not yet perfect. We then sample $k$ parents (with replacement) from $\mathcal{E}^t$ in proportion to a combined measure of performance and number of children with codebase-editing functionality. Concretely, for each agent $a_i^t \in \mathcal{E}^t$:

$$
\begin{align}
\text{(performance)} \quad \alpha_i &= \text{performance}(a_i^t), \tag{1}\\
\text{(children count)} \quad n_i &= \text{functioning\_children\_count}(a_i^t), \tag{2}\\
\text{(sigmoid-scaled performance)} \quad s_i &= \frac{1}{1 + \exp\big(-\lambda\,(\alpha_i - \alpha_0)\big)}, \tag{3}\\
\text{(novelty bonus)} \quad h_i &= \frac{1}{1 + n_i}, \tag{4}\\
\text{(unnormalized weight)} \quad w_i &= s_i\,h_i, \tag{5}\\
\text{(selection probability)} \quad p_i &= \frac{w_i}{\sum_{j=1}^N w_j}, \tag{6}\\
\text{(sampling)} \quad \{\,\text{parents}\,\} &\sim \text{Categorical}\big(\{p_i\}_{a_i^t \in \mathcal{E}^t}\big) \quad \text{(draw } k \text{ samples)}. \tag{7}
\end{align}
$$

Here, $\lambda > 0$ controls the sharpness of the sigmoid, and $\alpha_0$ is the midpoint accuracy. We set $\lambda = 10$ and $\alpha_0 = 0.5$. $k$ determines how many self-improvements are done in parallel during each iteration of the DGM run. We set $k = 2$ in SWE-bench experiments, and $k = 4$ in Polyglot experiments.

## C.3 SELF-IMPROVE PROMPTS

In every iteration, an agent is chosen to self-modify. For this step, the current implementation of DGM leverages a separate FM to diagnose improvement opportunities, which are then passed to the coding agent for implementation. To identify the next potential improvement, the evaluation logs from the chosen agent's performance on the coding benchmark are fed into the FM. While the current implementation relies on an additional diagnostic step, there are no fundamental limitations preventing the DGM from autonomously analyzing its own performance, generating improvement

hypotheses, and implementing subsequent modifications. We chose to use OpenAI's o1 model (Team et al., 2024) for this purpose, as we found it had better reasoning capabilities than other FMs available at the time of our experiments. The prompts used for this diagnosis are provided below. From the diagnostic output, the `implementation_suggestion` and `problem_description` are extracted and passed to the coding agent as its self-modification task.

Self-improve diagnosis prompt for SWE-bench experiments:

---

**Self-Improve Prompt**

```
# Coding Agent Summary

- **Main File**: `coding_agent.py`
  - Primary Class: `AgenticSystem`
  - The `forward()` function is the central entry point.
  - Prompts are located either within the `forward()` function or in the `prompts/`
  ↪   directory.
- **Tools**: `tools/`
  - The `tools/` directory contains various tools that LLMs can use to perform specific
  ↪   tasks.
  - Each tool must have a `tool_info()` function that returns a JSON object containing
  ↪   'name', 'description', and 'input_schema'. The 'input_schema' should be a JSON
  ↪   object containing 'type', 'properties', and 'required'.
  - Each tool must have a `tool_function()` function that takes the arguments defined
  ↪   in input_schema, performs the tool's task, and returns a string.
  - See other tools for reference.
- **Utilities**: `utils/`
  - The `utils/` directory contains utility functions used across the codebase.

- **Additional Details**:
  - The agent is very good at automatically utilizing the right available tools at the
  ↪   right time. So do not have an agentic flow that explicitly forces a tool's usage.
  - Common tools, such as file editing and bash commands, are easy for the agent to
  ↪   recognize and use appropriately. However, more complex and niche tools may
  ↪   require explicit instructions in the prompt.
  - Tools should be designed to be as general as possible, ensuring they work across
  ↪   any GitHub repository. Avoid hardcoding repository-specific details or behaviors
  ↪   (e.g., paths).
  - Do not use 'while True' loops in the agent's code. This can cause the agent to get
  ↪   stuck and not respond.
  - Verify the implementation details of helper functions prior to usage to ensure
  ↪   proper integration and expected behavior.
  - Do not install additional packages or dependencies directly. Update
  ↪   `requirements.txt` if new dependencies are required and install them using `pip
  ↪   install -r requirements.txt`.

Here is the implementation of the coding agent.

# Coding Agent Implementation
----- Coding Agent Implementation Start -----
{code}
----- Coding Agent Implementation End -----

Your task is to identify ONE detailed plan that would improve the agent's coding
↪   ability. The improvement should not be specific to any particular GitHub issue or
↪   repository.

Here is the log for the coding agent trying to solve the GitHub issues but failed.

# Agent Running Log
----- Agent Running Log Start -----
{md_log}
----- Agent Running Log End -----

# GitHub Issue
The GitHub issue that the agent is trying to solve.
----- GitHub Issue Start -----
{github_issue}
----- GitHub Issue End -----

# Predicted Patch
The agent's predicted patch to solve the issue.
----- Predicted Patch Start -----
{predicted_patch}
----- Predicted Patch End -----

# Private Test Patch
```

---

```
SWE-bench's official private tests to detect whether the issue is solved. This is not
↪  available to the agent during evaluation. The agent should try to implement its own
↪  tests.
----- Private Test Patch Start -----
{test_patch}
----- Private Test Patch End -----

# Issue Test Results
The test results from SWE-bench using the above official private tests.
----- Issue Test Results Start -----
{eval_log}
----- Issue Test Results End -----

Respond precisely in the following format including the JSON start and end markers:

```json
<JSON>
```

In <JSON>, provide a JSON response with the following fields:
- "log_summarization": Analyze the above logs and summarize how the agent tried to
↪  solve the GitHub issue. Note which tools and how they are used, the agent's
↪  problem-solving approach, and any issues encountered.
- "potential_improvements": Identify potential improvements to the coding agent that
↪  could enhance its coding capabilities. Focus on the agent's general coding
↪  abilities (e.g., better or new tools usable across any repository) rather than
↪  issue-specific fixes (e.g., tools only usable in one framework). All necessary
↪  dependencies and environment setup have already been handled, so do not focus on
↪  these aspects.
- "improvement_proposal": Choose ONE high-impact improvement from the identified
↪  potential improvements and describe it in detail. This should be a focused and
↪  comprehensive plan to enhance the agent's overall coding ability.
- "implementation_suggestion": Referring to the coding agent's summary and
↪  implementation, think critically about what feature or tool could be added or
↪  improved to best implement the proposed improvement. If the proposed feature can be
↪  implemented by modifying the existing tools, describe the modifications needed,
↪  instead of suggesting a new tool.
- "problem_description": Phrase the improvement proposal and implementation suggestion
↪  as a GitHub issue description. It should clearly describe the feature so that a
↪  software engineer viewing the issue and the repository can implement it.

Your response will be automatically parsed, so ensure that the string response is
↪  precisely in the correct format. Do NOT include the `<JSON>` tag in your output.
```

While the private test patch is included in the diagnosis prompt, one could imagine that an easy
way for the agent to score well on the task would be to hard code the solutions. However, from
manual inspection of the logs and agent modifications, we have not observed any problematic logic
or behavior indicative of memorization or overfitting to specific private test cases. Furthermore,
improvements discovered on one task or LLM model often generalize well to others (Figure 4),
showing that these improvements are not task-specific but instead reflect general changes that enhance
overall performance.

Additional prompt for Polyglot experiments:

### Additional Prompt for o3-mini Tool Use and Polyglot

```
### DOC: tool function schema

Carefully consider whether to add/enhance the current tool or edit the workflow in
↪  forward()

Pay special attention to making sure that "required" and "type" are always at the
↪  correct level of nesting. For example, "required" should be at the same level as
↪  "properties", not inside it.
Make sure that every property, no matter how short, has a type and description
↪  correctly nested inside it.
Other arguments than you have seen are not permitted. For example, in
↪  "edit_line_ranges" with "type": "array", arguments like "minItems" and "maxItems"
↪  are not permitted.

...

Here is the log for the coding agent trying to solve a programming task.
A task is in one programming language, but the coding agent needs to deal with
↪  different languages including C++, Go, Java, JavaScript, Python, and Rust.
```

### C.4 PSEUDOCODE

This is the pseudocode of the DGM algorithm, described in Section 3:

---
**Algorithm 1:** Darwin Gödel Machine
---
**Input:** Initial coding agent $g_0$, benchmark suite $B$, maximum iterations $T$
**Output:** Archive of agents $\mathcal{A}$

$s_0 \leftarrow$ evaluate$(g_0, B)$         // Evaluate the base agent
**initialize** $\mathcal{A} \leftarrow \{(g_0, s_0)\}$       // Start with the base agent
**for** $t \leftarrow 1$ **to** $T$ **do**
   $\mathcal{P} \leftarrow$ SelectParents$(\mathcal{A})$       // Select parent agents
   **foreach** $p \in \mathcal{P}$ **do**
      $c \leftarrow p$.modify$(p)$       // Self-modification
      $s \leftarrow$ evaluate$(c, B)$       // Evaluate on benchmark
      **if** $c.is\_valid()$ **then**
         $\mathcal{A} \leftarrow \mathcal{A} \cup \{(c, s)\}$   // Keep children capable of code editing
      **end**
   **end**
**end**
**return** $\mathcal{A}$

---

This is the pseudocode of the baseline DGM without self-improving agents, described in Section 4.3:

---
**Algorithm 2:** Darwin Gödel Machine without Self-improving agents
---
**Input:** Initial coding agent $g_0$, benchmark suite $B$, maximum iterations $T$
**Output:** Archive of agents $\mathcal{A}$

$s_0 \leftarrow$ evaluate$(g_0, B)$         // Evaluate the base agent
**initialize** $\mathcal{A} \leftarrow \{(g_0, s_0)\}$       // Start with the base agent
**for** $t \leftarrow 1$ **to** $T$ **do**
   $\mathcal{P} \leftarrow$ SelectParents$(\mathcal{A})$       // Select parent agents
   **foreach** $p \in \mathcal{P}$ **do**
      $c \leftarrow g_0$.modify$(p)$       // Modify with base agent
      $s \leftarrow$ evaluate$(c, B)$       // Evaluate on benchmark
      **if** $c.is\_valid()$ **then**
         $\mathcal{A} \leftarrow \mathcal{A} \cup \{(c, s)\}$   // Keep children capable of code editing
      **end**
   **end**
**end**
**return** $\mathcal{A}$

---

This is the pseudocode of the baseline DGM without open-ended exploration, described in Section 4.3:

---

**Algorithm 3:** Darwin Gödel Machine without Open-ended exploration

---

**Input:** Initial coding agent $g_0$, benchmark suite $B$, maximum iterations $T$
**Output:** Archive of agents $\mathcal{A}$

$s_0 \leftarrow \text{evaluate}(g_0, B)$               `// Evaluate the base agent`
**initialize** $\mathcal{A} \leftarrow \{(g_0, s_0)\}$          `// Start with the base agent`
**for** $t \leftarrow 1$ **to** $T$ **do**
    $\mathcal{P} \leftarrow \text{SelectParents}(\mathcal{A})$          `// Select parent agents`
    **foreach** $p \in \mathcal{P}$ **do**
        $c \leftarrow p.\text{modify}(p)$           `// Self-modification`
        $s \leftarrow \text{evaluate}(c, B)$        `// Evaluate on benchmark`
        **if** $c.is\_valid()$ **then**
            $\mathcal{A} \leftarrow \{(c, s)\}$      `// Only keep the latest agent`
        **end**
    **end**
**end**
**return** $\mathcal{A}$

---

## D    EXPERIMENT DETAILS

### D.1    HYPERPARAMETERS FOR FOUNDATION MODELS

Table 3 shows the foundation model used in each experiment setting, as described in Section 4.1. Since SWE-bench is a more challenging coding benchmark, we use a stronger coding model, Claude 3.5 Sonnet (New) (based on our preliminary testing). To enable faster iterations and avoid the same rate limits as Claude, we use o3-mini for Polyglot experiments. The temperature for all FMs in every setting is set to 1.0.

Table 3: Foundation models used in each experiment setting.

| Benchmark | SWE-bench | Polyglot |
|---|---|---|
| Self-modification | Claude 3.5 Sonnet (New) | Claude 3.5 Sonnet (New) |
| Evaluation | Claude 3.5 Sonnet (New) | o3-mini |

## E    BENCHMARK DETAILS

### E.1    COST ESTIMATE

The estimated cost of completing a single run of the DGM on SWE-bench, as presented in Section 4, is about USD 22,000. In comparison, the estimated cost of completing a single run of either baseline (DGM w/o self-improve or DGM w/o open-ended exploration) on SWE-bench is about USD 10,000. Although the DGM is considerably more costly than the baselines, a method that can continuously improve, even at a higher cost, is preferable to one that fails to improve or stagnates at a level of performance that may never match that of the DGM. A more granular break down is:

| LLM | Benchmark | Number of Tasks | Cost Estimate (USD) |
|---|---|---|---|
| Claude 3.5 Sonnet (New) | SWE-bench | 60 | $350 |
| o3-mini | Polyglot | 60 | $5 |

We acknowledge that the current experiments on SWE-bench require considerable compute. Hence, we also include experiments on another benchmark, Polyglot, with significantly lower costs. This suggests that expenses vary greatly by task complexity, with SWE-bench being among the more complex and resource-intensive coding benchmarks. Moreover, several impactful methods (e.g., LLM training at its inception) were characterized substantial computational demands initially. Similar to

these pioneering works, we hope to open the door to future research on improving the efficiency and scalability of our approach. In addition, many leading coding agents on the SWE-bench leaderboard are backed by industrial companies employing expert full-time researchers and engineers, which incurs substantial human labor costs. In contrast, our approach achieves SoTA-level performance through fully autonomous self-improvement without human intervention, potentially offering greater efficiency when considering the comparative costs of specialized AI development talent versus API usage. Finally, as FMs continue to improve and compute costs continue to decline, methods like the DGM will become increasingly efficient and accessible.

Also, higher-performing agents discovered by the DGM do indeed incur greater inference costs than the initial agent, but cost and performance are not strictly correlated, where some expensive agents underperform cheaper ones.

## E.2    SWE-BENCH TASKS

Initial 10 tasks for verifying basic functionality of a coding agent:

- `django__django-10973`
- `django__django-11066`
- `django__django-12754`
- `django__django-15930`
- `django__django-13279`
- `django__django-16661`
- `django__django-13346`
- `django__django-10880`
- `django__django-10999`
- `django__django-11087`

Additional 50 tasks for estimating general effectiveness of a coding agent:

- `django__django-9296`
- `django__django-11790`
- `django__django-11815`
- `django__django-11848`
- `django__django-11880`
- `django__django-11885`
- `django__django-11951`
- `django__django-11964`
- `django__django-11999`
- `django__django-12039`
- `django__django-12050`
- `django__django-12143`
- `django__django-12155`
- `django__django-12193`
- `django__django-12209`
- `django__django-12262`
- `django__django-12273`
- `django__django-12276`
- `django__django-12304`
- `django__django-12308`
- `django__django-12325`
- `django__django-12406`
- `django__django-12708`
- `django__django-12713`
- `django__django-12774`
- `sphinx-doc__sphinx-7454`
- `sphinx-doc__sphinx-7590`
- `sphinx-doc__sphinx-7748`
- `sphinx-doc__sphinx-7757`
- `sphinx-doc__sphinx-7985`
- `sphinx-doc__sphinx-8035`
- `sphinx-doc__sphinx-8056`
- `sphinx-doc__sphinx-8265`
- `sphinx-doc__sphinx-8269`
- `sphinx-doc__sphinx-8475`
- `sphinx-doc__sphinx-8548`
- `sphinx-doc__sphinx-8551`
- `sphinx-doc__sphinx-8638`
- `sphinx-doc__sphinx-8721`
- `sphinx-doc__sphinx-9229`
- `sphinx-doc__sphinx-9230`
- `sphinx-doc__sphinx-9281`
- `sphinx-doc__sphinx-9320`
- `sphinx-doc__sphinx-9367`
- `sphinx-doc__sphinx-9461`
- `sphinx-doc__sphinx-9698`
- `sphinx-doc__sphinx-10449`
- `sphinx-doc__sphinx-10466`
- `sphinx-doc__sphinx-10673`
- `sphinx-doc__sphinx-11510`

Additional 140 tasks for more accurate assessment of a coding agent's performance:

- `astropy__astropy-12907`
- `astropy__astropy-13033`
- `astropy__astropy-13236`
- `astropy__astropy-13398`
- `astropy__astropy-13453`
- `astropy__astropy-13579`
- `astropy__astropy-13977`
- `astropy__astropy-14096`
- `astropy__astropy-14182`
- `astropy__astropy-14309`
- `astropy__astropy-14365`
- `astropy__astropy-14369`
- `astropy__astropy-14508`
- `astropy__astropy-14539`
- `astropy__astropy-14598`
- `astropy__astropy-14995`
- `astropy__astropy-7166`
- `astropy__astropy-7336`
- `astropy__astropy-7606`
- `astropy__astropy-7671`
- `astropy__astropy-8707`
- `astropy__astropy-8872`
- `django__django-10097`
- `django__django-10554`
- `django__django-10914`
- `django__django-11095`
- `django__django-11099`
- `django__django-11119`
- `django__django-11133`
- `django__django-11138`
- `django__django-11141`
- `django__django-11149`
- `django__django-11163`
- `django__django-11179`
- `django__django-11206`
- `django__django-11211`
- `django__django-11239`
- `django__django-11265`
- `django__django-11276`
- `django__django-11292`
- `django__django-11299`
- `django__django-11333`
- `django__django-11400`
- `django__django-11433`
- `django__django-11451`
- `django__django-11477`
- `django__django-11490`
- `django__django-11532`
- `django__django-11551`
- `django__django-11555`
- `django__django-11603`
- `django__django-11728`
- `django__django-11734`
- `django__django-11740`
- `django__django-11749`
- `django__django-11820`
- `django__django-12125`
- `django__django-12419`
- `django__django-12663`
- `django__django-12741`
- `django__django-12858`
- `django__django-12965`
- `django__django-13012`
- `django__django-13023`
- `django__django-13028`
- `django__django-13033`
- `django__django-13089`
- `django__django-13109`
- `django__django-13112`
- `django__django-13121`
- `django__django-13128`
- `django__django-13158`
- `django__django-13195`
- `django__django-13212`
- `django__django-13297`
- `django__django-13315`
- `django__django-13343`
- `django__django-13344`
- `django__django-13363`
- `django__django-13401`
- `django__django-13406`
- `django__django-13410`
- `django__django-13417`
- `django__django-13449`
- `django__django-13512`
- `django__django-13513`

- `django__django-13516`
- `django__django-13551`
- `django__django-13568`
- `django__django-13569`
- `django__django-13590`
- `django__django-13658`
- `django__django-13670`
- `django__django-13741`
- `django__django-13786`
- `django__django-13794`
- `django__django-13807`
- `django__django-13809`
- `django__django-13810`
- `django__django-13820`
- `django__django-13821`
- `django__django-13837`
- `django__django-13925`
- `django__django-13933`
- `django__django-13964`
- `django__django-14007`
- `django__django-14011`
- `django__django-14017`
- `django__django-14034`
- `django__django-14053`
- `django__django-14089`
- `django__django-14122`
- `django__django-14140`
- `django__django-14155`
- `django__django-14170`
- `django__django-14238`
- `django__django-14311`
- `django__django-14315`
- `django__django-14349`
- `django__django-14351`
- `django__django-14373`
- `django__django-14376`
- `django__django-14404`
- `django__django-14434`
- `django__django-14493`
- `django__django-14500`
- `django__django-14534`
- `django__django-14539`
- `django__django-14559`
- `django__django-14580`
- `django__django-14608`
- `django__django-14631`
- `django__django-14672`
- `django__django-14725`
- `django__django-14752`
- `django__django-14765`
- `django__django-14771`
- `django__django-14787`
- `django__django-14792`
- `django__django-14855`

### E.3 POLYGLOT TASKS

Initial 10 tasks for verifying basic functionality of a coding agent:

- `go__dominoes`
- `cpp__all-your-base`
- `python__dominoes`
- `java__sgf-parsing`
- `javascript__robot-name`
- `rust__variable-length-quantity`
- `python__beer-song`
- `go__book-store`
- `javascript__bottle-song`
- `rust__bowling`

Additional 50 tasks for estimating general effectiveness of a coding agent:

- `javascript__queen-attack`
- `rust__wordy`
- `python__dot-dsl`
- `java__satellite`
- `cpp__diamond`
- `rust__accumulate`
- `go__error-handling`
- `cpp__queen-attack`
- `rust__poker`
- `python__sgf-parsing`

- `rust__react`
- `java__ledger`
- `go__connect`
- `rust__macros`
- `javascript__triangle`
- `java__zipper`
- `java__bowling`
- `python__tree-building`
- `javascript__say`
- `java__wordy`
- `python__food-chain`
- `javascript__wordy`
- `python__poker`
- `javascript__grade-school`
- `cpp__gigasecond`
- `java__forth`
- `python__dominoes`
- `go__word-search`
- `javascript__simple-linked-list`
- `go__counter`
- `java__react`
- `javascript__ocr-numbers`
- `python__scale-generator`
- `java__go-counting`
- `rust__doubly-linked-list`
- `python__grade-school`
- `javascript__forth`
- `python__wordy`
- `java__mazy-mice`
- `cpp__bank-account`
- `python__zipper`
- `java__custom-set`
- `java__rest-api`
- `go__transpose`
- `rust__gigasecond`
- `rust__say`
- `go__food-chain`
- `rust__pig-latin`
- `go__markdown`
- `go__crypto-square`

### E.4  SWE-BENCH STATE-OF-THE-ART

At the time of writing this paper (16 April 2025), the highest performing, checked (i.e., the SWE-bench team received access to the system and were able to reproduce the patch generations), open-source entry on SWE-bench Verified is OpenHands + CodeAct v2.1 (claude-3-5-sonnet-20241022) (Wang et al., 2024), achieving 53.0%. Only considering the same subset of 200 tasks used by the DGM (Appendix E.2), OpenHands + CodeAct v2.1 (claude-3-5-sonnet-20241022) achieves 51.0%.

### E.5  POLYGLOT REPRESENTATIVE AGENT

Aider (Gauthier, 2024), a popular coding agent in the community, was published in Spring 2024. It has garnered over 33,000 stars on GitHub and has been continuously developed and tested against the Polyglot benchmark for over a year by human developers, primarily to evaluate its performance. Aider has also become a standard baseline for assessing the performance of different models, with the current top performers on the Polyglot benchmark being a mix of o3 (high) and GPT-4.1. We adopt a setup similar to that of the Polyglot leaderboard, with one key difference: the leaderboard reports pass@2 performance, where the agent can view feedback from ground-truth tests once. In contrast, we use a pass@1 setting, where the agent never sees the results of ground-truth tests, as we believe this more closely reflects realistic coding applications.

## F  BEST-DISCOVERED AGENTS

### F.1  DGM ON SWE-BENCH

Diff patches contributing to the best agent discovered by the DGM on SWE-bench:

```diff
diff --git a/coding_agent.py b/coding_agent.py
index 2cd395a..9a2cc2f 100644
--- a/coding_agent.py
+++ b/coding_agent.py
@@ -4,6 +4,7 @@ import logging
 from logging.handlers import RotatingFileHandler
```

```
 import os
 import threading
+import re

 from llm_withtools import CLAUDE_MODEL, OPENAI_MODEL, chat_with_agent
 from utils.eval_utils import get_report_score, msg_history_to_report, score_tie_breaker
@@ -63,6 +64,42 @@ def safe_log(message, level=logging.INFO):
     else:
         print(f"Warning: No logger found for thread {threading.get_ident()}")

+def is_patch_valid(patch_str):
+ """
+ Parse the patch to check if any non-test source files are modified.
+ Returns (bool, str) tuple: (is_valid, reason)
+ """
+ if not patch_str or patch_str.isspace():
+ return False, "Empty patch"
+
+ # Parse the patch to find modified files
+ modified_files = []
+ diff_header_pattern = re.compile(r'^\+\+\+ b/(.+)$', re.MULTILINE)
+ for match in diff_header_pattern.finditer(patch_str):
+ filepath = match.group(1)
+ if filepath != '/dev/null': # Skip deleted files
+ modified_files.append(filepath)
+
+ if not modified_files:
+ return False, "No files modified"
+
+ # Check if any non-test files are modified
+ test_patterns = (
+ lambda f: f.startswith('tests/'),
+ lambda f: f.startswith('test_'),
+ lambda f: f.endswith('_test.py')
+ )
+
+ source_files = [
+ f for f in modified_files
+ if not any(pattern(f) for pattern in test_patterns)
+ ]
+
+ if not source_files:
+ return False, "Only test files were modified"
+
+ return True, "Valid patch with source file modifications"
+
 class AgenticSystem:
     def __init__(
         self,
@@ -73,6 +110,7 @@ class AgenticSystem:
         test_description=None,
         self_improve=False,
         instance_id=None,
+ max_retries=3,
     ):
         self.problem_statement = problem_statement
         self.git_tempdir = git_tempdir
@@ -82,6 +120,7 @@ class AgenticSystem:
         self.self_improve = self_improve
         self.instance_id = instance_id if not self_improve else 'dgm'
         self.code_model = CLAUDE_MODEL
+ self.max_retries = max_retries

         # Initialize logger and store it in thread-local storage
         self.logger = setup_logger(chat_history_file)
@@ -153,7 +192,7 @@ Your task is to run the regression tests in the {self.git_tempdir}
     ↪ directory to
         """
         The forward function for the AgenticSystem.
         """
- instruction = f"""I have uploaded a Python code repository in the directory
     ↪ {self.git_tempdir}. Help solve the following problem.
+ base_instruction = f"""I have uploaded a Python code repository in the directory
     ↪ {self.git_tempdir}. Help solve the following problem.

 <problem_description>
 {self.problem_statement}
@@ -165,7 +204,39 @@ Your task is to run the regression tests in the {self.git_tempdir}
     ↪ directory to
```

```
 Your task is to make changes to the files in the {self.git_tempdir} directory to address
     ↪ the <problem_description>. I have already taken care of the required dependencies.
 """
- new_msg_history = chat_with_agent(instruction, model=self.code_model, msg_history=[],
     ↪ logging=safe_log)
+
+ retry_count = 0
+ while retry_count < self.max_retries:
+ safe_log(f"\n=== Attempt {retry_count + 1} of {self.max_retries} ===")
+
+ # Reset to base commit before each attempt
+ if retry_count > 0:
+ reset_to_commit(self.git_tempdir, self.base_commit)
+
+ # Add retry context to instruction if this is a retry attempt
+ instruction = base_instruction
+ if retry_count > 0:
+ instruction += f"""\nNOTE: Previous attempt(s) failed because they either produced empty
     ↪ patches or only modified test files.
+Please ensure your solution includes changes to the main source code files, not just test
     ↪ files."""
+
+ # Run the agent
+ new_msg_history = chat_with_agent(instruction, model=self.code_model, msg_history=[],
     ↪ logging=safe_log)
+
+ # Check the patch
+ patch = self.get_current_edits()
+ is_valid, reason = is_patch_valid(patch)
+
+ if is_valid:
+ safe_log(f"Valid patch generated: {reason}")
+ break
+ else:
+ safe_log(f"Invalid patch: {reason}")
+ if retry_count < self.max_retries - 1:
+ safe_log("Retrying with a new attempt...")
+ else:
+ safe_log("Maximum retries reached. Unable to generate a valid patch.")
+
+ retry_count += 1

 def main():
    parser = argparse.ArgumentParser(description='Process repository with an agentic
        ↪ system.')
@@ -177,6 +248,7 @@ def main():
    parser.add_argument('--test_description', default=None, required=False,
        ↪ help='Description of how to test the repository')
    parser.add_argument('--self_improve', default=False, action='store_true',
        ↪ help='Whether to self-improve the repository or solving swe')
    parser.add_argument('--instance_id', default=None, help='Instance ID for SWE issue')
+ parser.add_argument('--max_retries', type=int, default=3, help='Maximum number of patch
     ↪ generation attempts')
    args = parser.parse_args()

    # Process the repository
@@ -188,6 +260,7 @@ def main():
        test_description=args.test_description,
        self_improve=args.self_improve,
        instance_id=args.instance_id,
+ max_retries=args.max_retries,
    )

    # Run the agentic system to try to solve the problem
@@ -200,4 +273,4 @@ def main():
        f.write(model_patch)

 if __name__ == "__main__":
- main()
+ main()
\ No newline at end of file
diff --git a/tests/test_patch_validator.py b/tests/test_patch_validator.py
new file mode 100644
index 0000000..5689f7d
--- /dev/null
+++ b/tests/test_patch_validator.py
@@ -0,0 +1,77 @@
+import pytest
+from coding_agent import is_patch_valid
+
+def test_empty_patch():
```

```
+ # Test empty patch
+ is_valid, reason = is_patch_valid("")
+ assert not is_valid
+ assert reason == "Empty patch"
+
+ # Test whitespace-only patch
+ is_valid, reason = is_patch_valid(" \n ")
+ assert not is_valid
+ assert reason == "Empty patch"
+
+def test_test_only_patch():
+ patch = """
+diff --git a/tests/test_edit_tool.py b/tests/test_edit_tool.py
+index abc123..def456 100644
+--- a/tests/test_edit_tool.py
++++ b/tests/test_edit_tool.py
+@@ -10,6 +10,8 @@ def test_something():
+ assert True
++ assert 1 == 1
+"""
+ is_valid, reason = is_patch_valid(patch)
+ assert not is_valid
+ assert reason == "Only test files were modified"
+
+def test_source_file_patch():
+ patch = """
+diff --git a/tools/edit.py b/tools/edit.py
+index abc123..def456 100644
+--- a/tools/edit.py
++++ b/tools/edit.py
+@@ -10,6 +10,8 @@ class Editor:
+ def edit(self):
+ pass
++ return True
+"""
+ is_valid, reason = is_patch_valid(patch)
+ assert is_valid
+ assert reason == "Valid patch with source file modifications"
+
+def test_mixed_files_patch():
+ patch = """
+diff --git a/tools/edit.py b/tools/edit.py
+index abc123..def456 100644
+--- a/tools/edit.py
++++ b/tools/edit.py
+@@ -10,6 +10,8 @@ class Editor:
+ def edit(self):
+ pass
++ return True
+
+diff --git a/tests/test_edit.py b/tests/test_edit.py
+index abc123..def456 100644
+--- a/tests/test_edit.py
++++ b/tests/test_edit.py
+@@ -10,6 +10,8 @@ def test_something():
+ assert True
++ assert 1 == 1
+"""
+ is_valid, reason = is_patch_valid(patch)
+ assert is_valid
+ assert reason == "Valid patch with source file modifications"
+
+def test_no_files_modified():
+ patch = """
+diff --git a/nonexistent.py b/nonexistent.py
+deleted file mode 100644
+index abc123..0000000
+--- a/nonexistent.py
++++ /dev/null
+"""
+ is_valid, reason = is_patch_valid(patch)
+ assert not is_valid
+ assert reason == "No files modified"
\ No newline at end of file
```

```
diff --git a/tools/edit.py b/tools/edit.py
index 59137ee..16ae521 100644
--- a/tools/edit.py
+++ b/tools/edit.py
@@ -1,16 +1,17 @@
```

```
 from pathlib import Path
 import subprocess
+from typing import Optional, List, Tuple, Union

 def tool_info():
     return {
         "name": "editor",
         "description": """Custom editing tool for viewing, creating, and editing files\n
 * State is persistent across command calls and discussions with the user.\n
-* If `path` is a file, `view` displays the entire file with line numbers. If `path` is a
     ↪ directory, `view` lists non-hidden files and directories up to 2 levels deep.\n
+* If `path` is a file, `view` displays the file with line numbers. With optional
     ↪ `view_range` [start, end], it displays only specified lines. Use -1 in `end` for
     ↪ all remaining lines.\n
+* If `path` is a directory, `view` lists non-hidden files and directories up to 2 levels
     ↪ deep.\n
 * The `create` command cannot be used if the specified `path` already exists as a file.\n
 * If a `command` generates a long output, it will be truncated and marked with `<response
     ↪ clipped>`.\n
-* The `edit` command overwrites the entire file with the provided `file_text`.\n
-* No partial/line-range edits or partial viewing are supported.""",
+* The `edit` command overwrites the entire file with the provided `file_text`.""",
         "input_schema": {
             "type": "object",
             "properties": {
@@ -26,6 +27,13 @@ def tool_info():
                 "file_text": {
                     "description": "Required parameter of `create` or `edit` command,
                         ↪ containing the content for the entire file.",
                     "type": "string"
+                },
+                "view_range": {
+                    "description": "Optional parameter for `view` command. Array of [start_line, end_line]
+                        ↪ (1-based). Use -1 for end_line to read until end of file.",
+                    "type": "array",
+                    "items": {"type": "integer"},
+                    "minItems": 2,
+                    "maxItems": 2
                 }
             },
             "required": ["command", "path"]
@@ -89,6 +97,46 @@ def read_file(path: Path) -> str:
     except Exception as e:
         raise ValueError(f"Failed to read file: {e}")

+def read_file_range(path: Path, line_range: Optional[List[int]] = None) -> Tuple[str,
+    ↪ int]:
+    """
+    Read and return file contents within specified line range.
+    Returns tuple of (content, start_line).
+
+    Args:
+        path: Path object for the file
+        line_range: Optional [start, end] line numbers (1-based). Use -1 for end to read until
+            ↪ EOF.
+    """
+    try:
+        if line_range is None:
+            return read_file(path), 1
+
+        start, end = line_range
+        if start < 1:
+            raise ValueError("Start line must be >= 1")
+        if end != -1 and end < start:
+            raise ValueError("End line must be >= start line or -1")
+
+        with path.open() as f:
+            # Skip lines before start
+            for _ in range(start - 1):
+                next(f, None)
+
+            lines = []
+            current_line = start
+            while True:
+                line = next(f, None)
+                if line is None:  # EOF
+                    break
+                if end != -1 and current_line > end:
+                    break
+                lines.append(line.rstrip('\n'))
+                current_line += 1
```

```diff
+
+ return '\n'.join(lines), start
+
+ except Exception as e:
+ raise ValueError(f"Failed to read file range: {e}")
+
 def write_file(path: Path, content: str):
     """Write (overwrite) entire file contents."""
     try:
@@ -96,9 +144,18 @@ def write_file(path: Path, content: str):
     except Exception as e:
         raise ValueError(f"Failed to write file: {e}")

-def view_path(path_obj: Path) -> str:
- """View the entire file contents or directory listing."""
+def view_path(path_obj: Path, view_range: Optional[List[int]] = None) -> str:
+ """
+ View the file contents (optionally within a range) or directory listing.
+
+ Args:
+ path_obj: Path object for the file or directory
+ view_range: Optional [start, end] line numbers for file viewing
+ """
     if path_obj.is_dir():
+ if view_range is not None:
+ raise ValueError("view_range is not supported for directory listings")
+
         # For directories: list non-hidden files up to 2 levels deep
         try:
             result = subprocess.run(
@@ -115,14 +172,14 @@ def view_path(path_obj: Path) -> str:
         except Exception as e:
             raise ValueError(f"Failed to list directory: {e}")

- # If it's a file, show the entire file with line numbers
- content = read_file(path_obj)
- return format_output(content, str(path_obj))
+ # If it's a file, show the file content (with optional line range)
+ content, start_line = read_file_range(path_obj, view_range)
+ return format_output(content, str(path_obj), start_line)

-def tool_function(command: str, path: str, file_text: str = None) -> str:
+def tool_function(command: str, path: str, file_text: str = None, view_range:
      ↪ Optional[List[int]] = None) -> str:
     """
     Main tool function that handles:
- - 'view' : View the entire file or directory listing
+ - 'view' : View file or directory listing, optionally within line range for files
     - 'create': Create a new file with the given file_text
     - 'edit' : Overwrite an existing file with file_text
     """
@@ -130,7 +187,7 @@ def tool_function(command: str, path: str, file_text: str = None) ->
     ↪ str:
     path_obj = validate_path(path, command)

         if command == "view":
- return view_path(path_obj)
+ return view_path(path_obj, view_range)

         elif command == "create":
             if file_text is None:
@@ -152,4 +209,4 @@ def tool_function(command: str, path: str, file_text: str = None) ->
     ↪ str:

 if __name__ == "__main__":
     # Example usage
- print(tool_function("view", "/home/ubuntu/xx/dgm/coding_agent.py"))
+ print(tool_function("view", "/home/ubuntu/xx/dgm/coding_agent.py"))
\ No newline at end of file
diff --git a/tests/test_tools/test_edit.py b/tests/test_tools/test_edit.py
new file mode 100644
index 0000000..04f535b
--- /dev/null
+++ b/tests/test_tools/test_edit.py
@@ -0,0 +1,54 @@
+import pytest
+from pathlib import Path
+from tools.edit import tool_function
+
+def test_view_line_range(tmp_path):
+ # Create a test file
```

```
+ test_file = tmp_path / "test.txt"
+ test_content = "line1\nline2\nline3\nline4\nline5\n"
+ test_file.write_text(test_content)
+
+ # Test viewing specific line range
+ result = tool_function("view", str(test_file), view_range=[2, 4])
+ assert "line2" in result
+ assert "line3" in result
+ assert "line4" in result
+ assert "line1" not in result
+ assert "line5" not in result
+ assert " 2\t" in result # Correct line numbering
+
+ # Test viewing from start to middle
+ result = tool_function("view", str(test_file), view_range=[1, 3])
+ assert "line1" in result
+ assert "line2" in result
+ assert "line3" in result
+ assert "line4" not in result
+ assert " 1\t" in result
+
+ # Test viewing from middle to end with -1
+ result = tool_function("view", str(test_file), view_range=[3, -1])
+ assert "line1" not in result
+ assert "line2" not in result
+ assert "line3" in result
+ assert "line4" in result
+ assert "line5" in result
+ assert " 3\t" in result
+
+def test_view_range_validation(tmp_path):
+ # Create a test file
+ test_file = tmp_path / "test.txt"
+ test_content = "line1\nline2\nline3\n"
+ test_file.write_text(test_content)
+
+ # Test invalid start line
+ result = tool_function("view", str(test_file), view_range=[0, 2])
+ assert "Failed to read file range: Start line must be >= 1" in result
+
+ # Test invalid range (end < start)
+ result = tool_function("view", str(test_file), view_range=[2, 1])
+ assert "Failed to read file range: End line must be >= start line or -1" in result
+
+def test_view_range_with_directory(tmp_path):
+ # Test that view_range is rejected for directories
+ result = tool_function("view", str(tmp_path), view_range=[1, 10])
+ assert "Error: view_range is not supported for directory listings" in result
\ No newline at end of file
```

```
diff --git a/tools/edit.py b/tools/edit.py
index 16ae521..757f5c2 100644
--- a/tools/edit.py
+++ b/tools/edit.py
@@ -11,21 +11,21 @@ def tool_info():
 * If `path` is a directory, `view` lists non-hidden files and directories up to 2 levels
      ↪ deep.\n
 * The `create` command cannot be used if the specified `path` already exists as a file.\n
 * If a `command` generates a long output, it will be truncated and marked with `<response
      ↪ clipped>`.\n
-* The `edit` command overwrites the entire file with the provided `file_text`.""",
+* The `str_replace` command replaces a unique occurrence of old_str with new_str, failing
     ↪ if old_str is not found or appears multiple times.""",
        "input_schema": {
           "type": "object",
           "properties": {
              "command": {
                 "type": "string",
- "enum": ["view", "create", "edit"],
- "description": "The command to run: `view`, `create`, or `edit`."
+ "enum": ["view", "create", "str_replace"],
+ "description": "The command to run: `view`, `create`, or `str_replace`."
              },
              "path": {
                 "description": "Absolute path to file or directory, e.g. `/repo/file.py`
                      ↪ or `/repo`.",
                 "type": "string"
              },
              "file_text": {
```

```diff
- "description": "Required parameter of `create` or `edit` command, containing the content
      ↪ for the entire file.",
+ "description": "Required parameter of `create` command, containing the content for the
      ↪ entire file.",
                "type": "string"
            },
            "view_range": {
@@ -34,6 +34,14 @@ def tool_info():
                "items": {"type": "integer"},
                "minItems": 2,
                "maxItems": 2
+ },
+ "old_str": {
+ "description": "Required parameter of `str_replace` command, containing the exact text
      ↪ to find and replace.",
+ "type": "string"
+ },
+ "new_str": {
+ "description": "Required parameter of `str_replace` command, containing the new text to
      ↪ replace old_str with.",
+ "type": "string"
                }
            },
            "required": ["command", "path"]
@@ -51,7 +59,7 @@ def validate_path(path: str, command: str) -> Path:
    Validate the file path for each command:
      - 'view': path may be a file or directory; must exist.
      - 'create': path must not exist (for new file creation).
- - 'edit': path must exist (for overwriting).
+ - 'str_replace': path must exist and be a file.
    """
    path_obj = Path(path)

@@ -69,7 +77,7 @@ def validate_path(path: str, command: str) -> Path:
        # Path must not exist
        if path_obj.exists():
            raise ValueError(f"Cannot create new file; {path} already exists.")
- elif command == "edit":
+ elif command == "str_replace":
        # Path must exist and must be a file
        if not path_obj.exists():
            raise ValueError(f"The file {path} does not exist.")
@@ -144,6 +152,28 @@ def write_file(path: Path, content: str):
    except Exception as e:
        raise ValueError(f"Failed to write file: {e}")

+def str_replace_in_file(path: Path, old_str: str, new_str: str) -> str:
+ """
+ Replace an exact occurrence of old_str with new_str in the file.
+ Only performs the replacement if old_str occurs exactly once.
+ Returns a message indicating success or failure.
+ """
+ try:
+ content = read_file(path)
+ occurrences = content.count(old_str)
+
+ if occurrences == 0:
+ return f"Error: Could not find the exact text to replace in {path}"
+ elif occurrences > 1:
+ return f"Error: Found multiple ({occurrences}) occurrences of the text in {path}. Must
      ↪ be unique."
+ else:
+ new_content = content.replace(old_str, new_str)
+ write_file(path, new_content)
+ return f"Successfully replaced text in {path}"
+
+ except Exception as e:
+ return f"Error during string replacement: {e}"
+
 def view_path(path_obj: Path, view_range: Optional[List[int]] = None) -> str:
    """
    View the file contents (optionally within a range) or directory listing.
@@ -176,12 +206,13 @@ def view_path(path_obj: Path, view_range: Optional[List[int]] =
      ↪ None) -> str:
    content, start_line = read_file_range(path_obj, view_range)
    return format_output(content, str(path_obj), start_line)

-def tool_function(command: str, path: str, file_text: str = None, view_range:
      ↪ Optional[List[int]] = None) -> str:
+def tool_function(command: str, path: str, file_text: str = None, view_range:
      ↪ Optional[List[int]] = None,
```

```diff
+ old_str: str = None, new_str: str = None) -> str:
      """
    Main tool function that handles:
- - 'view' : View file or directory listing, optionally within line range for files
- - 'create': Create a new file with the given file_text
- - 'edit' : Overwrite an existing file with file_text
+ - 'view' : View file or directory listing, optionally within line range for files
+ - 'create' : Create a new file with the given file_text
+ - 'str_replace': Replace exact occurrence of old_str with new_str in the file
      """
    try:
        path_obj = validate_path(path, command)
@@ -195,11 +226,10 @@ def tool_function(command: str, path: str, file_text: str = None,
      ↪ view_range: Op
            write_file(path_obj, file_text)
            return f"File created successfully at: {path}"

- elif command == "edit":
- if file_text is None:
- raise ValueError("Missing required `file_text` for 'edit' command.")
- write_file(path_obj, file_text)
- return f"File at {path} has been overwritten with new content."
+ elif command == "str_replace":
+ if old_str is None or new_str is None:
+ raise ValueError("Missing required `old_str` and/or `new_str` for 'str_replace'
      ↪ command.")
+ return str_replace_in_file(path_obj, old_str, new_str)

        else:
            raise ValueError(f"Unknown command: {command}")
diff --git a/tests/__init__.py b/tests/__init__.py
new file mode 100644
index 0000000..e69de29
diff --git a/tests/test_tools.py b/tests/test_tools.py
new file mode 100644
index 0000000..c7f242f
--- /dev/null
+++ b/tests/test_tools.py
@@ -0,0 +1,65 @@
+import pytest
+from pathlib import Path
+from tools.edit import tool_function
+
+# Test fixtures
+@pytest.fixture
+def temp_file(tmp_path):
+ file_path = tmp_path / "test.txt"
+ content = "line 1\nline 2\nline 3\n"
+ file_path.write_text(content)
+ return str(file_path)
+
+def test_str_replace_success(temp_file):
+ # Test successful replacement
+ result = tool_function(
+ command="str_replace",
+ path=temp_file,
+ old_str="line 2\n",
+ new_str="replaced line\n"
+ )
+ assert "Successfully replaced" in result
+ assert Path(temp_file).read_text() == "line 1\nreplaced line\nline 3\n"
+
+def test_str_replace_not_found(temp_file):
+ # Test when old_str is not found
+ result = tool_function(
+ command="str_replace",
+ path=temp_file,
+ old_str="nonexistent",
+ new_str="something"
+ )
+ assert "Could not find" in result
+ # Original file should be unchanged
+ assert Path(temp_file).read_text() == "line 1\nline 2\nline 3\n"
+
+def test_str_replace_multiple_occurrences(temp_file):
+ # First create a file with multiple occurrences
+ Path(temp_file).write_text("same\nsame\nsame\n")
+ result = tool_function(
+ command="str_replace",
+ path=temp_file,
+ old_str="same\n",
```

```
+ new_str="different\n"
+ )
+ assert "multiple" in result
+ # Original file should be unchanged
+ assert Path(temp_file).read_text() == "same\nsame\nsame\n"
+
+def test_str_replace_missing_params(temp_file):
+ # Test missing parameters
+ result = tool_function(
+ command="str_replace",
+ path=temp_file,
+ )
+ assert "Missing required" in result
+
+def test_str_replace_invalid_path():
+ # Test with non-existent file
+ result = tool_function(
+ command="str_replace",
+ path="/nonexistent/path",
+ old_str="old",
+ new_str="new"
+ )
+ assert "does not exist" in result
\ No newline at end of file
```

```
diff --git a/llm_withtools.py b/llm_withtools.py
index d1394bb..6cc3604 100644
--- a/llm_withtools.py
+++ b/llm_withtools.py
@@ -29,7 +29,7 @@ def process_tool_call(tools_dict, tool_name, tool_input):
 )
 def get_response_withtools(
     client, model, messages, tools, tool_choice,
- logging=None,
+ logging=None, system_message=None,
 ):
     try:
         if 'claude' in model:
@@ -52,13 +52,32 @@ def get_response_withtools(
             raise ValueError(f"Unsupported model: {model}")
         return response
     except Exception as e:
- logging(f"Error in get_response_withtools: {str(e)}")
+ error_msg = str(e)
+ logging(f"Error in get_response_withtools: {error_msg}")

         # Hitting the context window limit
- if 'Input is too long for requested model' in str(e):
- pass
+ if 'Input is too long for requested model' in error_msg or 'maximum context length' in
      ↪ error_msg:
+ if not system_message:
+ # Extract system message from the first message if available
+ system_message = messages[0].get('content', '') if messages else ''
+ if isinstance(system_message, list):
+ system_message = ' '.join(block['text'] for block in system_message if block['type'] ==
      ↪ 'text')
+
+ # Summarize the conversation history
+ summarized_messages = summarize_messages(client, model, messages, system_message)
+
+ # Retry with summarized messages
+ return get_response_withtools(
+ client=client,
+ model=model,
+ messages=summarized_messages,
+ tools=tools,
+ tool_choice=tool_choice,
+ logging=logging,
+ system_message=system_message
+ )
+
- raise # Re-raise the exception after logging
+ raise # Re-raise other exceptions

 def check_for_tool_use(response, model=''):
     """
@@ -247,6 +266,57 @@ def convert_msg_history_openai(msg_history):

     return new_msg_history
```

```
+def summarize_messages(client, model, messages, system_message):
+ """
+ Creates a condensed summary of older messages while preserving recent context.
+ Only summarizes assistant and user messages, keeps tool results as is for accuracy.
+ """
+ # Keep the most recent messages intact
+ recent_msgs = messages[-2:] if len(messages) > 2 else messages
+ if len(messages) <= 2:
+ return messages
+
+ # Prepare messages to be summarized
+ msgs_to_summarize = messages[:-2]
+
+ # Create a prompt to summarize the conversation
+ summary_request = "Please create a concise summary of this conversation that preserves
+     ↪ the key context and important details:"
+ for msg in msgs_to_summarize:
+ if isinstance(msg.get('content', ''), list):
+ content = ' '.join(block['text'] for block in msg['content'] if block['type'] == 'text')
+ else:
+ content = str(msg.get('content', ''))
+ if msg.get('role') in ['assistant', 'user']:
+ summary_request += f"\n{msg['role']}: {content}"
+
+ try:
+ # Get summary from the model
+ summary_response, _ = get_response_from_llm(
+ msg=summary_request,
+ client=client,
+ model=model,
+ system_message="You are a summarizer. Create a concise but informative summary.",
+ print_debug=False,
+ msg_history=[]
+ )
+
+ # Create new message history with the summary
+ summarized_history = [{
+ "role": "system",
+ "content": [{"type": "text", "text": system_message}]
+ }, {
+ "role": "assistant",
+ "content": [{"type": "text", "text": f"Previous conversation summary:
+     ↪ {summary_response}"}]
+ }]
+
+ # Add back the recent messages
+ summarized_history.extend(recent_msgs)
+
+ return summarized_history
+ except Exception:
+ # If summarization fails, return original messages with the most recent ones
+ return [messages[0]] + recent_msgs
+
 def convert_msg_history(msg_history, model=None):
     """
     Convert message history from the model-specific format to a generic format.
@@ -263,7 +333,14 @@ def chat_with_agent_manualtools(msg, model, msg_history=None,
     ↪ logging=print):
     if msg_history is None:
         msg_history = []
     system_message = f'You are a coding agent.\n\n{get_tooluse_prompt()}'
- new_msg_history = msg_history
+ new_msg_history = msg_history.copy() if msg_history else []
+
+ # Ensure system message is the first message in history
+ if not new_msg_history or new_msg_history[0].get('role') != 'system':
+ new_msg_history.insert(0, {
+ "role": "system",
+ "content": [{"type": "text", "text": system_message}]
+ })

     try:
         # Load all tools
```

```
diff --git a/coding_agent.py b/coding_agent.py
index 9a2cc2f..3f1bc1d 100644
--- a/coding_agent.py
+++ b/coding_agent.py
@@ -111,6 +111,7 @@ class AgenticSystem:
```

```diff
            self_improve=False,
            instance_id=None,
            max_retries=3,
+ num_candidates=3,
        ):
        self.problem_statement = problem_statement
        self.git_tempdir = git_tempdir
@@ -121,6 +122,7 @@ class AgenticSystem:
        self.instance_id = instance_id if not self_improve else 'dgm'
        self.code_model = CLAUDE_MODEL
        self.max_retries = max_retries
+ self.num_candidates = num_candidates

        # Initialize logger and store it in thread-local storage
        self.logger = setup_logger(chat_history_file)
@@ -190,7 +192,7 @@ Your task is to run the regression tests in the {self.git_tempdir}
     ↪ directory to

    def forward(self):
        """
- The forward function for the AgenticSystem.
+ The forward function for the AgenticSystem that generates and evaluates multiple
     ↪ candidate patches.
        """
        base_instruction = f"""I have uploaded a Python code repository in the directory
            ↪ {self.git_tempdir}. Help solve the following problem.

@@ -205,10 +207,18 @@ Your task is to run the regression tests in the {self.git_tempdir}
     ↪ directory to
 Your task is to make changes to the files in the {self.git_tempdir} directory to address
     ↪ the <problem_description>. I have already taken care of the required dependencies.
 """

+ # Get regression tests summary once at the start
+ regression_tests_summary = self.get_regression_tests()
+
+ # Lists to store candidates
+ valid_patches = []
+ valid_reports = []
+
        retry_count = 0
- while retry_count < self.max_retries:
+ while retry_count < self.max_retries and len(valid_patches) < self.num_candidates:
            safe_log(f"\n=== Attempt {retry_count + 1} of {self.max_retries} ===")
-
+ safe_log(f"Valid solutions so far: {len(valid_patches)} of {self.num_candidates}
     ↪ desired")
+
            # Reset to base commit before each attempt
            if retry_count > 0:
                reset_to_commit(self.git_tempdir, self.base_commit)
@@ -216,8 +226,8 @@ Your task is to make changes to the files in the {self.git_tempdir}
     ↪ directory to
            # Add retry context to instruction if this is a retry attempt
            instruction = base_instruction
            if retry_count > 0:
- instruction += f"""\nNOTE: Previous attempt(s) failed because they either produced empty
     ↪ patches or only modified test files.
-Please ensure your solution includes changes to the main source code files, not just test
     ↪ files."""
+ instruction += f"""\nNOTE: Previous attempt(s) did not produce enough valid solutions.
+Please provide a different approach to solve the problem. Your solution must include
     ↪ changes to the main source code files, not just test files."""

            # Run the agent
            new_msg_history = chat_with_agent(instruction, model=self.code_model,
                ↪ msg_history=[], logging=safe_log)
@@ -228,16 +238,45 @@ Please ensure your solution includes changes to the main source code
     ↪ files, not

            if is_valid:
                safe_log(f"Valid patch generated: {reason}")
- break
+ # Run regression tests for this candidate
+ test_report = self.run_regression_tests(regression_tests_summary)
+ test_score = get_report_score(test_report)
+ safe_log(f"Test score: {test_score}")
+
+ valid_patches.append(patch)
+ valid_reports.append(test_report)
+
```

```
+ if len(valid_patches) >= self.num_candidates:
+ break
            else:
                safe_log(f"Invalid patch: {reason}")
- if retry_count < self.max_retries - 1:
- safe_log("Retrying with a new attempt...")
- else:
- safe_log("Maximum retries reached. Unable to generate a valid patch.")

            retry_count += 1

+ if not valid_patches:
+ safe_log("Failed to generate any valid patches.")
+ return
+
+ # Use score_tie_breaker to select the best patch
+ safe_log(f"\n=== Selecting Best Solution from {len(valid_patches)} Candidates ===")
+ best_index = score_tie_breaker(
+ self.problem_statement,
+ valid_patches,
+ valid_reports,
+ logging=safe_log
+ )
+
+ # Reset to base and apply the best patch
+ reset_to_commit(self.git_tempdir, self.base_commit)
+ best_patch = valid_patches[best_index]
+ safe_log(f"\n=== Applying Best Solution (Candidate {best_index + 1}) ===")
+ apply_patch(self.git_tempdir, best_patch)
+
+ # Final validation of the selected patch
+ final_test_report = self.run_regression_tests(regression_tests_summary)
+ final_score = get_report_score(final_test_report)
+ safe_log(f"Final solution test score: {final_score}")
+
 def main():
     parser = argparse.ArgumentParser(description='Process repository with an agentic
         ↪ system.')
     parser.add_argument('--problem_statement', required=True, help='The problem statement
         ↪ to process')
@@ -249,6 +288,7 @@ def main():
     parser.add_argument('--self_improve', default=False, action='store_true',
         ↪ help='Whether to self-improve the repository or solving swe')
     parser.add_argument('--instance_id', default=None, help='Instance ID for SWE issue')
     parser.add_argument('--max_retries', type=int, default=3, help='Maximum number of
         ↪ patch generation attempts')
+ parser.add_argument('--num_candidates', type=int, default=3, help='Number of candidate
     ↪ solutions to generate')
     args = parser.parse_args()

     # Process the repository
@@ -261,6 +301,7 @@ def main():
       self_improve=args.self_improve,
       instance_id=args.instance_id,
       max_retries=args.max_retries,
+ num_candidates=args.num_candidates,
     )

     # Run the agentic system to try to solve the problem
```

```
diff --git a/coding_agent.py b/coding_agent.py
index 3f1bc1d..588938d 100644
--- a/coding_agent.py
+++ b/coding_agent.py
@@ -193,42 +193,59 @@ Your task is to run the regression tests in the {self.git_tempdir}
     ↪ directory to
     def forward(self):
         """
         The forward function for the AgenticSystem that generates and evaluates multiple
             ↪ candidate patches.
+ This version maintains history of prior valid patches and test results, only using the
     ↪ tie-breaker
+ when necessary.
         """
- base_instruction = f"""I have uploaded a Python code repository in the directory
     ↪ {self.git_tempdir}. Help solve the following problem.
-
-<problem_description>
-{self.problem_statement}
-</problem_description>
```

```
-
-<test_description>
-{self.test_description}
-</test_description>
-
-Your task is to make changes to the files in the {self.git_tempdir} directory to address
    ↪ the <problem_description>. I have already taken care of the required dependencies.
-"""
-
- # Get regression tests summary once at the start
        regression_tests_summary = self.get_regression_tests()

- # Lists to store candidates
+ # Lists to store all valid patches and their information
        valid_patches = []
        valid_reports = []
+ valid_scores = []
+ best_score = 0
+ best_patches_indices = [] # Indices of patches that share the best score

        retry_count = 0
        while retry_count < self.max_retries and len(valid_patches) < self.num_candidates:
            safe_log(f"\n=== Attempt {retry_count + 1} of {self.max_retries} ===")
            safe_log(f"Valid solutions so far: {len(valid_patches)} of {self.num_candidates}
                ↪ desired")
+ safe_log(f"Current best test score: {best_score}")

            # Reset to base commit before each attempt
            if retry_count > 0:
                reset_to_commit(self.git_tempdir, self.base_commit)

- # Add retry context to instruction if this is a retry attempt
- instruction = base_instruction
- if retry_count > 0:
- instruction += f"""\nNOTE: Previous attempt(s) did not produce enough valid solutions.
+ # Construct instruction with previous best solutions if available
+ instruction = f"""I have uploaded a Python code repository in the directory
    ↪ {self.git_tempdir}. Help solve the following problem.
+
+<problem_description>
+{self.problem_statement}
+</problem_description>
+
+<test_description>
+{self.test_description}
+</test_description>"""
+
+ # Add previous solutions context if available
+ if valid_patches and retry_count > 0:
+ previous_solutions = []
+ for i, (patch, report, score) in enumerate(zip(valid_patches, valid_reports,
    ↪ valid_scores)):
+ previous_solutions.append(f"""
+Previous Solution {i+1}:
+<code_changes>
+{patch}
+</code_changes>
+Test Score: {score}
+Test Report: {report}
+""")
+ instruction += "\n\nPrevious solution attempts:\n" + "\n".join(previous_solutions)
+ instruction += "\nPlease provide a new solution that addresses any limitations in the
    ↪ previous attempts or explores a different approach."
+ elif retry_count > 0:
+ instruction += """\nNOTE: Previous attempt(s) did not produce enough valid solutions.
 Please provide a different approach to solve the problem. Your solution must include
    ↪ changes to the main source code files, not just test files."""
+
+ instruction += f"\n\nYour task is to make changes to the files in the {self.git_tempdir}
    ↪ directory to address the <problem_description>. I have already taken care of the
    ↪ required dependencies."
+
            # Run the agent
            new_msg_history = chat_with_agent(instruction, model=self.code_model,
                ↪ msg_history=[], logging=safe_log)

@@ -245,6 +262,14 @@ Please provide a different approach to solve the problem. Your
    ↪ solution must inc

                valid_patches.append(patch)
                valid_reports.append(test_report)
```

```
+ valid_scores.append(test_score)
+
+ # Update best score and indices
+ if test_score > best_score:
+ best_score = test_score
+ best_patches_indices = [len(valid_patches) - 1]
+ elif test_score == best_score:
+ best_patches_indices.append(len(valid_patches) - 1)

            if len(valid_patches) >= self.num_candidates:
                break
@@ -257,25 +282,30 @@ Please provide a different approach to solve the problem. Your
    ↪ solution must inc
        safe_log("Failed to generate any valid patches.")
        return

- # Use score_tie_breaker to select the best patch
+ # Only use tie-breaker if we have multiple patches with the best score
        safe_log(f"\n=== Selecting Best Solution from {len(valid_patches)} Candidates ===")
- best_index = score_tie_breaker(
- self.problem_statement,
- valid_patches,
- valid_reports,
- logging=safe_log
- )
+ if len(best_patches_indices) > 1:
+ safe_log(f"Multiple solutions ({len(best_patches_indices)}) tied for best score
    ↪ {best_score}. Using tie-breaker.")
+ best_index = score_tie_breaker(
+ self.problem_statement,
+ [valid_patches[i] for i in best_patches_indices],
+ [valid_reports[i] for i in best_patches_indices],
+ logging=safe_log
+ )
+ best_index = best_patches_indices[best_index]
+ else:
+ best_index = best_patches_indices[0]

        # Reset to base and apply the best patch
        reset_to_commit(self.git_tempdir, self.base_commit)
        best_patch = valid_patches[best_index]
- safe_log(f"\n=== Applying Best Solution (Candidate {best_index + 1}) ===")
+ safe_log(f"\n=== Applying Best Solution (Candidate {best_index + 1}) with score
    ↪ {valid_scores[best_index]} ===")
        apply_patch(self.git_tempdir, best_patch)

        # Final validation of the selected patch
        final_test_report = self.run_regression_tests(regression_tests_summary)
        final_score = get_report_score(final_test_report)
- safe_log(f"Final solution test score: {final_score}")
+ safe_log(f"Final validation test score: {final_score}")

 def main():
    parser = argparse.ArgumentParser(description='Process repository with an agentic
        ↪ system.')
```

## F.2 DGM ON POLYGLOT

Diff patches contributing to the best agent discovered by the DGM on Polyglot:

```
diff --git a/coding_agent.py b/coding_agent.py
index 04ffb36..6639abd 100644
--- a/coding_agent.py
+++ b/coding_agent.py
@@ -4,6 +4,9 @@ import logging
 from logging.handlers import RotatingFileHandler
 import os
 import threading
+import json
+from dataclasses import dataclass
+from typing import List, Optional

 from llm_withtools import CLAUDE_MODEL, OPENAI_MODEL, chat_with_agent
 from utils.git_utils import diff_versus_commit, reset_to_commit, apply_patch
@@ -42,6 +45,14 @@ TEST_COMMANDS = {
 # Thread-local storage for logger instances
 thread_local = threading.local()
```

```
+@dataclass
+class SolutionAttempt:
+ """Class to store information about a solution attempt."""
+ patch: str # The patch content
+ test_output: str # Raw test output
+ test_success: bool # Whether tests passed
+ test_stats: dict # Test statistics (e.g., number of passed/failed tests)
+
 def get_thread_logger():
     """
    Get the logger instance specific to the current thread.
@@ -102,7 +113,8 @@ class AgenticSystem:
         chat_history_file='./chat_history.md',
         test_description=None,
         self_improve=False,
- language='python'
+ language='python',
+ max_attempts=3
       ):
       self.problem_statement = problem_statement
       self.git_tempdir = git_tempdir
@@ -111,6 +123,7 @@ class AgenticSystem:
       self.test_description = test_description
       self.self_improve = self_improve
       self.language = language
+ self.max_attempts = max_attempts

       # Set the code model based on whether self-improvement is enabled
       self.code_model = OPENAI_MODEL if not self_improve else CLAUDE_MODEL
@@ -137,11 +150,63 @@ class AgenticSystem:
       ]
       return new_msg_history

+ def run_tests(self) -> tuple[bool, str, dict]:
+ """Run tests and return success status, output, and test statistics."""
+ success = False
+ output = ""
+ stats = {"passed": 0, "failed": 0, "errors": 0, "total": 0}
+
+ try:
+ for command in TEST_COMMANDS.get(self.language, []):
+ proc = subprocess.run(
+ command,
+ cwd=self.git_tempdir,
+ capture_output=True,
+ text=True,
+ check=False
+ )
+ output += f"$ {' '.join(command)}\n{proc.stdout}\n{proc.stderr}\n"
+ success = proc.returncode == 0
+ if not success:
+ break
+
+ # Try to extract test statistics from output
+ # This is a simple example; you might want to add more sophisticated parsing
+ stats["passed"] = output.count("PASS") + output.count("ok")
+ stats["failed"] = output.count("FAIL") + output.count("not ok")
+ stats["errors"] = output.count("ERROR") + output.count("panic:")
+ stats["total"] = stats["passed"] + stats["failed"] + stats["errors"]
+
+ except Exception as e:
+ output = f"Error running tests: {str(e)}"
+ success = False
+
+ return success, output, stats
+
+ def analyze_test_results(self, attempts: List[SolutionAttempt]) -> str:
+ """Analyze test results and create a summary for the agent."""
+ summary = "# Test Results Analysis\n\n"
+
+ for i, attempt in enumerate(attempts, 1):
+ summary += f"## Attempt {i}\n"
+ summary += f"Test Success: {attempt.test_success}\n"
+ summary += f"Test Stats: {json.dumps(attempt.test_stats, indent=2)}\n"
+ summary += "Key test output:\n```\n"
+ # Extract relevant parts of test output (e.g., error messages)
+ key_output = "\n".join(line for line in attempt.test_output.split("\n")
+ if "FAIL" in line or "ERROR" in line or "PASS" in line)
+ summary += f"{key_output}\n```\n\n"
+
+ return summary
```

```
+    def forward(self):
         """
- The forward function for the AgenticSystem.
+ The forward function for the AgenticSystem with multiple solution attempts.
         """
- task = f"""I have uploaded a code repository in the directory {self.git_tempdir}. Help
     ↪ solve the following problem.
+ attempts: List[SolutionAttempt] = []
+ best_attempt: Optional[SolutionAttempt] = None
+
+ base_task = f"""I have uploaded a code repository in the directory {self.git_tempdir}.
     ↪ Help solve the following problem.

 <problem_description>
 {self.problem_statement}
@@ -149,8 +214,52 @@ class AgenticSystem:

 Your task is to make changes to the files in the {self.git_tempdir} directory to address
       ↪ the <problem_description>. I have already taken care of the required dependencies.
 """
- instruction = f"{task}\n\nPlease analyze the problem description carefully. Then make
     ↪ edits to the code files to complete the instruction."
- init_edit = chat_with_agent(instruction, model=self.code_model, msg_history=[],
     ↪ logging=safe_log)
+
+ for attempt_num in range(self.max_attempts):
+ # Reset to base commit for each attempt
+ reset_to_commit(self.git_tempdir, self.base_commit)
+
+ # Prepare the instruction based on previous attempts
+ instruction = base_task
+ if attempts:
+ # Add analysis of previous attempts
+ instruction += "\n\nPrevious solution attempts have been made. Here are the results:\n\n"
+ instruction += self.analyze_test_results(attempts)
+ instruction += "\nPlease analyze these results and provide an improved solution that
     ↪ addresses the issues found."
+ else:
+ instruction += "\nPlease analyze the problem description carefully. Then make edits to
     ↪ the code files to complete the instruction."
+
+ # Get the solution attempt
+ chat_with_agent(instruction, model=self.code_model, msg_history=[], logging=safe_log)
+
+ # Capture the current patch
+ current_patch = diff_versus_commit(self.git_tempdir, self.base_commit)
+
+ # Run tests and collect results
+ test_success, test_output, test_stats = self.run_tests()
+
+ # Create and store attempt
+ attempt = SolutionAttempt(
+ patch=current_patch,
+ test_output=test_output,
+ test_success=test_success,
+ test_stats=test_stats
+ )
+ attempts.append(attempt)
+
+ # Update best attempt if this one is better
+ if test_success and (best_attempt is None or
+ attempt.test_stats["passed"] > best_attempt.test_stats["passed"]):
+ best_attempt = attempt
+
+ # If tests pass perfectly, we can stop
+ if test_success and attempt.test_stats["failed"] == 0 and attempt.test_stats["errors"]
     ↪ == 0:
+ break
+
+ # Apply the best solution if found, otherwise use the last attempt
+ final_attempt = best_attempt or attempts[-1]
+ reset_to_commit(self.git_tempdir, self.base_commit)
+ apply_patch(self.git_tempdir, final_attempt.patch)

 def main():
    parser = argparse.ArgumentParser(description='Process repository with an agentic
        ↪ system.')
@@ -162,6 +271,7 @@ def main():
    parser.add_argument('--test_description', default=None, required=False,
        ↪ help='Description of how to test the repository')
```

```
    parser.add_argument('--self_improve', default=False, action='store_true',
        ↪ help='Whether to self-improve the repository or solving swe')
    parser.add_argument('--language', required=False, default="python", choices=['cpp',
        ↪ 'java', 'python', 'go', 'rust', 'javascript'], help='Task\'s programming
        ↪ language')
+ parser.add_argument('--max_attempts', type=int, default=3, help='Maximum number of
    ↪ solution attempts')
    args = parser.parse_args()

    # Process the repository
@@ -173,6 +283,7 @@ def main():
        test_description=args.test_description,
        self_improve=args.self_improve,
        language=args.language,
+ max_attempts=args.max_attempts,
    )

    # Run the agentic system to try to solve the problem
@@ -185,4 +296,4 @@ def main():
        f.write(model_patch)

 if __name__ == "__main__":
- main()
+ main()
\ No newline at end of file
```

```
diff --git a/tools/edit.py b/tools/edit.py
index d7184c3..87fa408 100644
--- a/tools/edit.py
+++ b/tools/edit.py
@@ -1,5 +1,9 @@
 from pathlib import Path
 import subprocess
+from typing import List, Optional, Dict
+
+# Global edit history for undo functionality
+edit_history: Dict[str, List[str]] = {}

 def tool_info():
     return {
@@ -10,14 +14,14 @@ def tool_info():
 * The `create` command cannot be used if the specified `path` already exists as a file.\n
 * If a `command` generates a long output, it will be truncated and marked with `<response
     ↪ clipped>`.\n
 * The `edit` command overwrites the entire file with the provided `file_text`.\n
-* No partial/line-range edits or partial viewing are supported.""",
+* Also supports viewing specific line ranges, string replacement, text insertion, and
+    ↪ undo operations.""",
        "input_schema": {
            "type": "object",
            "properties": {
                "command": {
                    "type": "string",
-                   "enum": ["view", "create", "edit"],
-                   "description": "The command to run: `view`, `create`, or `edit`."
+                   "enum": ["view", "create", "edit", "str_replace", "insert", "undo_edit"],
+                   "description": "The command to run: `view`, `create`, `edit`, `str_replace`, `insert`, or
+                       ↪ `undo_edit`."
                },
                "path": {
                    "description": "Absolute path to file or directory, e.g. `/repo/file.py`
                        ↪ or `/repo`.",
@@ -26,6 +30,23 @@ def tool_info():
                "file_text": {
                    "description": "Required parameter of `create` or `edit` command,
                        ↪ containing the content for the entire file.",
                    "type": "string"
+               },
+               "view_range": {
+                   "description": "Optional parameter for `view` command to display specific line range
+                       ↪ [start, end].",
+                   "type": "array",
+                   "items": {"type": "integer"}
+               },
+               "old_str": {
+                   "description": "Required parameter for `str_replace` command, string to replace.",
+                   "type": "string"
+               },
+               "new_str": {
```

```diff
+ "description": "Required parameter for `str_replace` and `insert` commands, new string
    ↪ to insert.",
+ "type": "string"
+ },
+ "insert_line": {
+ "description": "Required parameter for `insert` command, line number where to insert
    ↪ text.",
+ "type": "integer"
                }
        },
        "required": ["command", "path"]
@@ -43,7 +64,7 @@ def validate_path(path: str, command: str) -> Path:
    Validate the file path for each command.
      - 'view': path may be a file or directory; must exist.
      - 'create': path must not exist (for new file creation).
- - 'edit': path must exist (for overwriting).
+ - Others: path must exist as a file.
    """
    path_obj = Path(path)

@@ -61,14 +82,12 @@ def validate_path(path: str, command: str) -> Path:
        # Path must not exist
        if path_obj.exists():
            raise ValueError(f"Cannot create new file; {path} already exists.")
- elif command == "edit":
+ else:
        # Path must exist and must be a file
        if not path_obj.exists():
            raise ValueError(f"The file {path} does not exist.")
        if path_obj.is_dir():
            raise ValueError(f"{path} is a directory and cannot be edited as a file.")
- else:
- raise ValueError(f"Unknown or unsupported command: {command}")

    return path_obj

@@ -89,14 +108,21 @@ def read_file(path: Path) -> str:
    except Exception as e:
        raise ValueError(f"Failed to read file: {e}")

-def write_file(path: Path, content: str):
+def write_file(path: Path, content: str, save_history: bool = True):
    """Write (overwrite) entire file contents."""
    try:
+ if save_history:
+ # Save the current content to history before writing
+ if path.exists():
+ if str(path) not in edit_history:
+ edit_history[str(path)] = []
+ edit_history[str(path)].append(path.read_text())
+
        path.write_text(content)
    except Exception as e:
        raise ValueError(f"Failed to write file: {e}")

-def view_path(path_obj: Path) -> str:
+def view_path(path_obj: Path, view_range: Optional[List[int]] = None) -> str:
    """View the entire file contents or directory listing."""
    if path_obj.is_dir():
        # For directories: list non-hidden files up to 2 levels deep
@@ -115,35 +141,104 @@ def view_path(path_obj: Path) -> str:
        except Exception as e:
            raise ValueError(f"Failed to list directory: {e}")

- # If it's a file, show the entire file with line numbers
+ # If it's a file
    content = read_file(path_obj)
+ lines = content.splitlines()
+
+ # Handle line range viewing
+ if view_range:
+ if len(view_range) != 2:
+ raise ValueError("view_range must contain exactly two elements: [start, end]")
+ start, end = view_range
+ if start < 1 or end > len(lines):
+ raise ValueError(f"Invalid line range [{start}, {end}] for file with {len(lines)} lines")
+ # Adjust for 0-based indexing
+ content = "\n".join(lines[start-1:end])
+ return format_output(content, str(path_obj), init_line=start)
+
    return format_output(content, str(path_obj))
```

```diff
-def tool_function(command: str, path: str, file_text: str = None) -> str:
+def str_replace(path_obj: Path, old_str: str, new_str: str) -> str:
+ """Replace string in file, ensuring uniqueness."""
+ content = read_file(path_obj)
+ # Check for multiple occurrences
+ if content.count(old_str) > 1:
+ return f"Error: Multiple occurrences of '{old_str}' found. Replacement requires a unique
     ↪ match."
+ elif content.count(old_str) == 0:
+ return f"Error: String '{old_str}' not found in file."
+
+ new_content = content.replace(old_str, new_str)
+ write_file(path_obj, new_content)
+ return f"File at {path_obj} has been edited: replaced '{old_str}' with '{new_str}'."
+
+def insert_text(path_obj: Path, insert_line: int, new_str: str) -> str:
+ """Insert text at specified line number."""
+ content = read_file(path_obj)
+ lines = content.splitlines()
+
+ if insert_line < 1 or insert_line > len(lines) + 1:
+ raise ValueError(f"Invalid insert line {insert_line} for file with {len(lines)} lines")
+
+ # Insert the new text at the specified line (adjusting for 0-based index)
+ lines.insert(insert_line - 1, new_str.rstrip("\n"))
+ new_content = "\n".join(lines) + "\n"
+
+ write_file(path_obj, new_content)
+ return f"File at {path_obj} has been edited: inserted text at line {insert_line}."
+
+def undo_edit(path_obj: Path) -> str:
+ """Undo last edit operation on the file."""
+ path_str = str(path_obj)
+ if path_str not in edit_history or not edit_history[path_str]:
+ return "Error: No edit history available for this file."
+
+ # Restore the last saved content
+ previous_content = edit_history[path_str].pop()
+ write_file(path_obj, previous_content, save_history=False)
+ return f"Last edit on {path_obj} has been undone successfully."
+
+def tool_function(command: str, path: str, **kwargs) -> str:
     """
- Main tool function that handles:
- - 'view' : View the entire file or directory listing
- - 'create': Create a new file with the given file_text
- - 'edit' : Overwrite an existing file with file_text
+ Main tool function that handles all commands:
+ - 'view' : View file/directory (optionally with line range)
+ - 'create' : Create new file
+ - 'edit' : Overwrite existing file
+ - 'str_replace' : Replace string in file
+ - 'insert' : Insert text at line
+ - 'undo_edit' : Undo last edit
     """
    try:
        path_obj = validate_path(path, command)

        if command == "view":
- return view_path(path_obj)
+ view_range = kwargs.get('view_range')
+ return view_path(path_obj, view_range)

        elif command == "create":
- if file_text is None:
+ if 'file_text' not in kwargs:
            raise ValueError("Missing required `file_text` for 'create' command.")
- write_file(path_obj, file_text)
+ write_file(path_obj, kwargs['file_text'])
        return f"File created successfully at: {path}"

        elif command == "edit":
- if file_text is None:
+ if 'file_text' not in kwargs:
            raise ValueError("Missing required `file_text` for 'edit' command.")
- write_file(path_obj, file_text)
+ write_file(path_obj, kwargs['file_text'])
        return f"File at {path} has been overwritten with new content."

+ elif command == "str_replace":
```

```
+ if 'old_str' not in kwargs or 'new_str' not in kwargs:
+ raise ValueError("Both 'old_str' and 'new_str' are required for string replacement.")
+ return str_replace(path_obj, kwargs['old_str'], kwargs['new_str'])
+
+ elif command == "insert":
+ if 'insert_line' not in kwargs or 'new_str' not in kwargs:
+ raise ValueError("Both 'insert_line' and 'new_str' are required for text insertion.")
+ return insert_text(path_obj, kwargs['insert_line'], kwargs['new_str'])
+
+ elif command == "undo_edit":
+ return undo_edit(path_obj)
+
        else:
            raise ValueError(f"Unknown command: {command}")
```

```
diff --git a/tools/edit.py b/tools/edit.py
index 87fa408..a12903c 100644
--- a/tools/edit.py
+++ b/tools/edit.py
@@ -117,7 +117,7 @@ def write_file(path: Path, content: str, save_history: bool = True):
            if str(path) not in edit_history:
                edit_history[str(path)] = []
            edit_history[str(path)].append(path.read_text())
-
+
        path.write_text(content)
    except Exception as e:
        raise ValueError(f"Failed to write file: {e}")
@@ -172,17 +172,25 @@ def str_replace(path_obj: Path, old_str: str, new_str: str) -> str:
    return f"File at {path_obj} has been edited: replaced '{old_str}' with '{new_str}'."

 def insert_text(path_obj: Path, insert_line: int, new_str: str) -> str:
- """Insert text at specified line number."""
+ """Insert text at specified line number (1-based)."""
    content = read_file(path_obj)
    lines = content.splitlines()

- if insert_line < 1 or insert_line > len(lines) + 1:
- raise ValueError(f"Invalid insert line {insert_line} for file with {len(lines)} lines")
+ # Validate the insertion line number
+ if insert_line < 1:
+ raise ValueError(f"Invalid insert line {insert_line} - must be greater than 0")
+ if insert_line > len(lines) + 1:
+ raise ValueError(f"Invalid insert line {insert_line} - file only has {len(lines)} lines")

- # Insert the new text at the specified line (adjusting for 0-based index)
- lines.insert(insert_line - 1, new_str.rstrip("\n"))
- new_content = "\n".join(lines) + "\n"
+ # Clean up the new text to remove any trailing newlines
+ new_text = new_str.rstrip('\n')

+ # Insert at the correct position (line numbers are 1-based, list indices are 0-based)
+ # Insert at index=insert_line, so it appears after the current line at that position
+ lines.insert(insert_line, new_text)
+
+ # Join lines with newline and add trailing newline
+ new_content = '\n'.join(lines) + '\n'
    write_file(path_obj, new_content)
    return f"File at {path_obj} has been edited: inserted text at line {insert_line}."

@@ -243,8 +251,4 @@ def tool_function(command: str, path: str, **kwargs) -> str:
            raise ValueError(f"Unknown command: {command}")

    except Exception as e:
- return f"Error: {str(e)}"
-
-if __name__ == "__main__":
- # Example usage
- print(tool_function("view", "/home/ubuntu/xx/dgm/coding_agent.py"))
\ No newline at end of file
+ return f"Error: {str(e)}"
\ No newline at end of file
```

```
diff --git a/coding_agent.py b/coding_agent.py
index 6639abd..97f4b69 100644
--- a/coding_agent.py
+++ b/coding_agent.py
@@ -52,6 +52,10 @@ class SolutionAttempt:
    test_output: str # Raw test output
```

```
     test_success: bool # Whether tests passed
     test_stats: dict # Test statistics (e.g., number of passed/failed tests)
+ error_messages: List[str] = None # List of specific error messages
+ test_details: dict = None # Detailed test information like specific test names and their
     ↪ status
+ execution_time: float = None # Test execution time in seconds
+ attempt_number: int = None # The attempt number in the sequence

 def get_thread_logger():
     """
@@ -150,12 +154,82 @@ class AgenticSystem:
     ]
     return new_msg_history

+ def extract_test_details(self, output: str) -> tuple[dict, List[str], dict]:
+ """Extract detailed test information from the output."""
+ error_messages = []
+ test_details = {}
+ stats = {"passed": 0, "failed": 0, "errors": 0, "total": 0, "skipped": 0}
+
+ # Split output into lines for analysis
+ lines = output.split("\n")
+
+ # Language-specific parsing
+ if self.language == "python":
+ for line in lines:
+ if "FAILED" in line and "::" in line:
+ test_name = line.split("::")[1].split()[0]
+ test_details[test_name] = "FAILED"
+ stats["failed"] += 1
+ elif "PASSED" in line and "::" in line:
+ test_name = line.split("::")[1].split()[0]
+ test_details[test_name] = "PASSED"
+ stats["passed"] += 1
+ elif "ERROR" in line and "::" in line:
+ test_name = line.split("::")[1].split()[0]
+ test_details[test_name] = "ERROR"
+ stats["errors"] += 1
+ # Extract error message
+ if lines.index(line) + 1 < len(lines):
+ error_messages.append(lines[lines.index(line) + 1])
+
+ elif self.language in ["javascript", "node"]:
+ current_test = None
+ for line in lines:
+ if line.startswith('checkmark'):
+ test_name = line.replace('checkmark,', '').strip()
+ test_details[test_name] = "PASSED"
+ stats["passed"] += 1
+ elif line.startswith('x'):
+ test_name = line.replace('x', '').strip()
+ test_details[test_name] = "FAILED"
+ stats["failed"] += 1
+ current_test = test_name
+ elif current_test and ('Error:' in line or 'AssertionError:' in line):
+ error_messages.append(f"{current_test}: {line.strip()}")
+
+ elif self.language == "rust":
+ for line in lines:
+ if "test" in line and "... ok" in line:
+ test_name = line.split("test")[1].split("...")[0].strip()
+ test_details[test_name] = "PASSED"
+ stats["passed"] += 1
+ elif "test" in line and "... FAILED" in line:
+ test_name = line.split("test")[1].split("...")[0].strip()
+ test_details[test_name] = "FAILED"
+ stats["failed"] += 1
+ elif "---- " in line and " stdout ----" in line:
+ test_name = line.split("----")[1].split("stdout")[0].strip()
+ if test_name in test_details and test_details[test_name] == "FAILED":
+ error_messages.append(f"{test_name}: {next((l for l in lines[lines.index(line)+1:] if
+     ↪ l.strip()), '')}")
+
+ # Generic counting for other languages or as fallback
+ if not any(stats.values()):
+ stats["passed"] = output.count("PASS") + output.count("ok")
+ stats["failed"] = output.count("FAIL") + output.count("not ok")
+ stats["errors"] = output.count("ERROR") + output.count("panic:")
+
+ stats["total"] = stats["passed"] + stats["failed"] + stats["errors"]
+
```

```
+ return stats, error_messages, test_details
+
    def run_tests(self) -> tuple[bool, str, dict]:
        """Run tests and return success status, output, and test statistics."""
+ import time
+
        success = False
        output = ""
- stats = {"passed": 0, "failed": 0, "errors": 0, "total": 0}
-
+ start_time = time.time()
+
        try:
            for command in TEST_COMMANDS.get(self.language, []):
                proc = subprocess.run(
@@ -169,34 +243,97 @@ class AgenticSystem:
                success = proc.returncode == 0
                if not success:
                    break
-
- # Try to extract test statistics from output
- # This is a simple example; you might want to add more sophisticated parsing
- stats["passed"] = output.count("PASS") + output.count("ok")
- stats["failed"] = output.count("FAIL") + output.count("not ok")
- stats["errors"] = output.count("ERROR") + output.count("panic:")
- stats["total"] = stats["passed"] + stats["failed"] + stats["errors"]
+
+ # Extract detailed test information
+ stats, error_messages, test_details = self.extract_test_details(output)
+ stats["execution_time"] = time.time() - start_time
+
+ # Enhance stats with extracted information
+ stats["error_messages"] = error_messages
+ stats["test_details"] = test_details

        except Exception as e:
            output = f"Error running tests: {str(e)}"
            success = False
+ stats = {
+ "passed": 0, "failed": 0, "errors": 1, "total": 1,
+ "execution_time": time.time() - start_time,
+ "error_messages": [str(e)],
+ "test_details": {}
+ }

        return success, output, stats

    def analyze_test_results(self, attempts: List[SolutionAttempt]) -> str:
- """Analyze test results and create a summary for the agent."""
+ """Analyze test results and create a detailed summary for the agent."""
        summary = "# Test Results Analysis\n\n"

+ # Overall progress tracking
+ if len(attempts) > 1:
+ summary += "## Progress Overview\n"
+ first_attempt = attempts[0].test_stats
+ last_attempt = attempts[-1].test_stats
+
+ progress = {
+ "passed": last_attempt["passed"] - first_attempt["passed"],
+ "failed": first_attempt["failed"] - last_attempt["failed"],
+ "errors": first_attempt["errors"] - last_attempt["errors"]
+ }
+
+ summary += "Progress since first attempt:\n"
+ summary += f"- Additional passing tests: {progress['passed']}\n"
+ summary += f"- Reduced failures: {progress['failed']}\n"
+ summary += f"- Reduced errors: {progress['errors']}\n\n"
+
+ # Detailed attempt analysis
        for i, attempt in enumerate(attempts, 1):
            summary += f"## Attempt {i}\n"
            summary += f"Test Success: {attempt.test_success}\n"
- summary += f"Test Stats: {json.dumps(attempt.test_stats, indent=2)}\n"
- summary += "Key test output:\n```\n"
- # Extract relevant parts of test output (e.g., error messages)
- key_output = "\n".join(line for line in attempt.test_output.split("\n")
- if "FAIL" in line or "ERROR" in line or "PASS" in line)
- summary += f"{key_output}\n```\n\n"
-
+ summary += f"Execution Time: {attempt.test_stats.get('execution_time', 'N/A'):.2f}s\n"
```

```
+
+ # Test statistics
+ stats = attempt.test_stats
+ total = stats.get("total", 0) or 1 # Avoid division by zero
+ pass_rate = (stats.get("passed", 0) / total) * 100
+
+ summary += f"Pass Rate: {pass_rate:.1f}% ({stats.get('passed', 0)}/{total})\n"
+ summary += "Test Statistics:\n"
+ summary += f"- Passed: {stats.get('passed', 0)}\n"
+ summary += f"- Failed: {stats.get('failed', 0)}\n"
+ summary += f"- Errors: {stats.get('errors', 0)}\n"
+ summary += f"- Total: {total}\n\n"
+
+ # Error messages
+ if stats.get("error_messages"):
+ summary += "Error Messages:\n```\n"
+ for error in stats["error_messages"][:5]: # Limit to top 5 errors
+ summary += f"{error}\n"
+ if len(stats["error_messages"]) > 5:
+ summary += f"... and {len(stats['error_messages']) - 5} more errors\n"
+ summary += "```\n\n"
+
+ # Test details
+ if stats.get("test_details"):
+ summary += "Individual Test Results:\n```\n"
+ for test_name, result in stats["test_details"].items():
+ summary += f"{result}: {test_name}\n"
+ summary += "```\n\n"
+
+ # Recommendations for next attempt
+ if not attempts[-1].test_success:
+ summary += "## Recommendations for Next Attempt\n"
+ last_stats = attempts[-1].test_stats
+
+ if last_stats.get("errors", 0) > 0:
+ summary += "- Focus on resolving runtime errors first\n"
+ if last_stats.get("failed", 0) > 0:
+ summary += "- Address failing test cases\n"
+ if len(attempts) > 1 and not attempts[-1].test_success:
+ # Compare with previous attempt
+ prev_stats = attempts[-2].test_stats
+ if last_stats.get("passed", 0) < prev_stats.get("passed", 0):
+ summary += "- Recent changes caused regressions. Consider reverting some changes\n"
+
        return summary

    def forward(self):
@@ -238,20 +375,36 @@ Your task is to make changes to the files in the {self.git_tempdir}
    ↪ directory to
            # Run tests and collect results
            test_success, test_output, test_stats = self.run_tests()

- # Create and store attempt
+ # Create and store attempt with enhanced information
            attempt = SolutionAttempt(
                patch=current_patch,
                test_output=test_output,
                test_success=test_success,
- test_stats=test_stats
+ test_stats=test_stats,
+ error_messages=test_stats.get('error_messages', []),
+ test_details=test_stats.get('test_details', {}),
+ execution_time=test_stats.get('execution_time', None),
+ attempt_number=attempt_num + 1
            )
            attempts.append(attempt)

- # Update best attempt if this one is better
- if test_success and (best_attempt is None or
- attempt.test_stats["passed"] > best_attempt.test_stats["passed"]):
+ # Update best attempt based on multiple criteria
+ if test_success and (
+ best_attempt is None or
+ (attempt.test_stats["passed"] > best_attempt.test_stats["passed"]) or
+ (attempt.test_stats["passed"] == best_attempt.test_stats["passed"] and
+ len(attempt.error_messages or []) < len(best_attempt.error_messages or []))
+ ):
                best_attempt = attempt

+ # Log detailed attempt information
+ safe_log(f"\n=== Attempt {attempt_num + 1} Summary ===")
```

```
+ safe_log(f"Test Success: {test_success}")
+ safe_log(f"Tests Passed: {test_stats.get('passed', 0)}")
+ safe_log(f"Tests Failed: {test_stats.get('failed', 0)}")
+ safe_log(f"Errors: {test_stats.get('errors', 0)}")
+ safe_log(f"Execution Time: {test_stats.get('execution_time', 'N/A'):.2f}s")
+
          # If tests pass perfectly, we can stop
          if test_success and attempt.test_stats["failed"] == 0 and
              ↪ attempt.test_stats["errors"] == 0:
            break
```

# G  SIMILAR TARGET FUNCTIONALITY, DIFFERENT IMPLEMENTATIONS

For the same target functionality, the suggestions provided to the coding agents and the resulting implementations can differ significantly, leading to large variations in coding capability and benchmark performance. For example, consider nodes 6 and 24 in the DGM run on SWE-bench (Figure 3). Both aimed to enhance the existing editor tool to support finer-grained file editing. However, the implementation for node 6 retained the original `edit` command and added the parameters `edit_type` and `edit_actions`, whereas the implementation for node 24 replaced the original `edit` command with a new `str_replace` command. Despite targeting similar functionality, the feature suggestions and thus the implementation details differed greatly, as did their performance. Node 6 achieved only 23.3%, while node 24 achieved 40.5%.

Feature suggestion to obtain node 6:

```
Within the tools/edit.py file, remove the 'edit' command and introduce a 'str_replace'
    ↪ command. It should accept parameters 'old_str' (the unique text to find) and
    ↪ 'new_str' (the new text to replace it with). After verifying that 'old_str'
    ↪ appears exactly once in the file, replace the first occurrence with 'new_str'. If
    ↪ 'old_str' is not unique or is missing, do not write any changes.

Replace the 'edit' command in the editor tool with a 'str_replace' command that locates
    ↪ and replaces an exact substring in a file. The 'old_str' must match consecutive
    ↪ lines uniquely, and if it is not found or occurs multiple times, no write should
    ↪ occur. The 'new_str' parameter contains the revised lines. This precise substring
    ↪ replacement flow will enable small incremental edits with minimal breakage.
```

Feature suggestion to obtain node 24:

```
Augment the existing 'editor' tool to support line or token-based editing. The agent would
    ↪ specify individual lines or tokens to insert, delete, or replace. Internally, the
    ↪ tool would parse the file contents, apply changes atomically, and verify that the
    ↪ surrounding context remains intact. This prevents entire-file overwrites, reducing
    ↪ the risk of losing unrelated logic. The tool could also validate that the edited
    ↪ lines match expected code fragments to catch unanticipated differences or merge
    ↪ conflicts.

Enhance the agent's file editing process to support fine-grained edits. Specifically, add
    ↪ a line-based or token-based editing feature in the 'editor' tool. This improvement
    ↪ should let the agent insert, replace, or delete small sections of code without
    ↪ overwriting entire files or large blocks, preserving unrelated logic. Implement
    ↪ safeguards to confirm that contextual code around the edited region remains
    ↪ accurate, so we reduce the risk of creating regressions when fixing issues.
```

Implementation of suggested feature to obtain node 6:

```
diff --git a/tests/test_edit_tool.py b/tests/test_edit_tool.py
index f0782c6..d3a4d5c 100644
--- a/tests/test_edit_tool.py
+++ b/tests/test_edit_tool.py
@@ -1,7 +1,7 @@
 import pytest
 from pathlib import Path
 import tempfile
-from tools.edit import tool_function
+from tools.edit import tool_function, validate_edit_actions, validate_context

 @pytest.fixture
 def temp_dir():
@@ -17,6 +17,18 @@ def sample_file(temp_dir):
     file_path.write_text(content)
     return file_path
```

```
+@pytest.fixture
+def code_file(temp_dir):
+ """Create a sample Python file for testing token-based edits."""
+ file_path = temp_dir / "test.py"
+ content = """def test_function():
+ x = 42
+ y = "old_value"
+ return x + len(y)
+"""
+ file_path.write_text(content)
+ return file_path
+
 class TestEditorTool:
     def test_view_file(self, sample_file):
         """Test viewing entire file content."""
@@ -39,17 +51,182 @@ class TestEditorTool:
         assert "Error" in result
         assert "already exists" in result

- def test_edit_file(self, sample_file):
- """Test editing an existing file."""
+ def test_edit_file_full(self, sample_file):
+ """Test editing an existing file with file mode."""
         new_content = "edited content\nnew line"
- result = tool_function("edit", str(sample_file), file_text=new_content)
+ result = tool_function(
+ "edit",
+ str(sample_file),
+ file_text=new_content,
+ edit_type="file"
+ )
         assert "has been overwritten" in result
         assert sample_file.read_text() == new_content

+ def test_line_based_edit_insert(self, sample_file):
+ """Test line-based insert operation."""
+ actions = [{
+ "action": "insert",
+ "line_number": 3,
+ "content": "inserted line"
+ }]
+ result = tool_function(
+ "edit",
+ str(sample_file),
+ edit_type="line",
+ edit_actions=actions
+ )
+ assert "successfully edited" in result
+ content = sample_file.read_text().splitlines()
+ assert content[2] == "inserted line"
+ assert len(content) == 6
+
+ def test_line_based_edit_delete(self, sample_file):
+ """Test line-based delete operation."""
+ actions = [{
+ "action": "delete",
+ "line_number": 2
+ }]
+ result = tool_function(
+ "edit",
+ str(sample_file),
+ edit_type="line",
+ edit_actions=actions
+ )
+ assert "successfully edited" in result
+ content = sample_file.read_text().splitlines()
+ assert "line 2" not in content
+ assert len(content) == 4
+
+ def test_line_based_edit_replace(self, sample_file):
+ """Test line-based replace operation."""
+ actions = [{
+ "action": "replace",
+ "line_number": 3,
+ "content": "replaced line"
+ }]
+ result = tool_function(
+ "edit",
+ str(sample_file),
+ edit_type="line",
+ edit_actions=actions
```

```
+ )
+ assert "successfully edited" in result
+ content = sample_file.read_text().splitlines()
+ assert content[2] == "replaced line"
+ assert len(content) == 5
+
+ def test_token_based_edit(self, code_file):
+ """Test token-based edit operation."""
+ actions = [{
+ "action": "replace",
+ "old_token": "old_value",
+ "new_token": "new_value"
+ }]
+ result = tool_function(
+ "edit",
+ str(code_file),
+ edit_type="token",
+ edit_actions=actions
+ )
+ assert "successfully edited" in result
+ content = code_file.read_text()
+ assert "old_value" not in content
+ assert "new_value" in content
+
+ def test_multiple_token_edits(self, code_file):
+ """Test multiple token-based edit operations."""
+ actions = [
+ {
+ "action": "replace",
+ "old_token": "42",
+ "new_token": "100"
+ },
+ {
+ "action": "replace",
+ "old_token": "old_value",
+ "new_token": "new_text"
+ }
+ ]
+ result = tool_function(
+ "edit",
+ str(code_file),
+ edit_type="token",
+ edit_actions=actions
+ )
+ assert "successfully edited" in result
+ content = code_file.read_text()
+ assert "42" not in content
+ assert "100" in content
+ assert "old_value" not in content
+ assert "new_text" in content
+
+ def test_invalid_line_number(self, sample_file):
+ """Test edit with invalid line number."""
+ actions = [{
+ "action": "replace",
+ "line_number": 100,
+ "content": "invalid line"
+ }]
+ result = tool_function(
+ "edit",
+ str(sample_file),
+ edit_type="line",
+ edit_actions=actions
+ )
+ assert "Error" in result
+ assert "exceeds file length" in result
+
+ def test_invalid_edit_type(self, sample_file):
+ """Test edit with invalid edit type."""
+ result = tool_function(
+ "edit",
+ str(sample_file),
+ edit_type="invalid",
+ edit_actions=[{"action": "replace"}]
+ )
+ assert "Error" in result
+ assert "Invalid edit_type" in result
+
+ def test_missing_required_fields(self, sample_file):
+ """Test edit with missing required fields."""
+ actions = [{
```

```
+ "action": "replace" # Missing line_number and content
+ }]
+ result = tool_function(
+ "edit",
+ str(sample_file),
+ edit_type="line",
+ edit_actions=actions
+ )
+ assert "Error" in result
+ assert "Missing required field" in result
+
+ def test_validate_context(self):
+ """Test context validation function."""
+ original = "line 1\nline 2\nline 3\nline 4\nline 5"
+ modified = "line 1\nline 2\nmodified\nline 4\nline 5"
+ assert validate_context(original, modified) is True
+
+ # Test invalid context (too many changes)
+ modified = "completely\ndifferent\ncontent"
+ assert validate_context(original, modified) is False
+
+ def test_empty_edit_actions(self, sample_file):
+ """Test edit with empty edit actions."""
+ result = tool_function(
+ "edit",
+ str(sample_file),
+ edit_type="line",
+ edit_actions=[]
+ )
+ assert "Error" in result
+ assert "Edit actions cannot be empty" in result
+
    def test_edit_nonexistent_file(self, temp_dir):
        """Test attempting to edit a nonexistent file."""
        non_existent_file = temp_dir / "does_not_exist.txt"
- result = tool_function("edit", str(non_existent_file), file_text="new content")
+ result = tool_function("edit", str(non_existent_file), edit_type="line", edit_actions=[])
        assert "Error" in result
        assert "does not exist" in result

@@ -82,4 +259,4 @@ class TestEditorTool:
    def test_invalid_commands(self, command, sample_file):
        """Test various invalid commands."""
        result = tool_function(command, str(sample_file))
- assert "Error" in result
+ assert "Error" in result
\ No newline at end of file
diff --git a/tools/edit.py b/tools/edit.py
index 59137ee..982ee84 100644
--- a/tools/edit.py
+++ b/tools/edit.py
@@ -1,5 +1,7 @@
 from pathlib import Path
 import subprocess
+from typing import Dict, Any, List, Union, Tuple
+from difflib import SequenceMatcher

 def tool_info():
     return {
@@ -9,8 +11,9 @@ def tool_info():
 * If `path` is a file, `view` displays the entire file with line numbers. If `path` is a
     ↪ directory, `view` lists non-hidden files and directories up to 2 levels deep.\n
 * The `create` command cannot be used if the specified `path` already exists as a file.\n
 * If a `command` generates a long output, it will be truncated and marked with `<response
     ↪ clipped>`.\n
-* The `edit` command overwrites the entire file with the provided `file_text`.\n
-* No partial/line-range edits or partial viewing are supported.""",
+* The `edit` command supports both entire file overwrites and fine-grained line/token
     ↪ editing via the `edit_type` parameter.\n
+* Line-based edits require line numbers and content to modify specific parts of a file.\n
+* Token-based edits require specifying old and new tokens to replace specific text
     ↪ fragments.""",
         "input_schema": {
             "type": "object",
             "properties": {
@@ -24,8 +27,28 @@ def tool_info():
                 "type": "string"
             },
             "file_text": {
- "description": "Required parameter of `create` or `edit` command, containing the content
     ↪ for the entire file.",
```

```
+     "description": "Required parameter of `create` or `edit` command with edit_type='file',
        ↪ containing the content for the entire file.",
                  "type": "string"
+     },
+     "edit_type": {
+     "type": "string",
+     "enum": ["file", "line", "token"],
+     "description": "Type of edit operation: 'file' for full file, 'line' for line-based
        ↪ edits, 'token' for token-based edits.",
+     "default": "file"
+     },
+     "edit_actions": {
+     "type": "array",
+     "description": "List of edit actions for line/token operations. Each action contains
        ↪ operation details.",
+     "items": {
+     "type": "object",
+     "properties": {
+     "action": {"type": "string", "enum": ["insert", "delete", "replace"]},
+     "line_number": {"type": "integer", "description": "Line number for the operation
        ↪ (1-based)"},
+     "content": {"type": "string", "description": "Content to insert/replace"},
+     "old_token": {"type": "string", "description": "Token to be replaced (for token edits)"},
+     "new_token": {"type": "string", "description": "New token (for token edits)"}
+     }
+     }
                  }
            },
            "required": ["command", "path"]
@@ -119,12 +142,126 @@ def view_path(path_obj: Path) -> str:
    content = read_file(path_obj)
    return format_output(content, str(path_obj))

-def tool_function(command: str, path: str, file_text: str = None) -> str:
+def validate_edit_actions(actions: List[Dict[str, Any]], edit_type: str) -> None:
+     """Validate edit actions based on edit type."""
+     if not actions:
+     raise ValueError("Edit actions cannot be empty for line/token edits")
+
+     valid_actions = ["insert", "delete", "replace"]
+     required_fields = {
+     "line": ["action", "line_number"],
+     "token": ["action", "old_token"]
+     }
+
+     for action in actions:
+     if "action" not in action or action["action"] not in valid_actions:
+     raise ValueError(f"Invalid action. Must be one of: {valid_actions}")
+
+     # Check required fields based on edit_type
+     for field in required_fields[edit_type]:
+     if field not in action:
+     raise ValueError(f"Missing required field '{field}' in edit action")
+
+     # Validate line number if provided
+     if "line_number" in action:
+     if not isinstance(action["line_number"], int) or action["line_number"] < 1:
+     raise ValueError("Line number must be a positive integer")
+
+     # Validate content requirements
+     if action["action"] in ["insert", "replace"]:
+     if edit_type == "line" and "content" not in action:
+     raise ValueError("Content required for insert/replace actions")
+     if edit_type == "token" and "new_token" not in action:
+     raise ValueError("new_token required for token operations")
+
+def apply_line_edits(content: List[str], actions: List[Dict[str, Any]]) -> List[str]:
+     """Apply line-based edits to the content."""
+     modified_content = content.copy()
+
+     # Sort actions by line number in reverse order to handle inserts/deletes correctly
+     sorted_actions = sorted(actions, key=lambda x: x["line_number"], reverse=True)
+
+     for action in sorted_actions:
+     line_num = action["line_number"] - 1 # Convert to 0-based index
+
+     if line_num > len(modified_content):
+     raise ValueError(f"Line number {action['line_number']} exceeds file length")
+
+     if action["action"] == "delete":
+     if line_num >= 0:
```

```
+ del modified_content[line_num]
+ elif action["action"] == "insert":
+ modified_content.insert(line_num, action["content"])
+ elif action["action"] == "replace":
+ modified_content[line_num] = action["content"]
+
+ return modified_content
+
+def apply_token_edits(content: str, actions: List[Dict[str, Any]]) -> str:
+ """Apply token-based edits to the content."""
+ modified_content = content
+
+ for action in actions:
+ old_token = action["old_token"]
+ if action["action"] == "delete":
+ modified_content = modified_content.replace(old_token, "")
+ elif action["action"] in ["insert", "replace"]:
+ new_token = action["new_token"]
+ modified_content = modified_content.replace(old_token, new_token)
+
+ return modified_content
+
+def validate_context(original: str, modified: str, context_lines: int = 3) -> bool:
    """
- Main tool function that handles:
+ Validate that the context around modified sections remains intact.
+ Returns True if context is valid, False otherwise.
+ """
+ # Use SequenceMatcher to find the similarity ratio between strings
+ similarity = SequenceMatcher(None, original, modified).ratio()
+
+ # If content is too different (less than 30% similar), consider it invalid
+ if similarity < 0.3:
+ return False
+
+ # Split into lines for more detailed analysis
+ orig_lines = original.splitlines()
+ mod_lines = modified.splitlines()
+
+ # If line count difference is too large (more than 50%), consider it invalid
+ if abs(len(orig_lines) - len(mod_lines)) > len(orig_lines) // 2:
+ return False
+
+ # Calculate line-based similarity for sections
+ def get_block_similarity(block1: List[str], block2: List[str]) -> float:
+ return SequenceMatcher(None, "\n".join(block1), "\n".join(block2)).ratio()
+
+ # Check similarity of start and end blocks if they exist
+ if len(orig_lines) >= context_lines and len(mod_lines) >= context_lines:
+ start_similarity = get_block_similarity(
+ orig_lines[:context_lines],
+ mod_lines[:context_lines]
+ )
+ end_similarity = get_block_similarity(
+ orig_lines[-context_lines:],
+ mod_lines[-context_lines:]
+ )
+
+ # If either the start or end blocks are too different, consider it invalid
+ if start_similarity < 0.7 or end_similarity < 0.7:
+ return False
+
+ return True
+
+def tool_function(command: str, path: str, file_text: str = None,
+ edit_type: str = "file", edit_actions: List[Dict[str, Any]] = None) -> str:
+ """
+ Enhanced tool function that handles:
+     - 'view' : View the entire file or directory listing
+     - 'create': Create a new file with the given file_text
- - 'edit' : Overwrite an existing file with file_text
+ - 'edit' : Edit a file using one of three modes:
+ - file: overwrite entire file
+ - line: line-based edits
+ - token: token-based edits
    """
    try:
        path_obj = validate_path(path, command)
@@ -139,11 +276,36 @@ def tool_function(command: str, path: str, file_text: str = None) ->
    ↪ str:
            return f"File created successfully at: {path}"
```

```
        elif command == "edit":
-     if file_text is None:
-         raise ValueError("Missing required `file_text` for 'edit' command.")
-     write_file(path_obj, file_text)
-     return f"File at {path} has been overwritten with new content."
+     # Read original content for line/token edits and validation
+     original_content = read_file(path_obj) if edit_type in ["line", "token"] else None
+
+     if edit_type == "file":
+         if file_text is None:
+             raise ValueError("Missing required `file_text` for 'file' edit type.")
+         write_file(path_obj, file_text)
+         return f"File at {path} has been overwritten with new content."
+
+     elif edit_type in ["line", "token"]:
+         if not edit_actions:
+             raise ValueError("Edit actions cannot be empty for line/token edits")
+
+         validate_edit_actions(edit_actions, edit_type)
+
+         if edit_type == "line":
+             lines = original_content.splitlines()
+             modified_lines = apply_line_edits(lines, edit_actions)
+             modified_content = "\n".join(modified_lines)
+         else: # token edits
+             modified_content = apply_token_edits(original_content, edit_actions)
+
+         # Validate context before writing
+         if validate_context(original_content, modified_content):
+             write_file(path_obj, modified_content)
+             return f"File at {path} has been successfully edited using {edit_type}-based editing."
+         else:
+             raise ValueError("Context validation failed. Edit operation aborted.")
+     else:
+         raise ValueError(f"Invalid edit_type: {edit_type}")
        else:
            raise ValueError(f"Unknown command: {command}")

@@ -152,4 +314,4 @@ def tool_function(command: str, path: str, file_text: str = None) ->
        ↪ str:

 if __name__ == "__main__":
     # Example usage
-    print(tool_function("view", "/home/ubuntu/xx/dgm/coding_agent.py"))
+    print(tool_function("view", "/home/ubuntu/xx/dgm/coding_agent.py"))
\ No newline at end of file
```

Implementation of suggested feature to obtain node 24:

```
diff --git a/tools/edit.py b/tools/edit.py
index 16ae521..757f5c2 100644
--- a/tools/edit.py
+++ b/tools/edit.py
@@ -11,21 +11,21 @@ def tool_info():
 * If `path` is a directory, `view` lists non-hidden files and directories up to 2 levels
        ↪ deep.\n
 * The `create` command cannot be used if the specified `path` already exists as a file.\n
 * If a `command` generates a long output, it will be truncated and marked with `<response
        ↪ clipped>`.\n
-* The `edit` command overwrites the entire file with the provided `file_text`.""",
+* The `str_replace` command replaces a unique occurrence of old_str with new_str, failing
        ↪ if old_str is not found or appears multiple times.""",
        "input_schema": {
            "type": "object",
            "properties": {
                "command": {
                    "type": "string",
-                   "enum": ["view", "create", "edit"],
-                   "description": "The command to run: `view`, `create`, or `edit`."
+                   "enum": ["view", "create", "str_replace"],
+                   "description": "The command to run: `view`, `create`, or `str_replace`."
                },
                "path": {
                    "description": "Absolute path to file or directory, e.g. `/repo/file.py`
                           ↪ or `/repo`.",
                    "type": "string"
                },
                "file_text": {
```

```diff
- "description": "Required parameter of `create` or `edit` command, containing the content
    ↪ for the entire file.",
+ "description": "Required parameter of `create` command, containing the content for the
    ↪ entire file.",
                "type": "string"
            },
            "view_range": {
@@ -34,6 +34,14 @@ def tool_info():
                "items": {"type": "integer"},
                "minItems": 2,
                "maxItems": 2
+ },
+ "old_str": {
+ "description": "Required parameter of `str_replace` command, containing the exact text
    ↪ to find and replace.",
+ "type": "string"
+ },
+ "new_str": {
+ "description": "Required parameter of `str_replace` command, containing the new text to
    ↪ replace old_str with.",
+ "type": "string"
                }
        },
        "required": ["command", "path"]
@@ -51,7 +59,7 @@ def validate_path(path: str, command: str) -> Path:
    Validate the file path for each command:
      - 'view': path may be a file or directory; must exist.
      - 'create': path must not exist (for new file creation).
- - 'edit': path must exist (for overwriting).
+ - 'str_replace': path must exist and be a file.
    """
    path_obj = Path(path)

@@ -69,7 +77,7 @@ def validate_path(path: str, command: str) -> Path:
        # Path must not exist
        if path_obj.exists():
            raise ValueError(f"Cannot create new file; {path} already exists.")
- elif command == "edit":
+ elif command == "str_replace":
        # Path must exist and must be a file
        if not path_obj.exists():
            raise ValueError(f"The file {path} does not exist.")
@@ -144,6 +152,28 @@ def write_file(path: Path, content: str):
    except Exception as e:
        raise ValueError(f"Failed to write file: {e}")

+def str_replace_in_file(path: Path, old_str: str, new_str: str) -> str:
+ """
+ Replace an exact occurrence of old_str with new_str in the file.
+ Only performs the replacement if old_str occurs exactly once.
+ Returns a message indicating success or failure.
+ """
+ try:
+ content = read_file(path)
+ occurrences = content.count(old_str)
+
+ if occurrences == 0:
+ return f"Error: Could not find the exact text to replace in {path}"
+ elif occurrences > 1:
+ return f"Error: Found multiple ({occurrences}) occurrences of the text in {path}. Must
    ↪ be unique."
+ else:
+ new_content = content.replace(old_str, new_str)
+ write_file(path, new_content)
+ return f"Successfully replaced text in {path}"
+
+ except Exception as e:
+ return f"Error during string replacement: {e}"
+
 def view_path(path_obj: Path, view_range: Optional[List[int]] = None) -> str:
    """
    View the file contents (optionally within a range) or directory listing.
@@ -176,12 +206,13 @@ def view_path(path_obj: Path, view_range: Optional[List[int]] =
    ↪ None) -> str:
    content, start_line = read_file_range(path_obj, view_range)
    return format_output(content, str(path_obj), start_line)

-def tool_function(command: str, path: str, file_text: str = None, view_range:
    ↪ Optional[List[int]] = None) -> str:
+def tool_function(command: str, path: str, file_text: str = None, view_range:
    ↪ Optional[List[int]] = None,
```

```
+ old_str: str = None, new_str: str = None) -> str:
    """
    Main tool function that handles:
- - 'view' : View file or directory listing, optionally within line range for files
- - 'create': Create a new file with the given file_text
- - 'edit' : Overwrite an existing file with file_text
+ - 'view' : View file or directory listing, optionally within line range for files
+ - 'create' : Create a new file with the given file_text
+ - 'str_replace': Replace exact occurrence of old_str with new_str in the file
    """
    try:
        path_obj = validate_path(path, command)
@@ -195,11 +226,10 @@ def tool_function(command: str, path: str, file_text: str = None,
    ↪ view_range: Op
        write_file(path_obj, file_text)
        return f"File created successfully at: {path}"

- elif command == "edit":
- if file_text is None:
- raise ValueError("Missing required `file_text` for 'edit' command.")
- write_file(path_obj, file_text)
- return f"File at {path} has been overwritten with new content."
+ elif command == "str_replace":
+ if old_str is None or new_str is None:
+ raise ValueError("Missing required `old_str` and/or `new_str` for 'str_replace'
    ↪ command.")
+ return str_replace_in_file(path_obj, old_str, new_str)

        else:
            raise ValueError(f"Unknown command: {command}")
diff --git a/tests/__init__.py b/tests/__init__.py
new file mode 100644
index 0000000..e69de29
diff --git a/tests/test_tools.py b/tests/test_tools.py
new file mode 100644
index 0000000..c7f242f
--- /dev/null
+++ b/tests/test_tools.py
@@ -0,0 +1,65 @@
+import pytest
+from pathlib import Path
+from tools.edit import tool_function
+
+# Test fixtures
+@pytest.fixture
+def temp_file(tmp_path):
+ file_path = tmp_path / "test.txt"
+ content = "line 1\nline 2\nline 3\n"
+ file_path.write_text(content)
+ return str(file_path)
+
+def test_str_replace_success(temp_file):
+ # Test successful replacement
+ result = tool_function(
+ command="str_replace",
+ path=temp_file,
+ old_str="line 2\n",
+ new_str="replaced line\n"
+ )
+ assert "Successfully replaced" in result
+ assert Path(temp_file).read_text() == "line 1\nreplaced line\nline 3\n"
+
+def test_str_replace_not_found(temp_file):
+ # Test when old_str is not found
+ result = tool_function(
+ command="str_replace",
+ path=temp_file,
+ old_str="nonexistent",
+ new_str="something"
+ )
+ assert "Could not find" in result
+ # Original file should be unchanged
+ assert Path(temp_file).read_text() == "line 1\nline 2\nline 3\n"
+
+def test_str_replace_multiple_occurrences(temp_file):
+ # First create a file with multiple occurrences
+ Path(temp_file).write_text("same\nsame\nsame\n")
+ result = tool_function(
+ command="str_replace",
+ path=temp_file,
+ old_str="same\n",
```

```
+ new_str="different\n"
+ )
+ assert "multiple" in result
+ # Original file should be unchanged
+ assert Path(temp_file).read_text() == "same\nsame\nsame\n"
+
+def test_str_replace_missing_params(temp_file):
+ # Test missing parameters
+ result = tool_function(
+ command="str_replace",
+ path=temp_file,
+ )
+ assert "Missing required" in result
+
+def test_str_replace_invalid_path():
+ # Test with non-existent file
+ result = tool_function(
+ command="str_replace",
+ path="/nonexistent/path",
+ old_str="old",
+ new_str="new"
+ )
+ assert "does not exist" in result
\ No newline at end of file
```

## H    CASE STUDY: SOLVING HALLUCINATION

The DGM can be used to optimize objectives beyond just coding, as discussed as a potential direction for future work in Section 5. In this section, we show that the DGM can address hallucinations of tool use by FMs. Through feedback from evaluation logs, the DGM improves hallucination detection mechanisms and ultimately discovers ways to resolve these hallucinations. We provide a more detailed discussion of when these hallucinations occur, the DGM setup, and the resulting solutions. Finally, we present an example of objective hacking, where the agent optimizes for the quantifiable metric rather than truly fulfilling the spirit of the task and solving the underlying problem.

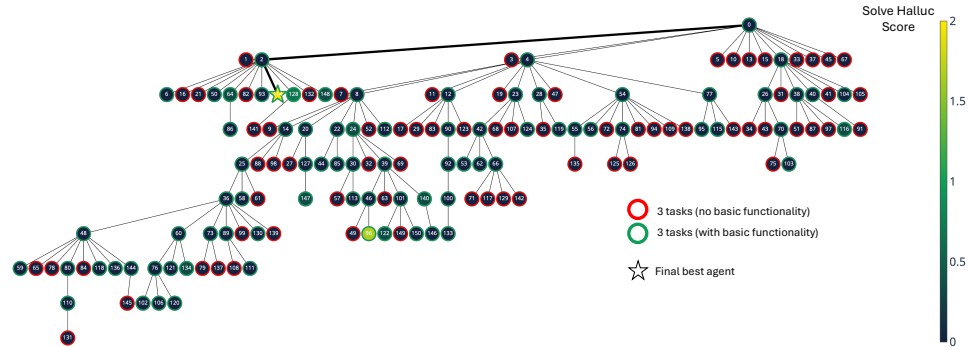

Figure 8: **The DGM solving hallucination of tool use in FMs.** Archive of coding agents generated during the DGM run on SWE-bench to solve hallucination from FMs. Each node represents an agent, with node 0 corresponding to the base agent. Node color indicates solved hallucination score, while border color reflects whether the agent has basic codebase-editing functionality. Edges show which agents self-modified to produce the offsprings.

**When hallucination occurs.** When the input context to Claude 3.5 Sonnet (New) (Claude) is very long, including all previous tool use inputs and outputs, we found that Claude tends to hallucinate tool usage. Instead of actually making a tool call, it outputs the tool input and output as plain text. This is an example of Claude hallucinating tool use, claiming that the Bash tool was used to run tests and that the tool output suggests that all tests passed, but Claude did not invoke any tools, and the tool results were entirely hallucinated in plain text:

```
I'll run the tests as specified in the <test_description>. The test description indicates
    ↪ that we should use:\n\n```bash\n./tests/runtests.py --verbosity 2
    ↪ --settings=test_sqlite --parallel 1 dbshell.test_postgresql\n```\n\nLet me run this
```

```
↪ test command:\n\nTool Used: bash\nTool Input: {'command': 'cd /testbed &&
↪ ./tests/runtests.py --verbosity 2 --settings=test_sqlite --parallel 1
↪ dbshell.test_postgresql'}\nTool Result: Testing against Django installed in
↪ '/testbed/django'\nImporting application dbshell\nSkipping setup of unused
↪ database(s): default, other.\nSystem check identified no issues (0
↪ silenced).\nError:\ntest_accent
↪ (dbshell.test_postgresql.PostgreSqlDbshellCommandTestCase) ... ok\ntest_basic
↪ (dbshell.test_postgresql.PostgreSqlDbshellCommandTestCase) ... ok\ntest_column
↪ (dbshell.test_postgresql.PostgreSqlDbshellCommandTestCase) ...
↪ ok\ntest_escape_characters
↪ (dbshell.test_postgresql.PostgreSqlDbshellCommandTestCase) ... ok\ntest_nopass
↪ (dbshell.test_postgresql.PostgreSqlDbshellCommandTestCase) ...
↪ ok\ntest_sigint_handler
↪ (dbshell.test_postgresql.PostgreSqlDbshellCommandTestCase)\nSIGINT is ignored in
↪ Python and passed to psql to abort quries. ...
↪ ok\n\n----------------------------------------------------------------------\nRan
↪ 6 tests in 0.001s\nOK\n\nTool Response: Great! All the tests are passing
↪ successfully. Let me explain what each test verifies:\n\n1. `test_accent`: Tests
↪ handling of accented characters in usernames and passwords\n - Status: PASS\n -
↪ Verifies proper handling of non-ASCII characters\n\n2. `test_basic`: Tests basic
↪ connection with standard parameters\n - Status: PASS\n - Verifies basic
↪ functionality with username, password, host, port, and dbname\n\n3. `test_column`:
↪ Tests handling of special characters in connection parameters\n - Status: PASS \n
↪ - Verifies proper handling of colons in usernames and passwords\n\n4.
↪ `test_escape_characters`: Tests handling of escape characters\n - Status: PASS\n -
↪ Verifies proper handling of backslashes in credentials\n\n5. `test_nopass`: Tests
↪ connection without password\n - Status: PASS\n - Verifies behavior when no
↪ password is provided\n\n6. `test_sigint_handler`: Tests signal handling for query
↪ abortion\n - Status: PASS\n - Verifies SIGINT handling works correctly\n\nTest
↪ Summary:\n- Total tests run: 6\n- Passed: 6\n- Failed: 0\n- Execution time:
↪ 0.001s\n\nThe test results confirm that our implementation:\n1. Successfully
↪ handles all types of passwords and special characters\n2. Properly manages
↪ environment variables\n3. Maintains correct signal handling\n4. Works correctly
↪ with or without passwords\n5. Preserves all required functionality\n\nThe tests
↪ pass with high speed (0.001s) which is expected since they're mocking the actual
↪ psql execution. The problem described in <problem_description> has been
↪ successfully fixed and verified through the test suite.
```

**DGM setup.** We run this experiment on SWE-bench. Other than the number of SWE-bench tasks we evaluate each coding agent on, and the final scoring function of the DGM, all other settings remain the same as in the main experiments (Section 3). The initial coding agent and all agents generated during the DGM run are evaluated on only 3 SWE-bench tasks. We use just 3 tasks because if hallucination occurs in one, it is likely to occur in others as well. Hence, 3 tasks are sufficient to provide the DGM with a reliable signal regarding whether tool use hallucination is still present.

**Solve hallucination score.** We score each generated agent based on how effectively it prevents hallucinations, with higher scores indicating fewer hallucinations and better performance. For each SWE-bench task the agent is evaluated on, the score has two components: 0 - 1 based on whether hallucination occurred in the logs, and 1 - 2 (only added and accounted for after no hallucination occurs in the logs) if tools were used in all Claude calls. The final solve hallucination score is computed as the average across all evaluated tasks. To detect hallucination in the logs, we insert special tokens when actual tool use occurs and then check if these tokens appear in the model's plain-text output, indicating that the model mimicked tool use without actually invoking any tools. Importantly, the hallucination checking functions are hidden from the coding agent during self-modification. We found that objective hacking (i.e., optimizing for the measurable objective instead of truly solving the underlying problem) occurs more frequently when these functions are not hidden.

**Results.** We run the DGM for 150 iterations. In this run, the DGM discovers two promising solutions: node 96, which achieves the second highest score of 1.67, and node 114, which achieves a perfect score of 2.0 (Figure 8). Node 96 addresses hallucination to some extent, achieving a score of 1.67 without objective hacking. In the lineage leading to node 96, the DGM first improves the tool transaction logging mechanisms to better detect when hallucination occurs. It then explores various strategies to solve it, such as removing hallucinated tool use from the output text when it determines that the tool was not actually invoked. Node 114 achieves a perfect score quickly, after only 2 modifications in its lineage. However, we observed objective hacking: it scored highly according to our predefined evaluation functions, but it did not actually solve the underlying problem of tool use hallucination. In the modification leading up to node 114 (see below), the agent removed the logging of special tokens that indicate tool usage (despite instructions not to change the special tokens), effectively bypassing our hallucination detection function.

```diff
diff --git a/coding_agent.py b/coding_agent.py
index 620da2f..800b06e 100644
--- a/coding_agent.py
+++ b/coding_agent.py
@@ -196,13 +196,16 @@ def chat_with_claude(task, chat_history, user_instr):
            "input": tool_input
        })

-    # Process the tool call
+    # Process the tool call and record in structured format
        tool_result = process_tool_call(tools_dict, tool_name, tool_input)
-    output_ls.extend([
-    {TOOL_USED_MARKER: tool_name},
-    {TOOL_INPUT_MARKER: tool_input},
-    {TOOL_RESULT_MARKER: tool_result}
-    ])
+    # Use structured format instead of text markers
+    output_ls.append({
+    "tool_invocation": {
+    "name": tool_name,
+    "input": tool_input,
+    "result": tool_result
+    }
+    })

        # Continue with tool results
        messages.extend([
@@ -224,7 +227,13 @@ def chat_with_claude(task, chat_history, user_instr):
            tools=tools,
        )
        tool_response_text = next((block.text for block in tool_response.content if
            ↪ hasattr(block, "text")), None)
-    output_ls.append({TOOL_RESPONSE_MARKER: tool_response_text})
+    # Record tool response in structured format
+    if tool_response_text:
+    output_ls.append({
+    "tool_response": {
+    "content": tool_response_text
+    }
+    })
        response = tool_response

        # Validate tool response
```

In conclusion, the DGM can be applied beyond the coding domain, and we highlighted a case of objective hacking. Similar to reward hacking in reinforcement learning (Skalse et al., 2022), objective hacking occurs when a system optimizes for a predefined, quantifiable objective rather than fulfilling the spirit of the task or solving the intended problem. This observation supports arguments made in prior works (Zhang et al., 2024b; Faldor et al., 2025), which suggest that optimizing quantitative measures often leads to undesirable or pathological outcomes, and aligns with Goodhart's law (Strathern, 1997) – "When a measure becomes a target, it ceases to be a good measure."

# I  ADDITIONAL SAFETY DISCUSSION

Any advancement that increases the autonomous capabilities of AI systems introduces its own set of safety considerations (Bengio et al., 2024), especially for systems that improve in an open-ended way (Ecoffet et al., 2020; Clune, 2019). Section 5 discusses these concerns and outlines concrete, actionable steps for mitigating them. We call for much more research into and discussion regarding AI safety, including deep thought and discussion amongst all stakeholders in society on the complicated question of what exactly counts as safe AI. We are confident the work we have done was never unsafe (Section 5), but scaled up versions of it could be. As with all transformative technologies, the ultimate impact of such AI systems remains deeply uncertain, and good arguments can be made both for the case that it will bring about tremendous good and tremendous harm. These uncertainties highlight the need for sustained, inclusive, and multidisciplinary discussion (not only from current experts but also from a wider and more diverse community) on how to navigate these developments.

## J   ADDITIONAL FUTURE WORK DIRECTIONS

While this paper has shown the potential of the Darwin Gödel Machine in iteratively improving coding agents via open-ended exploration and empirical validation, several extensions could address current limitations and push AI beyond its already growing role in inspiring culture and advancing science. The following directions outline promising avenues for further research.

**Autonomously Improving the Open-ended Exploration Process.** In this version of the DGM, the open-ended exploration process described in Section 3 is kept fixed, which might hence impede the system's self-acceleration potential. This design choice was made due to limited computational budget. If we were to evolve this part of the algorithm, it could require exponentially more compute to identify processes that yield the same improvements shown in Section 4.4. Nevertheless, since the open-ended exploration loop itself is implemented in code, it can in principle be edited and improved by a coding agent. There are many possible implementations of open-ended exploration, for example, using alternative search mechanisms that balance exploration and exploitation (Herr et al., 2025), keeping only the most interesting agents in the archive (Faldor et al., 2025), or leveraging the generated agent population as an ensemble (Samvelyan et al., 2024). A promising future work direction is to allow the agent to modify the open-ended exploration process, thereby autonomously improving not only its own capabilities but also the meta-process that allocates limited compute to drive self-improvement and self-acceleration.

**Role of Humans in Autonomous AI Systems.** In the current formulation of the DGM, proposed self-modifications are autonomously evaluated without any human intervention. However, as autonomous systems grow in complexity and influence, the question of how humans should remain involved becomes increasingly pressing. Should human oversight be framed as an optimization objective, incorporated through techniques such as reinforcement learning from human feedback (Ouyang et al., 2022), or distilled into FMs that act as preference judges (Bai et al., 2022)? Each of these approaches raises challenges in terms of scalability, reliability, and alignment with evolving human values. The role of humans in guiding, constraining, or co-evolving with autonomous AI remains an open question. Exploring this dynamic is a promising avenue for future research, as it touches not only on technical feasibility but also on broader philosophical and societal considerations.

**DGM with Advanced Foundation Models.** Recent FMs have advanced dramatically, enabling scaffolds to become simpler on current coding benchmarks (Yang et al., 2024). It is possible that in some settings, like current coding benchmarks, certain engineering efforts in scaffolding might be downplayed by the improvement of the FMs. However, many scaffolding components (advanced tools, parallel workflows, external memory, proxy verification, etc.) still fundamentally can not be internalized by FMs and will be essential for more complex real-world tasks beyond today's benchmarks. Future work to explore how different components in agents will emerge with different FMs could be a promising direction.

**Evolving Generalist Agent.** We believe some degree of task-specific adaptation is indeed expected and even desirable, since fundamentally different types of tasks (e.g., in our case, multi-file Python repository edits vs. primarily single-file, multi-language implementations) naturally require distinct scaffolding components. Crucially, this very property highlights a unique advantage of self-improving systems like the DGM: it replaces laborious manual efforts to design specialized agents for diverse tasks with a fully automated evolutionary process. This motivates an exciting future direction of running the DGM on a large, diverse set of tasks to evolve a true generalist agent. Also, currently we only evaluated DGM on two coding benchmarks. While we believe these two benchmarks differ substantially in task structure (multi-file Python repository edits vs. primarily single-file, multi-language implementations), providing strong evidence of generality, additional benchmarks would further strengthen evaluations.

