# OpenReview forum: "Darwin Gödel Machine: Open-Ended Evolution of Self-Improving Agents"
_ICLR.cc/2026/Conference — ICLR 2026 Poster_

### Official Review · Reviewer_VRNN · 2025-10-15

**Soundness:** 3
**Presentation:** 3
**Contribution:** 2
**Rating:** 4
**Confidence:** 4

**Summary:**

This paper introduces the Darwin Gödel Machine, a system designed for the open-ended evolution of self-improving AI coding agents. The core idea is to merge two powerful concepts: self-referential code modification and population-based, open-ended exploration. Unlike the theoretical Gödel Machine, which requires formal proofs of improvement, DGM uses empirical validation on coding benchmarks to guide its evolution. The system maintains a growing archive of all generated agent variants, allowing it to select parents for modification from a diverse pool of "stepping stones". Empirically, DGM demonstrates a significant performance increase on SWE-bench (from 20.0% to 50.0%) and Polyglot (from 14.2% to 30.7%). Ablation studies confirm that both the self-improvement mechanism and the open-ended exploration are essential for this sustained progress.

**Strengths:**

1. The performance gains on two distinct and challenging coding benchmarks are substantial and compelling. The ability to automatically discover sophisticated improvements is a demonstration of the system's efficacy.

2. The paper's primary strength is the successful and novel synthesis of self-referential modification with population-based open-endedness. While these ideas exist in other contexts, their combination here creates a virtuous cycle: better agents become better at creating even more capable offspring, and the evolutionary framework prevents the process from easily getting stuck.

**Weaknesses:**

1. The paper's claim to novelty in the self-referential aspect needs clearer differentiation from prior work. The paper does cite Yin et al. (2024) for the Gödel Agent and notes that in DGM, the downstream task (coding) directly aligns with the self-improvement task (also coding). However, this distinction is brief. A more detailed discussion is needed to clarify what specific limitations of the Gödel Agent's self-reference framework the DGM overcomes, or how its implementation of the concept is fundamentally different and more advantageous beyond the stated task alignment. Without this, the contribution appears more incremental than foundational in this specific dimension.

2. The DGM agent is not fully self-referential. The DGM agent can modify its own coding logic, but the open-ended exploration process itself (i.e., archive maintenance, parent selection strategy) remains fixed and human-designed. This limits the system's full autonomy, as it cannot learn to improve how it explores and evolves. The paper acknowledges this as future work, but it is a major limitation of the current system.

3. The paper is transparent about the high computational cost (an estimated $22,000 for a single SWE-bench run), which is a significant barrier to reproducibility and further research by the academic community. While the results are impressive, the cost may limit the practical applicability and exploration of this method.
4. What is the signal from the environment to support improvement? Is a validation dataset required for self-improvement?

**Questions:**

1. What are the anticipated challenges in allowing the DGM to modify its own open-ended exploration process (e.g., the parent selection mechanism in Appendix C.2)? Would this require a separate reward signal, or could it emerge from the existing objective?

2. Can you discuss potential scenarios where the core assumption of task-capability alignment might break down? For example, could an agent over-specialize in solving small bug fixes (like in SWE-bench) in a way that makes it less capable of performing large-scale architectural refactoring on its own codebase?

3. The analysis highlights the lineage of the final best agent, which traverses some lower-performing nodes. Does an analysis of "dead-end" evolutionary branches offer any insights into common failure modes or deceptive local optima in the agent design space?

---

> ### Author Response · Authors · 2025-11-22
> **Response to Reviewer VRNN (Part 1/3)**
>
> Thank you for reviewing our work. We appreciate the positive comments from you and other reviewers. For example, the recognition that the paper has a “strong motivation” and addresses an “important direction likely to remain relevant in the long term” (3Ycd), and that the DGM represents a “successful and novel synthesis of self-referential modification with population-based open-endedness” (VRNN). We are grateful for the observation that our system shows “clear signs of self-improvement” (Cgiz) and that our ablations “clearly demonstrate the contribution of different components of the proposed algorithm” (3Ycd) as well as confirm that “both the self-improvement mechanism and the open-ended exploration are essential for sustained progress” (VRNN). Reviewers noted that the paper is “clearly written” and “does a good job of setting out the broader vision” (G2FT), with “excellent presentation” (3Ycd) and “useful positioning” connecting the work to meta-learning and Gödel machines (3Ycd). We appreciate the acknowledgment that our system “introduces a novel formulation of self-improving AI” (3Ycd), that it “got [its] novel algorithm to work” (Cgiz), and that the performance gains on challenging coding benchmarks are “substantial and compelling” (VRNN). Finally, we are encouraged by the view that the paper is “original” (3Ycd), and that it takes meaningful steps toward automated, open-ended self-improvement in AI systems.
>
> We have significantly strengthened the manuscript based on your comments and those of other reviewers. We next address each of your concerns and questions.
>
> ---
>
> > The paper's claim to novelty in the self-referential aspect needs clearer differentiation from prior work. The paper does cite Yin et al. (2024) for the Gödel Agent and notes that in DGM, the downstream task (coding) directly aligns with the self-improvement task (also coding). However, this distinction is brief. A more detailed discussion is needed to clarify what specific limitations of the Gödel Agent's self-reference framework the DGM overcomes, or how its implementation of the concept is fundamentally different and more advantageous beyond the stated task alignment. Without this, the contribution appears more incremental than foundational in this specific dimension.
>
> In Yin et al. (2024), a single system is used both to solve downstream tasks and to recursively modify itself. However, as we note in the manuscript, the downstream tasks or meta-utility in the Yin et al. (2024) do not necessarily align with the capabilities required for effective self-improvement. This misalignment can limit the extent to which improvements on the downstream task provide meaningful guidance for meta-level modifications. Hence, the improvements do not necessarily lead to an improvement in self-improvement itself. In contrast, the DGM is designed such that the downstream task (i.e., coding) directly aligns with the skills required for modifying its own codebase. As a result, progress on the downstream task provides a reliable and informative signal for self-improvement, enabling the possibility of self-accelerating development. We empirically validate this hypothesis in the “DGM w/o self-improve” baseline, while the Yin et al. (2024) paper does not include this key analysis or experiment.
>
> Another important difference is the nature of the downstream tasks each system targets. Yin et al. (2024) optimizes for non-coding domains. This distinction is non-trivial: an agent that performs well on non-coding tasks may not possess the capabilities required to edit code at the level demanded by repository-scale benchmarks such as SWE-bench. In contrast, the DGM operates on challenging coding tasks, thereby demonstrating a self-referential, self-improving system in a domain that directly exercises and validates the core capabilities needed for recursive code modification.
>
> The DGM also introduces another key difference: an open-ended exploration mechanism. The DGM maintains an archive of agents generated throughout its evolution and continually explores different branches. Indeed, this is the reason for the “Darwin” in its name. This mechanism helps the system avoid local optima and allows it to accumulate diverse “stepping-stone” innovations that may not yield immediate gains but eventually lead to a higher performing solution. We empirically show this open-ended accumulation of stepping stones is helpful for the performance in our DGM w/o open-ended exploration control. Yin et al. (2024) does not include such an open-ended exploration process.
>
> Together, these distinctions constitute substantive conceptual and practical advances over prior self-referential frameworks. We clarify these points more explicitly in the revised manuscript.

---

> > ### Author Response · Authors · 2025-11-22
> > **Response to Reviewer VRNN (Part 2/3)**
> >
> > > The DGM agent is not fully self-referential. The DGM agent can modify its own coding logic, but the open-ended exploration process itself (i.e., archive maintenance, parent selection strategy) remains fixed and human-designed. This limits the system's full autonomy, as it cannot learn to improve how it explores and evolves. The paper acknowledges this as future work, but it is a major limitation of the current system.
> >
> > We agree that the current implementation of the DGM does not yet support self-modification of the open-ended exploration process. As noted, we discuss this limitation in detail and outline it as an important direction for future work in Appendix I of the original manuscript (Appendix J in the revised version). However, nothing in the DGM frameworks precludes adding these degrees of freedom, thus the idea does include these exciting possibilities. We simply narrowed the scope to make the science and, critically, costs more tractable for this first paper on the subject.
> >
> > > The paper is transparent about the high computational cost (an estimated $22,000 for a single SWE-bench run), which is a significant barrier to reproducibility and further research by the academic community. While the results are impressive, the cost may limit the practical applicability and exploration of this method.
> >
> > We acknowledge that the current experiment on SWE-bench requires considerable compute, and have included this as a limitation in the manuscript (Section 6). That said, we also demonstrate significantly lower costs in another benchmark Polyglot (estimated at ~$200), which suggests that expenses vary greatly by task complexity, where SWE-bench is among the more complex and resource-intensive in coding benchmarks. Also, many impactful approaches (e.g., LLM training when it first started gaining traction) similarly began with high computational costs. Similar to these pioneering works, we hope to open the door to future work on improving the efficiency and scalability of the approach. Moreover, many leading coding agents on the SWE-bench leaderboard are backed by industrial companies that pay expert full-time researchers and engineers to improve performance, incurring substantial human labor costs. In contrast, our approach achieves SoTA-level performance through fully autonomous self-improvement without human intervention during execution, potentially offering savings when factoring in the comparative costs of specialized AI development talent versus API usage. Finally, as foundation models and compute costs continue to plummet exponentially (a recent estimate put it at a 300x drop in one year! https://x.com/chatgpt21/status/1990516566073729362), methods like the DGM will become increasingly accessible and practical.
> >
> > > What is the signal from the environment to support improvement? Is a validation dataset required for self-improvement?
> >
> > In our current implementation, we do not use a separate validation dataset because we only sought improvement on that dataset as a test of the system for this first paper, rather than an agent that generalizes. Instead, the signals that drive improvement come directly from performance on the benchmark tasks as well as logs from failed attempts. These provide sufficient feedback for the agent to propose and implement self-modifications. However, interestingly and importantly, despite the absence of a dedicated validation set, we observe that the improvements discovered by the DGM do not overfit to the training tasks. Rather, they generalize across different tasks and even across different underlying models, as shown in Figure 4 and discussed in Section 4.4. This suggests that the self-improvements discovered by the system capture broadly useful capabilities rather than task-specific shortcuts. In future work, one could use standard machine learning methods for specifically seeking general agents (e.g. training on a wide distribution of tasks).

---

> > > ### Author Response · Authors · 2025-11-22
> > > **Response to Reviewer VRNN (Part 3/3)**
> > >
> > > > What are the anticipated challenges in allowing the DGM to modify its own open-ended exploration process (e.g., the parent selection mechanism in Appendix C.2)? Would this require a separate reward signal, or could it emerge from the existing objective?
> > >
> > > Allowing the DGM to modify its own open-ended exploration process introduces several challenges. First, to be efficient, the agent would need to reason about what kinds of meta-level changes are necessary. Second, such modifications would likely require significantly more computation, since evaluating alternative exploration strategies generally requires multiple experimental runs (or at least many evaluations) to estimate their effects. When we human scientists developed the DGM, we typically adjusted parent-selection strategies only after observing several iterations and developing an intuition for the exploration-exploitation trade-off; enabling an AI system to acquire similar intuition is an interesting area for future work.
> > >
> > > It remains uncertain whether a separate reward signal is necessary for modifying the open-ended exploration process. In one sense, the existing task-level reward (e.g., SWE-bench performance) is sufficient, because any meta-level modification (e.g., adjusting the parent-selection mechanism) ultimately aims to improve the same downstream objective. However, in another sense, enabling an agent to modify its own exploration process requires evaluating the long-term derivative of task performance. This higher-order signal is more difficult to estimate than the immediate task performance, and in practice may require introducing an auxiliary reward or proxy that captures improvement at the population or lineage level. This is an interesting direction for future work, which we highlight in Appendix I of the original manuscript (Appendix J in the revised version).
> > >
> > > > Can you discuss potential scenarios where the core assumption of task-capability alignment might break down? For example, could an agent over-specialize in solving small bug fixes (like in SWE-bench) in a way that makes it less capable of performing large-scale architectural refactoring on its own codebase?
> > >
> > > The core assumption of task-capability alignment breaks when the skills required for self-improvement differ substantially from those needed for the downstream task. For example, if the downstream task were simply to generate an endless stream of random numbers, success on that task would provide no useful signal for improving the agent’s ability to modify its own codebase. More subtly, it could be possible that, even within coding domains, an agent could over-specialize in tasks such as small bug fixes (e.g., those in SWE-bench) in ways that do not transfer to larger-scale architectural refactoring or meta-level reasoning about its own design. These considerations highlight the importance of selecting an appropriate set of tasks for the agent to optimize. An important avenue for future work is determining how this target task set should itself evolve over time to promote more general and transferable self-improvement capabilities (discussed in Section 6).
> > >
> > > > The analysis highlights the lineage of the final best agent, which traverses some lower-performing nodes. Does an analysis of "dead-end" evolutionary branches offer any insights into common failure modes or deceptive local optima in the agent design space?
> > >
> > > We did not qualitatively observe any patterns that would unambiguously indicate whether a given branch is a dead end or a promising variation. It is difficult to precisely determine when a node should be considered a true “dead-end”. In our current implementation, branching is stopped only when an agent lacks basic code-editing capabilities, since such an agent cannot modify itself and would therefore be stuck in its existing implementation forever. Beyond this criterion, identifying which evolutionary paths are genuinely unproductive versus merely slow to yield improvements is nontrivial (or computationally expensive). Understanding and characterizing these “dead-end” branches, along with developing principled methods for deciding which paths are worth further exploration, is an important direction for future work. We discuss this topic in more detail in Appendix I of the original manuscript (Appendix J in the revised version).
> > >
> > > ---
> > >
> > > We thank the reviewer once more for the helpful feedback. We have made many improvements to the paper thanks to your comments and those of the other reviewers. We think it is much improved and hope you agree. If, after reviewing our responses and revisions, you feel that a higher score is warranted, we would deeply appreciate your reconsideration. While the paper did receive two high scores, the current 4s could prevent its publication, yet we believe the community would benefit from learning about this work. Thank you!

---

### Official Review · Reviewer_Cgiz · 2025-10-31

**Soundness:** 3
**Presentation:** 2
**Contribution:** 2
**Rating:** 4
**Confidence:** 3

**Summary:**

The paper proposes Darwin Godel Machines (DGMs) which is a system for self-improving AI except that it uses a fixed LLM (which they refer to as a frozen FM). DGM keeps an archive of agent variants and uses a parent-selection scheme to choose agents to self-modify. Each child is empirically evaluated on coding benchmarks and, if functional, added back to the archive. Empirical results are presented.

**Strengths:**

They got their novel algorithm to work, with clear signs of self improvement. Ablations show that parts of their algorithm are helping. Authors plan on releasing code.

**Weaknesses:**

1. The whole setup is the same as Recursive Self-Improving Code Generation (RSICG, Zelikman et al 2023b) which this paper cites, and the paper should be presented as a follow-up and improvement over that work. Both works consider the idea of recursive self-improvement with a fixed FM.
2. Empirically, it should be compared to that work. (Side note about the name: aren't Godel machines more about provable improvements?)
3. The paper repeatedly claims its main novelty is RSICG specifically:
> "the first self-improving system powered by FMs with open-ended exploration, where progress on its evaluation benchmarks can directly translate into better self-improvement capabilities"
What does this mean and how does it go beyond Zelikman et al?
4. The main paper does not contain much detail at all about the algorithm, which is mainly described and defined in the appendix.
5. The ethical risks of RSI are not discussed. But clearly, the development of RSI poses potential risks that numerous luminaries claim are existential. See [https://superintelligence-statement.org/](https://superintelligence-statement.org/) for example. The discussion around safety focuses on the fact that the this paper's experiments were run in a sandbox, but there wasn't much concern that the system in the paper is superintelligent. The real risk is that it's advancing science towards that goal without clear discussion of why the benefits of this progress outweigh the risks.

**Questions:**

Is this RSICG and, if so, why is it not presented in that light? What is the novelty in this paper? Why is it not compared empirically to Zelikman et al, whose code is available online? Can you define the algorithm in the paper?

**Details Of Ethics Concerns:**

The ethical risks of RSI are not discussed. However, the development of Recursive Self-Improving AI systems poses potential risks that numerous luminaries claim are existential. See [https://superintelligence-statement.org/](https://superintelligence-statement.org/) for example. The risk of advancing science towards that goal is not discussed, and the paper needs a discussion of why the benefits of this progress outweigh the risks.

---

> ### Author Response · Authors · 2025-11-22
> **Response to Reviewer Cgiz (Part 1/3)**
>
> Thank you for reviewing our work. We appreciate the positive comments from you and other reviewers. For example, the recognition that the paper has a “strong motivation” and addresses an “important direction likely to remain relevant in the long term” (3Ycd), and that the DGM represents a “successful and novel synthesis of self-referential modification with population-based open-endedness” (VRNN). We are grateful for the observation that our system shows “clear signs of self-improvement” (Cgiz) and that our ablations “clearly demonstrate the contribution of different components of the proposed algorithm” (3Ycd) as well as confirm that “both the self-improvement mechanism and the open-ended exploration are essential for sustained progress” (VRNN). Reviewers noted that the paper is “clearly written” and “does a good job of setting out the broader vision” (G2FT), with “excellent presentation” (3Ycd) and “useful positioning” connecting the work to meta-learning and Gödel machines (3Ycd). We appreciate the acknowledgment that our system “introduces a novel formulation of self-improving AI” (3Ycd), that it “got [its] novel algorithm to work” (Cgiz), and that the performance gains on challenging coding benchmarks are “substantial and compelling” (VRNN). Finally, we are encouraged by the view that the paper is “original” (3Ycd), and that it takes meaningful steps toward automated, open-ended self-improvement in AI systems.
>
> We have significantly strengthened the manuscript based on your comments and those of other reviewers. We next address each of your concerns and questions.
>
> ---
>
> > aren't Godel machines more about provable improvements?
>
> Gödel Machines (Schmidhuber, 2007) are indeed about provable improvements. However, in practice, and without strong restrictive assumptions about the system, it is impossible to formally prove whether a given modification to an AI system will be beneficial. Instead of requiring formal proofs, the DGM empirically validates self-modifications against a benchmark, allowing the system to improve and explore based on observed results. However, using empirical validation runs into the risk of getting stuck in a local optimum, especially since the search space the DGM operates in (i.e., all computable algorithms) is so vast. To address this, the DGM takes inspiration from open-ended algorithms  (e.g. Darwinian and cultural evolution) and uses open-ended exploration to continuously accumulate and explore an archive of solutions. We discuss the inspiration from Godel Machines and Darwinian evolution in greater detail in the abstract, Section 1, Section 3, and Appendix B.
>
> > The main paper does not contain much detail at all about the algorithm, which is mainly described and defined in the appendix.
> > Can you define the algorithm in the paper?
>
> We have moved the details of the parent selection mechanism into the main text of the revised manuscript, which we think was the main missing detail (and we agree it should be in the main text).
>
> > The ethical risks of RSI are not discussed. But clearly, the development of RSI poses potential risks that numerous luminaries claim are existential. See https://superintelligence-statement.org/ for example. The discussion around safety focuses on the fact that the this paper's experiments were run in a sandbox, but there wasn't much concern that the system in the paper is superintelligent. The real risk is that it's advancing science towards that goal without clear discussion of why the benefits of this progress outweigh the risks.
>
> We agree that superintelligence may bring both significant potential benefits and serious risks. As with all new technology, its ultimate impact remains uncertain. In this paper, we have devoted substantial time and effort to discussing these concerns and outlining concrete, actionable steps for mitigating them (Section 5).
>
> We have also updated the manuscript to further highlight the uncertainties surrounding safety aspects of recursive self-improvement and superintelligence, and the need for broader, ongoing discussion (not only from current experts but also from a wider and more diverse community) on how to navigate these developments and what it would even mean to achieve safe superintelligence. We have added this discussion to Appendix I of the revised manuscript.

---

> ### Author Response · Authors · 2025-11-22
> **Response to Reviewer Cgiz (Part 2/3)**
>
> > The whole setup is the same as Recursive Self-Improving Code Generation (RSICG, Zelikman et al 2023b) which this paper cites, and the paper should be presented as a follow-up and improvement over that work. Both works consider the idea of recursive self-improvement with a fixed FM.
> > Empirically, it should be compared to that work.
> > The paper repeatedly claims its main novelty is RSICG specifically:
> "the first self-improving system powered by FMs with open-ended exploration, where progress on its evaluation benchmarks can directly translate into better self-improvement capabilities" What does this mean and how does it go beyond Zelikman et al?
> > Is this RSICG and, if so, why is it not presented in that light? What is the novelty in this paper? Why is it not compared empirically to Zelikman et al, whose code is available online?
>
> The paper you refer to is “Self-Taught Optimizer (STOP): Recursively Self-Improving Code Generation” (Zelikman et al., 2023). We refer to this work as STOP, following its title. We agree that both STOP and the DGM investigate recursive self-improvement with fixed foundation models (FMs). However, the DGM introduces several conceptual, algorithmic, and empirical differences that place it beyond STOP, and we have revised the paper to make these distinctions more salient.
>
> STOP evaluates self-improvement on relatively self-contained program synthesis tasks. The DGM instead evaluates on SWE-bench and Polyglot, which require reasoning over large codebases, understanding cross-file dependencies, and integrating fixes or new functionality into multi-module software. These behaviors are much closer to real-world software engineering. These tasks are both harder and more aligned with the skills required for self-modification, which contributes to key conceptual differences described below.
>
> In STOP, the skills required to perform downstream tasks differ from the skills required to modify the improver itself. Because of this mismatch (and to get self-referentiality), STOP introduces a meta-utility function: an additional handcrafted objective that scores the improver based on its average performance over downstream tasks. The improver and downstream task-solving programs are explicitly separated, with the improver modifying itself using the handcrafted meta-utility.
> In contrast, the DGM does not rely on a meta-utility function or a separate improver. The agent that solves downstream tasks is the same agent that modifies itself. This is possible because the downstream tasks we study (e.g., SWE-bench and Polyglot) require similar competencies needed for self-modification: multi-step reasoning, code synthesis, integration with existing code, and debugging. This alignment enables direct self-referentiality: improvements in downstream task performance translate immediately into improved self-improvement capability. We empirically validate this with our “DGM w/o self-improve” ablation.
> This unification of task-learning and self-modification, enabled by challenging real-world tasks, is a core conceptual departure from STOP.
>
> Because STOP separates the improver from the optimized programs it emits for downstream tasks, the generated programs are not transferable across tasks; only the improver is. In the DGM, by contrast, the agent itself is the optimized target for downstream tasks. It is directly transferable across downstream tasks and FMs. This leads to qualitatively different transfer results compared to STOP, which we highlight more prominently in the paper.
>
> While both STOP and the DGM rely on empirical evaluation on benchmarks to obtain self-improvement signals, the DGM incorporates an open-ended exploration component that accumulates an archive of generated agents. This open-ended exploration helps the system avoid local optima and is integral to its design (supported by results of the “DGM w/o open-ended exploration” control), whereas STOP does not implement such a mechanism.
>
> In conclusion, while both systems fit within the broad umbrella of recursive self-improvement, the conceptual and design differences above lead to different problem settings, different assumptions, and different self-referentiality and self-improvement claims (improver-level vs agent-level). STOP’s code does not directly implement or evaluate on tasks comparable to those used in the DGM, and a direct empirical comparison would require substantial reinterpretation of STOP’s implementation. We clarify this distinction more carefully in the revised manuscript.

---

> ### Author Response · Authors · 2025-11-22
> **Response to Reviewer Cgiz (Part 3/3)**
>
> We thank the reviewer once more for the helpful feedback. We have made many improvements to the paper thanks to your comments and those of the other reviewers. We think it is much improved and hope you agree. If, after reviewing our responses and revisions, you feel that a higher score is warranted, we would deeply appreciate your reconsideration. While the paper did receive two high scores, the current 4s could prevent its publication, yet we believe the community would benefit from learning about this work. Thank you!

---

### Official Review · Reviewer_3Ycd · 2025-11-01

**Soundness:** 3
**Presentation:** 4
**Contribution:** 3
**Rating:** 8
**Confidence:** 5

**Summary:**

DGM is a self-improving system that autonomously evolves its own codebase. It maintains an archive of coding agents that iteratively self-modify and empirically validate improvements on coding benchmarks. Through this process, DGM becomes increasingly proficient at both solving coding tasks and performing future self-improvements. Empirical results show substantial gains on SWE-bench and Polyglot.

**Strengths:**

Strong motivation. It is reasonable to expect that AI research will eventually be automated. This paper addresses an important direction likely to remain relevant in the long term.

Clear positioning. The connection to both meta-learning and the Gödel machine helps situate the contribution within established frameworks for self-improving systems.

Useful Ablations. The ablations are particularly useful, as they clearly demonstrate the contribution of different components of the proposed algorithm.

Original Ideas. The paper introduces a novel formulation of self-improving AI that integrates meta-learning, evolution, and recursive self-improvement.

**Weaknesses:**

Baselines are missing. The baselines used for comparison are also designed by researchers (they are ablated versions of the proposed method), and no direct comparison is made against other self-improving or open-ended systems in the literature. Adding such a baseline would strengthen the empirical claims.

Only two benchmarks. SWE-bench and Polyglot are used in evaluation. Including an additional benchmark could help validate the generality of the approach.

Economic considerations. It is unclear whether better-performing agents come with proportionally greater inference costs. Theoretically, performance could improve simply by sampling n responses in parallel and increasing n at each generation step, especially in verifiable software engineering tasks. An analysis comparing the discovered agents to trivial test-time scaling (best of n) would help establish whether the improvements are meaningful rather than inefficient increase of computational costs.

see https://arxiv.org/pdf/2407.01502

The paper states: “Our framework envisions agents that can rewrite their own training scripts (including training a new foundation model (FM)). However, we do not show that in this paper, as training FMs is computationally intensive and would introduce substantial additional complexity, which we leave as future work.”

While theoretically plausible, the economic and computational cost of such a setup can make large-scale application economically unjustifiable.

**Questions:**

- Do better agents necessarily incur higher inference costs? Could you plot a Pareto frontier that includes these inference costs, and are the discovered agents Pareto-optimal?
- Is DGM without self-improvement equivalent to ADAS, and DGM without open-ended exploration equivalent to STOP (Self-Taught Optimizer)? Please discuss their similarities and differences.
- If DGM can be viewed as combining ADAS and STOP with evolution, what is the benefit of reframing it as a Darwin–Gödel Machine?

---

> ### Author Response · Authors · 2025-11-22
> **Response to Reviewer 3Ycd (Part 1/2)**
>
> Thank you for reviewing our work. We appreciate the positive comments from you and other reviewers. For example, the recognition that the paper has a “strong motivation” and addresses an “important direction likely to remain relevant in the long term” (3Ycd), and that the DGM represents a “successful and novel synthesis of self-referential modification with population-based open-endedness” (VRNN). We are grateful for the observation that our system shows “clear signs of self-improvement” (Cgiz) and that our ablations “clearly demonstrate the contribution of different components of the proposed algorithm” (3Ycd) as well as confirm that “both the self-improvement mechanism and the open-ended exploration are essential for sustained progress” (VRNN). Reviewers noted that the paper is “clearly written” and “does a good job of setting out the broader vision” (G2FT), with “excellent presentation” (3Ycd) and “useful positioning” connecting the work to meta-learning and Gödel machines (3Ycd). We appreciate the acknowledgment that our system “introduces a novel formulation of self-improving AI” (3Ycd), that it “got [its] novel algorithm to work” (Cgiz), and that the performance gains on challenging coding benchmarks are “substantial and compelling” (VRNN). Finally, we are encouraged by the view that the paper is “original” (3Ycd), and that it takes meaningful steps toward automated, open-ended self-improvement in AI systems.
>
> We have significantly strengthened the manuscript based on your comments and those of other reviewers. We next address each of your concerns and questions.
>
> ---
>
> > Baselines are missing. The baselines used for comparison are also designed by researchers (they are ablated versions of the proposed method), and no direct comparison is made against other self-improving or open-ended systems in the literature. Adding such a baseline would strengthen the empirical claims.
>
> Thank you for the suggestion. Our baseline DGM w/o self-improve directly implements the approach of ADAS (Hu et al., 2025), a representative open-ended meta-learning method for designing agentic systems. Similarly, DGM w/o open-ended exploration and the DGM Greedy variant reported in Table 1 closely reproduce the single-lineage, greedy self-improvement strategy of the concurrent self-improving coding agent in Robeyns et al. (2025), in which the system always builds on its most recent or best functional version without maintaining a persistent archive of diverse stepping stones. We clarify these correspondences explicitly in the main text in Section 4.3 and Appendix A.3.
>
> > Only two benchmarks. SWE-bench and Polyglot are used in evaluation. Including an additional benchmark could help validate the generality of the approach.
>
> We agree that, typically, additional benchmarks would further strengthen evaluations. However, running on SWE-bench and Polyglot is extremely expensive and has already exhausted our computational budget. We also believe these two benchmarks differ substantially in task structure (multi-file Python repository edits vs. primarily single-file, multi-language implementations), providing strong evidence of generality. We therefore leave evaluations on additional benchmarks to future work and explicitly have added this discussion to the future work section in the revised manuscript.
>
> > Economic considerations. It is unclear whether better-performing agents come with proportionally greater inference costs. Theoretically, performance could improve simply by sampling n responses in parallel and increasing n at each generation step, especially in verifiable software engineering tasks. An analysis comparing the discovered agents to trivial test-time scaling (best of n) would help establish whether the improvements are meaningful rather than inefficient increase of computational costs.
>
> We thank the reviewer for raising this important point. Higher-performing agents discovered by the DGM do indeed incur greater inference costs than the initial agent, but cost and performance are not strictly correlated, where some expensive agents underperform cheaper ones. Regarding test-time scaling, typical trivial best-of-n baselines in coding benchmarks assume access to ground-truth tests for selecting the best candidate, whereas our evaluations (and most real-world deployments) are strictly pass@1 with no ground-truth test feedback provided to the agent. Directly comparing to best-of-n therefore requires either changing the evaluation setting or designing sophisticated proxy verifiers, which is non-trivial. Interestingly, several top agents evolved by the DGM spontaneously implement internal parallel sampling workflows with proxy verification (i.e., the agent writing tests to select among multiple candidate patches), attempting to get best-of-n gains by checking against agent-generated unit tests within a single submission. We have added this discussion to Appendix E.1 in the revised manuscript.

---

> ### Author Response · Authors · 2025-11-22
> **Response to Reviewer 3Ycd (Part 2/2)**
>
> > The paper states: “Our framework envisions agents that can rewrite their own training scripts (including training a new foundation model (FM)). However, we do not show that in this paper, as training FMs is computationally intensive and would introduce substantial additional complexity, which we leave as future work.” While theoretically plausible, the economic and computational cost of such a setup can make large-scale application economically unjustifiable.
>
> We fully recognize the practical and economic challenges of a DGM variant that rewrites its own training scripts and trains new foundation models. Therefore, we leave it as future work in the paper. However, many now impactful paradigms (e.g., large-scale pre-training of LLMs) were once dismissed for similar cost concerns, yet became feasible as compute efficiency and model performance improved dramatically (a recent estimate suggests that the costs for using foundation models have dropped by roughly 300x in one year! https://x.com/chatgpt21/status/1990516566073729362). Recent work, such as DiscoPOP (Lu et al., 2024), demonstrates early steps toward automatically optimizing LLM training algorithms. We believe visionary research should outline plausible long-term directions even when current costs are prohibitive, especially as hardware and algorithmic efficiency continue their exponential progress. We discuss the current and future cost aspects of the DGM in Appendix E.1.
>
> > Is DGM without self-improvement equivalent to ADAS, and DGM without open-ended exploration equivalent to STOP (Self-Taught Optimizer)? Please discuss their similarities and differences.
>
> DGM without self-improvement is equivalent to ADAS (in coding agent setting). DGM without open-ended exploration is similar to STOP, but there are several key differences: unlike STOP, which separates the improver (meta-optimizer) from the optimized downstream agents and relies on a hand-designed meta-utility signal, DGM uses a single, unified coding agent that both solves downstream tasks and self-modifies, ensuring that benchmark performance gains directly enhance self-improvement capability without requiring any explicit meta-objective. Additionally, DGM-generated agents are transferable across tasks, whereas STOP’s task-specific programs are not.
>
> > If DGM can be viewed as combining ADAS and STOP with evolution, what is the benefit of reframing it as a Darwin–Gödel Machine?
>
> The Gödel Machine (Schmidhuber, 2007) predates both ADAS and STOP and represents the early vision of rigorous, self-referential self-improvement. Therefore, we present our work as a direct extension of the classic Gödel Machine—relaxing its infeasible formal-proof requirement through empirical validation while augmenting it with Darwinian open-ended evolution to achieve sustained real-world progress. We believe this framing honors the historical roots of the idea and emphasizes our core contribution.
>
> ---
>
> We hope we have answered all your questions and concerns. We have made many improvements to the paper thanks to your comments and those of the other reviewers. We think it is much improved and hope you agree. We sincerely appreciate your engagement with our work and hope that our revisions and clarifications reinforce your support for this line of research.

---

### Official Review · Reviewer_G2FT · 2025-11-01

**Soundness:** 2
**Presentation:** 3
**Contribution:** 3
**Rating:** 6
**Confidence:** 4

**Summary:**

This paper studies how LLM based coding agent system can autonomously improve by editing their own code base. The approach proposed in this paper improves on a line of similar work with the key introduction of a tree-structured archive of past coding agent implementations paired with a parent selection rule which the authors refer to as an "open-ended exploration loop". The authors evaluate their method on subsets of SWE-Bench and the Aider Polyglot benchmark, and demonstrate a good improvement from their first agent iteration to their last.

**Strengths:**

The paper is for the most part clearly written and does a good job of setting out the broader vision for self-improving agent systems and the promise that open-endedness holds. However, a lot of space is used up in the abstract and introduction for this vision-setting, sometimes introducing concepts which aren't returned to later. Perhaps more of the main body of the paper could be devoted to what is in my opinion the paper's main contribution, which is the "open-ended exploration" loop and details of the parent selection mechanism, which becomes somewhat lost in the main text, with key details pushed into the appendix. Indeed, the results in Figure 6 in the appendix show that this is the key enabler of the DGM's performance

**Weaknesses:**

The initial model performance (as shown, for instance in Figure 2 at iteration 0) is surprisingly low. Claude 3.5 Sonnet (new) [scored 49%](https://www.anthropic.com/news/3-5-models-and-computer-use) on the full SWE-Bench Verified benchmark using Anthropic's internal harness, while o3-mini (medium) [scores 53.8%](https://aider.chat/docs/leaderboards/) on Aider (albeit with pass@2 instead of pass@1 as used in the paper). What explains the seemingly large 29% and 39.8% drop, respectively, between the initial agent at iteration 0 used in the paper and the Anthropic and Aider scaffolds used with the same models? While I understand that Aider is a more mature scaffold which may explain the larger difference (and you may have used the old instead of the new benchmark so my 53.8% number may be wrong), I believe the agent scaffold used by Anthropic for 3.5 Sonnet (new) was fairly minimal. It might appear that the nature of the early improvements implemented by the DGM merely relate to fixing basic functionality in the agent loop. Moreover, it does not seem that the DGM (at not-insignificant monetary cost, Appendix E.1), has demonstrated an improvement beyond allowing the base model (3.5 Sonnet or o3-mini) to perform at its usual level of approximately 49% and 53% on SWE-Bench and Polyglot respectively, by fixing the initial scaffold.

The paper also only presents experiments on models released around the end of last year. In the intervening time, model labs have continued training models, improving their performance as coding agents by performing large-scale RL training with environments resembling simple agent scaffolds. For instance, the "bash only" [SWE-Bench leaderboard](https://www.swebench.com/) or recent papers such as [Dai et al., 2025](https://arxiv.org/pdf/2509.25873) demonstrate that modern models placed in minimal agent loops are outperforming those placed in more elaborate agent scaffolds. The corollary is that while agent scaffolds can support weaker models and guide them to perform tasks such as editing files or navigating codebases which they were not trained to do, as the foundation models are trained to perform these skills natively, such agent 'scaffolding' can become obsolete or even detrimental to agent performance. As a result, the paper's results and the relevance of its claims about being able to improve coding agents by editing their own codebase would be strengthened if they were re-validated with, for instance, Claude 4.5 Sonnet and GPT-5-thinking. Taking the SWE-Bench experiments for example, these should be started with the minimal [mini-swe-agent](https://github.com/SWE-agent/mini-swe-agent) which already achieves 70.6% on SWE-Bench Verified. It would be valuable to demonstrate that the kinds of agent scaffold improvements DGM is able to implement continue to lead to meaningful performance improvements with these modern models. It is possible that with a more sophisticated base model, the types of scaffold improvements may themselves become more sophisticated and abstract; moving beyond simple editing mechanisms and the like towards task management and other more abstract functions. However this has yet to be shown.

In light of the above, it seems that simply modifying the agent's codebase limits the "action space" and precludes achieving the continuous and open-ended recursive self improvement discussed in the introduction. While I note that fine-tuning the weights of the agent's model has been acknowledged by the authors as something for future work, I do believe that the paper would have been strengthened by examining this approach here as a first instance: potentially on a much simpler task than SWE-Bench and with a much smaller foundation model.

Finally, my reading of the transferability results of the DGM-discovered agent across different benchmarks and programming languages at the top of page 8 is less optimistic than the authors put it. The results seem to indicate slight over-fitting to the task the DGM self-improves with, rather than broad-based improvements to the coding agent's capabilities. What is however interesting is that SWE-Bench appears to be a better "teaching" task than polyglot, with the Polyglot-trained agent doing comparatively poorly on SWE-Bench, whereas the SWE-Bench trained agent almost matches the performance of the Polyglot-trained agent on Polyglot.

Nits:
- L243: The line reading "We evaluate the DGM on two popular benchmarks" reads awkwardly, given the discussion of SWE-Bench and Polyglot immediately prior.

**Questions:**

- Why is the baseline (iteration 0) performance of the initial agent quite low (20%) on SWE-Bench initially?
- Where does the 'representative agent baseline (Aider)' number come from on the right pane of Figure 2? Is this the score for o3-mini on Aider's *old* code editing benchmark?
- Looking at the curvature of the light blue 'Average of Archive' line in Figure 3, it appears to be negative from iteration 50 onwards. What do you think is the cause of this plateau, and does this point to a limitation in the open-ended improvement loop (perhaps new idea generation, or the FM's coding ability, or something else entirely)?
- What is the variance within each run? For example, in Figure 2, at times the green line (DGM without self-improvement) is ahead of the full DGM's blue line. While I understand the significant time and monetary costs of doing each run, if each experiment were to be repeated 100 times, do you think this trend would continue? In particular, what prevents over-specialization of the agent to the proxy training task (i.e. solving narrow issue-resolution tasks in SWE-Bench's style) which may in time preclude effective self-improvement?

---

> ### Author Response · Authors · 2025-11-22
> **Response to Reviewer G2FT (Part 1/3)**
>
> Thank you for reviewing our work. We appreciate the positive comments from you and other reviewers. For example, the recognition that the paper has a “strong motivation” and addresses an “important direction likely to remain relevant in the long term” (3Ycd), and that the DGM represents a “successful and novel synthesis of self-referential modification with population-based open-endedness” (VRNN). We are grateful for the observation that our system shows “clear signs of self-improvement” (Cgiz) and that our ablations “clearly demonstrate the contribution of different components of the proposed algorithm” (3Ycd) as well as confirm that “both the self-improvement mechanism and the open-ended exploration are essential for sustained progress” (VRNN). Reviewers noted that the paper is “clearly written” and “does a good job of setting out the broader vision” (G2FT), with “excellent presentation” (3Ycd) and “useful positioning” connecting the work to meta-learning and Gödel machines (3Ycd). We appreciate the acknowledgment that our system “introduces a novel formulation of self-improving AI” (3Ycd), that it “got [its] novel algorithm to work” (Cgiz), and that the performance gains on challenging coding benchmarks are “substantial and compelling” (VRNN). Finally, we are encouraged by the view that the paper is “original” (3Ycd), and that it takes meaningful steps toward automated, open-ended self-improvement in AI systems.
>
> We have significantly strengthened the manuscript based on your comments and those of other reviewers. We next address each of your concerns and questions.
>
> ---
>
> > However, a lot of space is used up in the abstract and introduction for this vision-setting, sometimes introducing concepts which aren't returned to later. Perhaps more of the main body of the paper could be devoted to what is in my opinion the paper's main contribution, which is the "open-ended exploration" loop and details of the parent selection mechanism, which becomes somewhat lost in the main text, with key details pushed into the appendix. Indeed, the results in Figure 6 in the appendix show that this is the key enabler of the DGM's performance.
>
> Thank you for your suggestions. We have expanded Section 3 with the following text: “Parent selection is roughly proportional to each agent's performance score and inversely proportional to the number of its children with codebase-editing functionality. This favors high-performing agents that have been underexplored (i.e., have fewer existing children), thereby promoting both exploitation of strong performers and exploration of promising but less-sampled lineages. All agents retain a non-zero selection probability, ensuring that any path to improvement remains feasible given sufficient compute.” We will also move the key ablation (currently Figure 6 in the appendix) that demonstrates the critical role of open-ended exploration into the main paper if page limits allow in the camera-ready version, or otherwise reference it explicitly and more prominently in Section 4.

---

> ### Author Response · Authors · 2025-11-22
> **Response to Reviewer G2FT (Part 2/3)**
>
> > The paper also only presents experiments on models released around the end of last year. In the intervening time, model labs have continued training models, improving their performance as coding agents by performing large-scale RL training with environments resembling simple agent scaffolds. For instance, the "bash only" SWE-Bench leaderboard or recent papers such as Dai et al., 2025 demonstrate that modern models placed in minimal agent loops are outperforming those placed in more elaborate agent scaffolds. The corollary is that while agent scaffolds can support weaker models and guide them to perform tasks such as editing files or navigating codebases which they were not trained to do, as the foundation models are trained to perform these skills natively, such agent 'scaffolding' can become obsolete or even detrimental to agent performance. As a result, the paper's results and the relevance of its claims about being able to improve coding agents by editing their own codebase would be strengthened if they were re-validated with, for instance, Claude 4.5 Sonnet and GPT-5-thinking. Taking the SWE-Bench experiments for example, these should be started with the minimal mini-swe-agent which already achieves 70.6% on SWE-Bench Verified. It would be valuable to demonstrate that the kinds of agent scaffold improvements DGM is able to implement continue to lead to meaningful performance improvements with these modern models. It is possible that with a more sophisticated base model, the types of scaffold improvements may themselves become more sophisticated and abstract; moving beyond simple editing mechanisms and the like towards task management and other more abstract functions. However this has yet to be shown. In light of the above, it seems that simply modifying the agent's codebase limits the "action space" and precludes achieving the continuous and open-ended recursive self improvement discussed in the introduction. While I note that fine-tuning the weights of the agent's model has been acknowledged by the authors as something for future work, I do believe that the paper would have been strengthened by examining this approach here as a first instance: potentially on a much simpler task than SWE-Bench and with a much smaller foundation model.
>
> We agree that recent foundation models have advanced dramatically, enabling scaffolds to become simpler on current coding benchmarks. It is possible that in some settings, like current coding benchmarks, certain engineering efforts in scaffolding might be downplayed by the improvement of the Foundation Models. However, many scaffolding components (advanced tools, parallel workflows, external memory, proxy verification, etc.) still fundamentally can not be internalized by Foundation Models and will be essential for more complex real-world tasks beyond today’s benchmarks. In this work, we aim to show the initial evidence in such a direction and inspire follow-up work to further study.  We have added this discussion to the future work section in the revised manuscript.
>
> > L243: The line reading "We evaluate the DGM on two popular benchmarks" reads awkwardly, given the discussion of SWE-Bench and Polyglot immediately prior.
>
> In the revised manuscript, we have moved this sentence to the beginning of the paragraph, before introducing SWE-bench and Polyglot.
>
> > Where does the 'representative agent baseline (Aider)' number come from on the right pane of Figure 2? Is this the score for o3-mini on Aider's old code editing benchmark?
>
> As Aider only released general accuracy scores on the official Polyglot benchmark, we reproduced the baseline using the official Aider code and benchmark to obtain results for each individual task. The reported score on the benchmark is 19.1%, compared to the score we got, 16.4%. We believe adopting either of them does not affect our analysis and conclusion. We will add this description in the revision.

---

> ### Author Response · Authors · 2025-11-22
> **Response to Reviewer G2FT (Part 3/3)**
>
> > Finally, my reading of the transferability results of the DGM-discovered agent across different benchmarks and programming languages at the top of page 8 is less optimistic than the authors put it. The results seem to indicate slight over-fitting to the task the DGM self-improves with, rather than broad-based improvements to the coding agent's capabilities. What is however interesting is that SWE-Bench appears to be a better "teaching" task than polyglot, with the Polyglot-trained agent doing comparatively poorly on SWE-Bench, whereas the SWE-Bench trained agent almost matches the performance of the Polyglot-trained agent on Polyglot.
>
> We believe some degree of task-specific adaptation is indeed expected and even desirable, since fundamentally different types of tasks (e.g., in our case, multi-file Python repository edits vs. primarily single-file, multi-language implementations) naturally require distinct scaffolding components. Crucially, this very property highlights a unique advantage of self-improving systems like the DGM: it replaces laborious manual efforts to design specialized agents for diverse tasks with a fully automated evolutionary process. That said, there is also a clear path to producing agentic systems that generalize: running the DGM on a large, diverse set of tasks to evolve a true generalist agent. We have added this discussion to the future work section in the revised manuscript.
>
> > Looking at the curvature of the light blue 'Average of Archive' line in Figure 3, it appears to be negative from iteration 50 onwards. What do you think is the cause of this plateau, and does this point to a limitation in the open-ended improvement loop (perhaps new idea generation, or the FM's coding ability, or something else entirely)?
>
> The slight negative curvature in the light-blue “Average of Archive” line after iteration ~50 may not be a plateau in discovery capability, but a characteristic of open-ended exploration. By encouraging the discovery of interestingly different agents, our method accepts a higher chance of generating individuals that perform worse than a pure exploitation-based hill-climbing approach (e.g., DGM-greedy ablation in Appendix A.3). Although this diversity-seeking behavior could slow the rate at which the archive *average* improves (or even make it go down), the performance of the best agent continues to rise. The current open-ended loop can be further improved through mechanisms such as archive pruning or smarter parent selection, which are active areas of ongoing research (e.g., LLM-first search, Herr et al., 2025; Huxley Gödel Machine, Wang et al., 2025), but in the long run past research in open-endedness suggests it is good to keep many diverse, high-quality stepping stones around because sometimes they are the key to major innovations that drive performance increases, even if they are initially lower-performing.
>
> > What is the variance within each run? For example, in Figure 2, at times the green line (DGM without self-improvement) is ahead of the full DGM's blue line. While I understand the significant time and monetary costs of doing each run, if each experiment were to be repeated 100 times, do you think this trend would continue?
>
> In Appendix A.4 Line 1305, we show the results of repeatedly running DGM three times on Polyglot. The results have a standard deviation of 2.3% demonstrating the robust trend of improvement.
>
> > In particular, what prevents over-specialization of the agent to the proxy training task (i.e. solving narrow issue-resolution tasks in SWE-Bench's style) which may in time preclude effective self-improvement?
>
> We try to prevent this by explicitly instructing agents to propose only general enhancements to their core capabilities rather than task-specific changes. Furthermore, any over-specialized modifications that do occur typically cause degraded general performance or future self-improvement iterations, and are thus less likely to generate offspring in the evolution algorithm, which favors generally capable agents.
>
> ---
>
> Again, thank you very much for your insightful and detailed comments. We have made many improvements to the paper thanks to your comments and those of the other reviewers. We think it is much improved and hope you agree. We sincerely appreciate your engagement with our work and hope that our revisions and clarifications reinforce your support for this line of research.

---

### Meta-Review · Area_Chair_y5KF · 2025-12-09

**Summary:**

Thanks to the reviewers for their valuable comments from many different perspectives. Overall, I think their main problems at present lie in:



- Innovation is not well described, especially in comparison with existing methods.

- Part of the explanation about the experimental results is not convincing.

- Baseline methods and benchmarks are inadequate.

- The description of the algorithm is not clear.



In addition, some reviewers mentioned the ethical risks of RSI and other issues.

**Reviewer Concerns:**

I appreciate the responses provided by the authors, and I think some of the reviewers' questions will be addressed, such as some explanations of the experimental details. However, due to the limitation of computing resources, the authors did not give positive answers to some reviewers' concerns, but only indicated that future research would be carried out.

After considering the comments of the reviewers and the responses of the authors, I believe that the overall content of this paper is clear, and the 'Open-Ended Evolution' proposed is also interesting, which should arouse the interest of some practitioners. But as the reviewers noted, some of the innovative statements seem overstated or do not take into account the actual cost constraints.



Overall, I think the paper can convince most of the reviewers and is above the acceptance threshold. I hope to see further improvements in the approach in the future.

**Reviewer Scores:**

For Reviewer G2FT, clarification about some of the details is acceptable, but explanations about the experimental setting may not work. I think this reviewer will maintain the score (**Rating:** 6).



For Reviewer 3Ycd, most of the explanations were acceptable. I think this reviewer will maintain the score (**Rating:** 8).



For Reviewer Cgiz, doubts about innovativeness may still exist. I think this reviewer will maintain the score (**Rating:** 4).



For Reviewer VRNN, doubts about innovativeness may remain. I think this reviewer will maintain the score (**Rating:** 4).

---

### Decision · Program_Chairs · 2026-01-26

Accept (Poster)